# Beyond Similarity for Personalization: User Memory Selection via Response-Utility Optimization

## Abstract

A common approach to personalization in large language models (LLMs) is to incorporate a subset of the user memory into the prompt at inference time to guide the model's generation. Existing methods to select these subsets primarily rely on similarity between user memory items and input queries, ignoring how features actually affect the model's predictive distribution. We propose **R**esponse-**U**tility optimization for **M**emory **S**election (RUMS), a novel user memory selection method, inspired by Bayesian Optimal Experimental Design, that directly quantifies how much each memory item reduces uncertainty in the model's response distribution. RUMS measures mutual information between a subset of user memory and model outputs to identify items that sharpen predictions beyond semantic similarity. Even more, RUMS, by design, automatically selects if personalization is beneficial at all. We demonstrate that this information-theoretic foundation enables more principled user memory selection that aligns more closely with human selection compared to state-of-the-art methods, and models 400x bigger. Additionally, we show that memory items selected using RUMS result in better response quality compared to existing approaches, while having up to 95% reduction in cost.

## 1 Introduction

Personalization has long been recognized as a key component in effective human–computer interaction, and large language models (LLMs) are no exception. Users increasingly expect models to remember relevant details and adapt responses to their unique preferences. To meet this demand, current approaches to personalization generally fall into two categories: training-based methods and inference-time methods. Training-based approaches, such as low-rank preference models (Bose et al., 2025) or parameter-efficient fine-tuning modules (Tan et al., 2024), adapt the model parameters for individual users. Although these methods can be effective for small populations, they become computationally prohibitive when scaled to many users. As a result, inference-time personalization, such as including user-specific context in the input prompt, is often the preferred solution.

However, inference-time methods raise a central question: **What information, if any, should be included to guide the model?** A naïve strategy is to include all stored user memories in the prompt. However, this often reduces performance, as irrelevant or distracting information may overwhelm the model and produce responses that are less accurate or sound unnatural (Li et al., 2023; Rajeev et al., 2025). Moreover, this strategy becomes increasingly impractical as the memory size grows. Therefore, existing methods instead attempt to select and retrieve relevant subsets of the user memory by measuring the relationship between a *memory item and the input query*. In other words, these methods try to pick out only the user memory items that seem most related to the user's current request. This is often done by heuristic retrieval (Lù, 2024), surface-level semantic similarity (Lei et al., 2023), or prompting LLMs (Handa et al., 2024). However, we argue that these approaches are fundamentally limited because they do not assess a *memory item's direct impact on the model's response* distribution. Instead, they rely on surface-level signals, making them more prone to missing important connections (under-selecting) or including superfluous ones (over-selecting). This can introduce noisy or irrelevant context that degrades response quality while also increasing latency, cost, and privacy risks (Hagström et al., 2025). These shortcomings highlight the need for lightweight, adaptable, response-aware methods that capture the true influence of user items on model outputs.

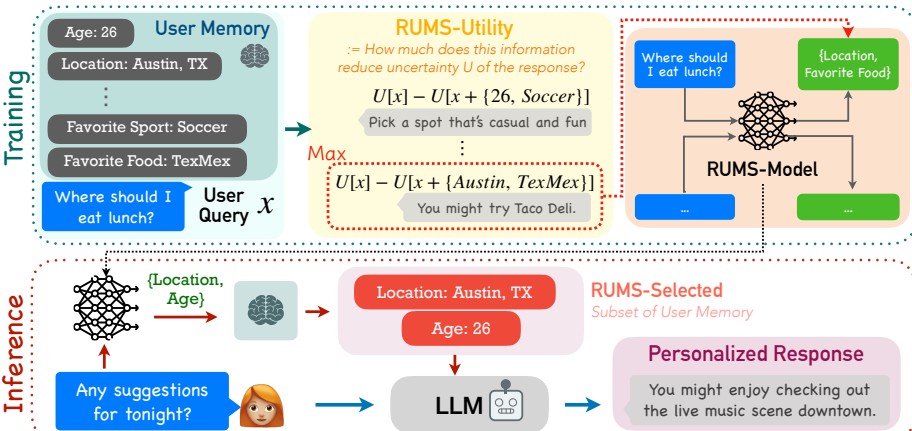

Figure 1: Overview of RUMS. RUMS-Utility quantifies how much memory subsets reduce response uncertainty (training phase), while RUMS-Models efficiently selects informative items at inference.

To bridge this gap, we introduce Response-Utility for Memory Selection (RUMS), which comprises of two key components: a novel utility function (*RUMS-Utility*) that quantifies the informativeness of user memory subsets, and a lightweight model (*RUMS-Model*) that efficiently approximates this utility computation at inference time.

*RUMS-Utility* draws inspiration from Bayesian Optimal Experimental Design (BOED) to directly measure how much each subset of user memories reduces uncertainty in the model's response distribution. Specifically, it computes the conditional mutual information between candidate memory subsets and the model's output distribution given the input query. This information-theoretic approach fundamentally differs from prior similarity-based methods, as rather than selecting memories that are semantically related to the query, it identifies those that meaningfully sharpen the model's predictions. The utility function also naturally handles the decision of whether personalization should be applied at all: when no memory subset achieves sufficient utility (i.e., does not reduce response uncertainty beyond a threshold), the system forgoes personalization entirely.

However, computing *RUMS-Utility* for all queries is computationally expensive, particularly with large user profiles. Therefore, we use the utility function during an offline training phase to generate high-quality training data in the form (user query, optimal memory subset). We then train *RUMS-Models*, lightweight encoder-classifiers based on DeBERTa (He et al., 2021), to directly predict memory items that are most informative given a query. This learned approach preserves the principled selection quality of the utility function while enabling efficient deployment. We show that RUMS enables more principled memory item selection that is more closely aligned to human selection compared to state-of-the-art (SOTA) memory selection methods and models $400\times$ its size. We also show that the items selected using RUMS result in better response quality than other methods, while lowering the cost of personalization overall.

Our work offers novel conceptual, theoretical, and empirical contributions to personalization:

- We propose RUMS, a lightweight, response-based method for user memory selection that directly accounts for the impact of memory items on model outputs.
- We introduce a principled utility function RUMS-Utility, designed to capture personalization benefit signal and can be used to support downstream training of a RUMS-Model.
- We show that RUMS (i) reliably identifies when personalization is beneficial, aligning closely with human judgments, (ii) selects memory subsets that align more strongly with human annotations than SOTA retrieval and prompting baselines, and (iii) improves generation quality over SOTA baselines and $400\times$ larger models on synthetic and real-world datasets, while adapting to underlying model capabilities and costing up to 95% less than other methods.

## 2 RELATED WORKS

**User Memory for LLM Personalization** Memory augmentation enables personalization in LLMs by retrieving subsets of user memory at inference time to guide responses. Simple approaches prepend user context to prompts, such as profile summaries (Richardson et al., 2023; Pan et al.,

2025) or past interactions (Kang et al., 2023; Liu et al., 2023). However, as the amount of user memory grows, these methods quickly become intractable. Due to this, there has been an increased trend in selecting a *subset* of the user memory to prepend. For example, Chen (2025) retrieves relevant historical items via semantic similarity for recommendations, while LaMP (Salemi et al., 2023) investigates personalized sentence retrieval. PEARL (Mysore et al., 2024) and ROPG (Salemi et al., 2024) train retrieval components with user signals to improve personalization. For a broader overview, see Liu et al. (2025). However, these methods rely on the similarity between user input and memory item, without considering their impact on the model's downstream response.

**Selective Personalization** A related line of reserach is selective conditioning or abstention which studies how models can ignore user memory or abstain from predictions when inputs are uninformative. Geifman & El-Yaniv (2017) introduce selective classification for neural networks, and recent work extends this to language models, proposing mechanisms to ignore input features or context that may degrade performance (Hagström et al., 2025). In memory selection, this corresponds to sometimes choosing no subset when personalization is unnecessary. Prior work typically relies on global thresholds (Li, 2023) or static policies (Joren et al., 2023), whereas our method learns a sequential, adaptive policy that dynamically selects which items to condition on based on the input and context.

**Information-theoretic Uses in NLP** Information-theoretic ideas have long informed NLP (Brown et al., 1992), spanning multimodal learning (Almudévar et al., 2025), information control (Ravfogel et al., 2020), interpretability (Chen et al., 2018), and prompt design (Zhang et al., 2022). In feature selection, EDDI (Ma et al., 2019) estimated conditional mutual information for individual features, while later work extended this to dynamic or sequential selection (Covert et al., 2023; Gadgil et al., 2024). Recent methods also use differences in next-token likelihood with and without context to guide generation through distribution editing (He et al., 2025). While much of this work has focused on domains such as image classification and fraud detection, the closest parallel to personalization arises in recommendation systems using Bayesian active learning to select a model query to ask the user, with the goal of improving recommendation quality (Handa et al., 2024). We draw on these principles to select memory items that maximize information gain for personalized responses.

## 3 METHOD

We aim to develop a method that selects a subset of user memory that maximizes the utility of the response generated with these items for the user.

**Problem Formulation** Consider user $u$ with query $x$ (e.g. "Where should I eat lunch?"). We let $\theta$ denote an LLM that generates an output $y = (y_1, ..., y_T)$ of size $T$ using distribution $p_\theta(y|x)$. We measure the quality of the response $y$ by utility to the user $U_u(y)$. We consider a set of memory items $\mathcal{M}$ (e.g. Location), in which user $u$ has specific values $m_u$ (e.g. Location: Austin, TX). Our aim is to select the memory subset $S \subseteq \mathcal{M}$ with instantiated values $m_{u,S} = \{m_u^{(a)} : a \in S\}$, which when given to the model with the input $x$ yields the highest response utility. Formally,

$$S^* = \arg \max_{S \subseteq \mathcal{M}} \mathbb{E}_{y \sim p_\theta(\cdot|x, m_{u,S})} \big[ U_u(y) \big]. \tag{1}$$

### 3.1 RUMS: RESPONSE UTILITY OPTIMIZED MEMORY SELECTION

Personalization is fundamentally an information acquisition problem: from a large space of user data, we must select the pieces that most improve, rather than distract from, the model's response. A key insight, is that this parallels a well-studied problem in information-theoretic, Bayesian Optimal Experimental Design (BOED) which is choosing experiments that provide the most information about an unknown parameter. Both tasks involve selecting informative observations from a large space to maximize utility under uncertainty.

Formally, in BOED, for unknown parameter $\phi$, design $s$, and potential outcome $y$, the optimal design is selected as $s^\star = \arg \max_s \Omega(\phi; s, y)$, where $\Omega$ is the utility function. However, since the actual outcome $y$ is unknown in advance, BOED instead maximizes the *expected* information gain (EIG):

$$\text{EIG}(s) := \mathbb{E}_{p(y|s)}[\Omega(\phi; s, y)] = \mathbb{E}_{p(y|s)} \big[ H_\phi(y) - H_\phi(y|s) \big], \tag{2}$$

where H is the Shannon entropy (definition in Appendix D.1).

Memory selection can thus be reframed as a *personalization* design problem, where each subset of user memory items serves as an experimental design and the model's response distribution as the outcome, transforming ad-hoc selection into a principled, theory-grounded optimization.

### 3.1.1 RUMS UTILITY FUNCTION

Applying BOED to LLM personalization poses three critical challenges. First, unlike classical BOED, which infers latent parameters, we aim to optimize the model's predictive distribution directly, focusing on better responses rather than user preferences. Second, LLMs' exponentially large, variable-length output spaces makes standard BOED computations intractable. Third, we require a method which also detects when personalization truly improves responses rather than adding noise. Our approach addresses all three through a novel BOED adaptation.

**Challenge 1: From parameter inference to distribution shaping.** Classical BOED methods aim to reduce uncertainty about an unknown parameter $\phi$ through experimental design $s$ (Equation (2)). In our setting, however, there is no latent parameter to infer. Instead, we want to shape the model's distribution over responses $y$ to be more precise and relevant. We address this difference by introducing a novel utility function based on the reduction in predictive entropy:

$$\text{EIG}(S, u) = \mathbb{E}_{p_\theta(y|x)} \left[ H_\theta(y \mid x) \right] - \mathbb{E}_{p_\theta(y|x, m_{u,S})} \left[ H_\theta(y \mid x, m_{u,S}) \right] \tag{3}$$

we use $H_\theta$ to be the Shannon entropy over the output distribution using the model $\theta$.

The key idea is that a memory subset $S$ is valuable to the extent it reduces uncertainty in the model's predictions for user $u$. For instance, for the query "where should I eat lunch?", without user information, the model spreads probability across countless plausible restaurants, yielding high entropy. However, adding even a single relevant fact, such as "Location: Austin, TX" or "Dietary preference: Vegan", sharply concentrates probability on relevant options. Therefore, by measuring this entropy reduction, the utility function directly quantifies how much $S$ sharpens predictions.

We note that some memory items are highly dependent on specific user values (e.g., Dietary Preference: Vegan vs. None), while others are informative regardless of value (e.g., Location). When the exact user is known, this information is applied directly; when unknown, averaging over many user profiles allows the method to transition from user-specific to user-agnostic.

**Challenge 2: From intractable to tractable.** However, when using LLMs, directly computing equation 3 is intractable because $y$ is a sequence of length $T$, and the entropy requires summing over the exponentially many possible outputs. We address this issue through a key theoretical insight.

**Proposition 1.** *The entropy of the sequence distribution can be decomposed as the sum of conditional token entropies. Given $p_\theta(\cdot \mid y_{<t}, x)$, the conditional next-token distribution given prefix $y_{<t}$,*

$$H_\theta(y \mid x) \coloneqq \mathbb{E}_{y \sim p_\theta(\cdot|x)} \left[ -\log P_\theta(y \mid x) \right] = \mathbb{E}_{y_{<t} \sim p_\theta(\cdot|x)} \left[ \sum_{t=1}^{T} H\big( p_\theta(y_t \mid y_{<t}, x) \big) \right].$$

This follows directly from the chain rule of probability and the law of iterated expectations (proof in Appendix E).

Although Proposition 1 reduces the problem to token-level entropies, the outer expectation over prefixes $y_{<t}$ remains intractable. We therefore employ Monte Carlo sampling, generating N sequences and computing:

$$\hat{H}_\theta(y \mid x) = \frac{1}{N} \sum_{i=1}^{N} \left[ \frac{1}{T} \sum_{t=1}^{T} H\big( p_\theta(y_t^i \mid y_{<t}^i, x) \big) \right]. \tag{4}$$

Here, $y^i = (y_1^i, \ldots, y_T^i)$ denotes the $i$-th sampled sequence. We note that $T \equiv T_N$ if token sizes is not fixed per sample. This approximation converges to the true per-token sequence entropy as $N \to \infty$ under i.i.d. sampling from $p_\theta(\cdot \mid x)$ with finite expectation. In practice, however, sequence samples are not strictly i.i.d., and factors such as constrained decoding or approximate distributions can introduce bias. Despite these limitations, we find that using $N = 5$ samples provides a good balance between accuracy and efficiency. For more empirical analysis, see Appendix B.1.

An analogous estimate is computed for $\hat{H}_\theta(y \mid x, m_{u,S})$ in order to approximate Equation (3). Together, we use the estimated EIG for personalization,

$$\widehat{\text{EIG}}(S, u) \coloneqq \left[ \hat{H}_\theta(y|x) - \hat{H}_\theta(y|x, m_{u,S}) \right] \tag{5}$$

Then, our final objective selects the memory subset that maximizes this utility:

$$S^*(x, u) = \arg\max_{S \subseteq M} \widehat{EIG}(S, u). \tag{6}$$

This provides an information-theoretic approach to memory selection that directly measures impact on response quality. For clarity, we define the maximum utility score for a given input $x$ and user $u$ as RUMS-Utility and the subset of user memory items selected by Equation (6) as RUMS-Subset.

**Challenge 3: Automatic detection of when personalization helps.** Unlike prior methods that always retrieve information, our formulation naturally identifies when personalization is unnecessary. When no memory subset achieves sufficient entropy reduction (i.e. $\widehat{EIG}(S) < \tau$ for threshold $\tau$), the system selects the empty set, preventing irrelevant information from degrading response quality. This built-in selectivity eliminates the need for separate pre-filtering mechanisms, reducing computational overhead in deployment.

### 3.1.2 RUMS-MODELS: LEARNING TO APPROXIMATE UTILITY

While RUMS offers principled memory selection, computing entropy reductions for all possible subsets at inference is prohibitive, especially with large user profiles. We address this via amortized learning, where RUMS generates high-quality training data offline, and a lightweight neural model is trained to predict optimal memory selections directly at inference.

**Training Data Generation** Our training pipeline uses RUMS to generate (user query, optimal memory subset) pairs, with a threshold $\beta$ determining whether a subset provides enough entropy reduction to be selected.

1. *Profile Generation and Candidate Reduction*: RUMS relies on how memory item *values* sharpen response distributions, so a set of plausible user profiles $u$ is needed to estimate utility. When exact profiles are unavailable or a more general framework is desired, a diverse population-like set can be generated or collected. Optionally, for very large profiles (e.g., $\geq$50 items), a larger LLM can first filter them to a smaller candidate set of $k$ items.
2. *Utility Computation*: For each candidate subset of memory, we compute the estimated utility score $\widehat{EIG}(S, u)$ given a random user profile $u$. To capture interaction effects between memory items, we evaluate subsets rather than individual items, revealing more utility than the sum of their individual contributions.
3. *Optimal Memory Subset Selection*: We select the subset with maximum utility above threshold $\beta$, or optionally the empty set if none qualify. Tuned on validation data, $\beta$ enables automatic abstention from personalization when it would not improve response quality (Appendix G.5).

Critically, we pair each query with many diverse user profiles, yielding a richer training set that supports user-agnostic selection policies. By observing how the same query benefits from different memory items depending on specific user values, the model learns to identify genuinely informative items rather than spurious correlations.

**Model Architecture and Training** We train RUMS-Models as lightweight classifiers to approximate the costly utility computation. For dynamic memory schemas, where item types vary across users, a binary classifier $f(x, m_i) \to \{0, 1\}$ evaluates each memory item $\{m_i\}_{i=1}^n$ independently, handling arbitrary structures but requiring $n$ forward passes. For static schemas with $d$ fixed item types, a multiclass classifier $f(x) \to \{0, 1\}^d$ outputs all selections in a single pass, preferred for efficiency. The resulting RUMS-Model matches the inference speed of simple similarity-based retrieval while retaining principled selection quality, making it practical for large-scale deployment.

## 4 RESULTS

In this section, we evaluate RUMS, our proposed response-utility method for memory selection. For clarity, we define three key outputs from RUMS. First, *RUMS-Utility* represents the maximum utility score computed using Equation (5). Second, *RUMS-Subset* is the subset of memory items chosen according to Equation (6). Finally, *RUMS-Models* are lightweight models trained on the data generated by RUMS. We evaluate our approach along three dimensions:

H1: The utility scores from RUMS-Utility provides a reliable signal for distinguishing between inputs that benefit from personalization and those that do not.

H2: The subsets selected, RUMS-Subset, contain more informative items than state-of-the-art baselines and larger models. Furthermore, the utility signal is strong enough that models trained on them (RUMS-Models) also outperform baselines and larger models.

H3: Memory subsets selected with RUMS-Models improve downstream response generation compared to state-of-the-art baselines and larger models.

## 4.1 EXPERIMENTAL SETUP

Across all experiments (H1–H3), we consider $n = 50$ static user memory items (e.g., *location*, *age*, *name*), and assumes that all values are given for each user (see Appendix G.3 for list of items). The task is to select up to $5$ items that are most relevant for answering a given user input $x$.

**Datasets.** To test both instances that would and would not benefit from personalization, we use a diverse mixture of possible user inputs. We evaluate across 3 datasets:

- **PersonaFeedback** (Tao et al., 2025): A synthetic dataset of user inputs generated by an LLM using randomly sampled, unique user memory data and scenarios. This process ensured that each input relied on personal details and would benefit from personalization. The original data is in Chinese, so we translated with a translation model (Tiedemann & Thottingal, 2020). Example: *"Do you have any music for the trip?"*

- **FreebaseQA** (Jiang et al., 2019): A dataset of trivia-style questions and answers. Example: *"What type of snake is a sidewinder?"*

- **WildChat** (Zhao et al., 2024): Real-world ChatGPT interactions (English only, non-toxic, and request-based). We use only the first human-written input. Examples: *"im hungry"; "who is Mohamed Gamal Mohamed Maniieh"*

By nature of the data, we classify PersonaFeedback as personalized inputs and FreebaseQA as non-personalized inputs, and refer to these together as the "Synthetic" dataset. For WildChat, two annotators labeled each input as personalized or non-personalized (agreement required). We denote these subsets as WildChat-Personal and WildChat-NonPersonal, which form the "Real-World" dataset.

**User Profiles.** User profiles are required for utility computation and evaluation. We first prompted GPT-4 (OpenAI et al., 2024) to generate 50 common user memory items. We then used the model again to generate consistent user profiles without contradictions (e.g., "vegan" with "favorite food = steak"). We generated $n = 100$ unique profiles each for training and testing. Details on prompts used, distributions of memory item values and generation process are provided in Appendix G.3.

**RUMS-Model Training** To create training data with RUMS, we used Llama-3.1-8B (AI, 2024b) to compute the entropy. Following the procedure in Section 3.1.2, GPT-4 was first prompted to reduce the candidate memory items from 50 to at most 10 for each query before computing the utility scores. The utility scores were estimated with $n = 5$ Monte Carlo samples. Lastly, a threshold of $t = 0.29$, determined on a held-out validation set, was applied to choose between the subset with maximum utility or the empty set (i.e., no personalization). We use $n = 2.5K$ from PersonaFeedback and 2K from FreebaseQA for training. Since both datasets are also used for evaluation, we rely on held-out splits to avoid overlap and interpret results as in-domain performance. To assess generalization, we also evaluate on WildChat, a real-world dataset from a different distribution.

We train two RUMS-Model variants: The first, RUMS-Binary, is a binary classifier trained on individual (query, memory value) pairs. Because each memory value is treated separately and negative labels dominate, this model tends to under-select, making it better suited for noise-sensitive settings. The second, RUMS-Multi, is a multiclass classifier trained on denser labels constructed by merging the selected subsets from the $u = 5$ user profiles per query. In this setup, all memory items chosen using any of the five profiles were marked as significant, encouraging broader selection. As a result, the model tends to over-select, which is advantageous for more robust models that can filter noise. Both are based on DeBERTa-v3-large (He et al., 2021). Full details are in Appendix G.5.

**Baselines** Current approaches to memory selection primarily rely on two strategies: semantic similarity-based retrieval and large language model prompting. To comprehensively evaluate against existing methods, we employ baselines representing both approaches: (1) **Semantic Similarity**: We compute cosine similarity between input and memory item embeddings using a sentence transformer (Wang et al., 2020), representing the dominant paradigm of surface-level similarity matching in cur-

Table 1: Recall (accuracy on personalized) and specificity (accuracy on non-personalized) comparison across datasets and methods. **Bolded** is the highest value in each category.

| Dataset | Recall (Personalized) | | | | Specificity (Non-Personalized) | | | |
|---|---|---|---|---|---|---|---|---|
| | RUMS-Utility | GPT4-Few | GPT4-Zero | Semantic Similarity | RUMS-Utility | GPT4-Few | GPT4-Zero | Semantic Similarity |
| Synthetic (n=20) | 94.4 | **100.0** | 33.3 | **100.0** | **94.4** | 50.0 | 50.0 | 0.0 |
| Real-World (n=31) | **92.3** | 76.9 | 53.9 | 7.7 | 96.0 | **100.0** | **100.0** | **100.0** |

rent retrieval systems, (2) **GPT-4 Prompting**: We use direct prompting of GPT-4 (*zero-shot* and *few-shot* - up to 4 in-context examples) as a representative LLM-based approach and upper bound on model capability, given its 400× larger size compared to RUMS-Models. For H3, which evaluates downstream generation quality, we thus additionally include established retrieval methods and naive baselines to ensure comprehensive coverage: (3) **SOTA Retrieval Methods**: We evaluate *BM25* (Lù, 2024) and *ReContriever* (Lei et al., 2023) as representative SOTA retrieval approaches for memory selection, (4) **Naive Baselines**: We include *None* (no items), *All* (50 items), and *Random* (5 items) to establish performance bounds and test robustness to different selection strategies.

### 4.2 H1: DOES RUMS-UTILITY DISTINGUISH PERSONALIZED FROM NON-PERSONALIZED INPUTS?

We hypothesize that entropy reduction when conditioning on user memory reflects its importance. Thus, inputs that do not benefit from personalization should yield low maximum utility scores (i.e., the maximum utility score across all subsets of candidate memory items for a specific user), while personalized inputs should yield high ones. To test this, we compare the maximum utility scores on a known non-personalized (FreebaseQA) and a known personalized (PersonaFeedback) datasets. We sampled 500 inputs per dataset and computed the maximum utility score using Equation (5) for each input. The resulting distributions were compared using divergence metrics and hypothesis testing.

**Utility scores separate personalized from non-personalized inputs.** In Figure 2 we show that personalized inputs exhibit much higher utility scores (mean = 0.50, std = 0.12) than non-personalized ones (mean = 0.12, std = 0.09). The Jensen–Shannon divergence between distributions is high at 0.66, and a two-sample Kolmogorov–Smirnov test yields a statistic of 0.82 ($p < .001$), strongly rejecting the null hypothesis of equal distributions. These results indicate that maximum utility scores reliably separate personalized from non-personalized inputs. For further exploration on hyperparameter selection and it's effects on the robustness of these results see Appendix B.1.

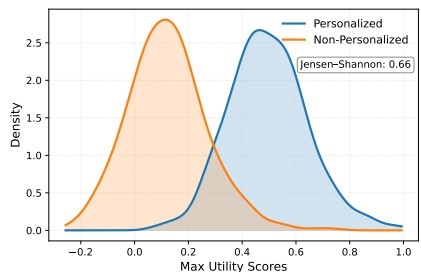

Figure 2: Distribution of maximum utility scores for personalized (PersonaFeedback) and non-personalized (FreebaseQA) inputs.

#### 4.2.1 H1′: DO UTILITY SCORES ALIGN WITH HUMAN JUDGMENTS OF PERSONALIZATION?

We next test whether personalization decisions made by utility score thresholding aligns with human judgment. We randomly sampled $n = 25$ inputs from each of PersonaFeedback, FreebaseQA, WildChat-NonPersonal, and WildChat-Personal. Four human annotators labeled whether each input benefits from personalization, which yielded an agreement between the parties of Cohen's $\kappa = 0.59$; we retained only inputs with unanimous agreement ($n = 64$).

In order to get a robust signal using RUMS, we used $n = 10$ Monte Carlo samples and $u = 10$ randomly selected user-profiles. We used the average maximum utility score over the profiles as RUMS-Utility. We then tuned a threshold on 20% validation data. We compared RUMS-Utility against GPT-4 (prompted to determine personalization necessity) and semantic similarity baselines, and the performance was measured by recall (accuracy in identifying personalized inputs) and specificity (accuracy in identifying non-personalized inputs). If both humans and the model selected no items, the sample was counted correct for both metrics.

Table 2: Precision, recall, and F1 scores of memory items selected compared to human gold label annotation across datasets and methods.

| Dataset | Metric | RUMS-Subset | RUMS-Binary | RUMS-Multi | Semantic Sim. | GPT4-Zero | GPT4-Few |
|---------|--------|-------------|-------------|------------|---------------|-----------|----------|
| Synthetic | Precision | **0.61** | 0.60 | 0.23 | 0.09 | 0.26 | 0.53 |
| | Recall | 0.70 | **0.89** | 0.39 | 0.23 | 0.42 | 0.65 |
| | F1 | 0.70 | **0.78** | 0.31 | 0.14 | 0.35 | 0.64 |
| Real World | Precision | **0.48** | 0.32 | 0.30 | 0.17 | 0.39 | 0.42 |
| | Recall | 0.36 | **0.60** | 0.38 | 0.27 | 0.52 | 0.39 |
| | F1 | **0.53** | 0.45 | 0.35 | 0.20 | 0.43 | 0.39 |

Table 3: Average win rate of our method against baselines on Synthetic (left) and Real World (right) datasets. Cells in green mark cases where our method outperforms the baseline, and cells in red mark cases where the baseline does better. Values with magnitude above 0.1 are **bolded** to highlight strong differences. Results are reported for two response models (LLaMA 70B Instruct and GPT-4) and two RUMS variants: RUMS-Binary (fewer items) and RUMS-Multi (more items).

| | LLaMA 70B | | GPT-4 | | | LLaMA 70B | | GPT-4 | |
|---|---|---|---|---|---|---|---|---|---|
| | Binary | Multi | Binary | Multi | | Binary | Multi | Binary | Multi |
| None | **0.25** | **-0.13** | **0.26** | **0.36** | None | 0.05 | 0.03 | **0.20** | **0.17** |
| All | **0.16** | **-0.19** | **-0.16** | 0.02 | All | -0.04 | **-0.10** | **-0.14** | -0.06 |
| Random | **0.34** | **0.13** | 0.06 | **0.19** | Random | **0.11** | **0.13** | 0.10 | 0.04 |
| Semantic Sim. | **0.27** | 0.09 | -0.03 | 0.05 | Semantic Sim. | -0.03 | 0.00 | -0.02 | 0.00 |
| BM25 | **0.19** | **-0.13** | **0.20** | **0.32** | BM25 | 0.01 | 0.05 | **0.18** | **0.13** |
| ReContriever | **0.27** | -0.03 | -0.04 | 0.03 | ReContriever | 0.06 | 0.01 | -0.05 | **0.13** |
| GPT4-Few | **0.18** | **-0.16** | -0.09 | -0.01 | GPT4-Few | 0.00 | -0.08 | 0.04 | **0.11** |

**RUMS-Utility correlate more strongly with human judgments.** As shown in Table 1, RUMS-Utility outperforms GPT-4 and semantic similarity baselines. Compared to GPT-4 FewShot, it achieves +7% recall and +15% specificity overall. Gains are strongest on synthetic data, with competitive results on real-world. Other baselines show imbalanced behavior (e.g., GPT-4 ZeroShot often avoids personalization, giving high specificity but low recall). In contrast, utility scores perform well on both metrics, reflecting their ability to capture each item's true effect on generation.

### 4.3 H2: Does RUMS-Subset Better Match Human Judgment?

Building on H1, we now ask whether our method identifies which specific memory items help personalization. We randomly selected $n = 15$ samples from each of PersonaFeedback, FreebaseQA, WildChat-NonPersonal, and WildChat-Personal for a total of $n = 60$. As before, four annotators selected any of 50 possible items (including "No Items") they judged useful, yielding $41\%$ majority ($k \geq 3$) agreement. While moderate, this level of agreement is expected given the subjective nature of the task. We therefore considered an item as selected by humans if a majority agreed, resulting in an average of 4.19 items selected per user query for inputs where at least one item was chosen.

We evaluated (i) direct selection from utility scores RUMS-Subset (up to 5) and (ii) RUMS-Models, the trained models described in Section 4. We compare RUMS-Subset and RUMS-Models against GPT-4 (prompted to select up to five relevant memory items, both zero- and few-shot) and semantic similarity (selecting up to five highest-scoring items with cosine similarity $> 0$). Performance was measured with precision, recall, and F1 against human labels.

**RUMS-Subset aligns more closely with human annotation.** As shown in Table 2, the two methods based on RUMS achieve on average $20\%$ higher precision, $31\%$ higher recall, and $9\%$ higher F1 compared to the baselines across both synthetic and real-world datasets. Semantic similarity performs worst, reflecting reliance on surface overlap rather than downstream utility. GPT-4 shows strong recall but substantially lower precision, indicating that it tends to over-select. In contrast, the results show that the raw utility scores, RUMS-Subset, themselves provide strong signal for identifying useful memory items, and RUMS-Models achieves similar F1, demonstrating that small models with only modest amounts of data can learn effective memory selection from utility scores. For examples of memory items selected see Appendix B.3

### 4.4 H3: DO ITEMS SELECTED BY RUMS-MODELS IMPROVE DOWNSTREAM GENERATION?

Having established that RUMS makes effective decisions about when and what to personalize (H1-H2), we now demonstrate that these selections translate to tangible improvements in response quality. This tests our core claim: principled selection leads to better personalization.

**Experimental Setup.** We evaluate RUMS on both synthetic (PersonaFeedback and FreebaseQA, 100 samples each) and real-world datasets (WildChat, 50 personalized and 50 non-personalized) to assess generalization. Responses are generated using two models: GPT-4, a high-performing model with robust capabilities, and LLaMA 70B Instruct (AI, 2024a), a smaller open-source model that is relatively more susceptible to noise. We evaluate response quality using RUMS-Models against all baselines described in Section 4.1. Performance is measured by GPT-4 win rate using chain-of-thought prompting and confidence thresholding (Appendix G.7).

**RUMS improves response quality across diverse settings** Table 3 reports generation win rates comparing our method against the baselines synthetic and real-world datasets. On the synthetic dataset, RUMS-Binary consistently outperforms all baselines, achieving at least a 15% higher win rate when responses are generated using the less robust LLaMA 70B model. Most notably, it has an 18% winrate over GPT4-Fewshot, a model 400× it's size. Most of these gains come from non-personalized inputs, where a lower-capability model struggles to ignore noisy items, reducing its performance. In contrast, when using a more capable model such as GPT-4 for response generation, RUMS-Multi outperforms almost all baselines, primarily driven by personalized inputs. In this setting, the model can leverage more items, including potentially noisy ones, while still filtering out unhelpful items to a large extent. Real-world results follow the same trend. RUMS-Binary achieves a 12% higher win rate than the best SOTA baseline, ReContriever with LLaMA 70B, while RUMS-Multi exceeds GPT4-FewShot by 11% and BM25/ReContriever by 13% with GPT-4. The *All* baseline performs reasonably with only 50 items, but prior work shows scaling degrades performance (Rajeev et al., 2025). See Appendix B.5 for breakdowns.

| Method | Synthetic | Real World |
|---|---|---|
| All | 50.00 | 50.00 |
| None | 0.00 | 0.00 |
| Random | 5.00 | 5.00 |
| Semantic Similarity | 4.63 | 4.79 |
| BM25 | 0.60 | 0.35 |
| ReContriever | 5.00 | 5.00 |
| GPT4-Few | 2.80 | 3.00 |
| RUMS-Binary | 1.58 | 2.22 |

(a) Average number of memory items by method and dataset type.

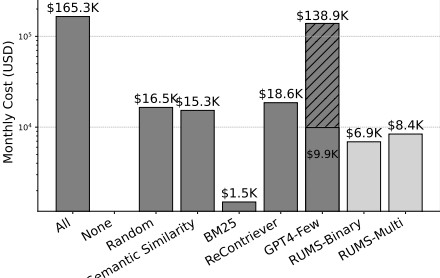

(b) Estimated additional average monthly cost by method. Diagonal line indicate added cost of LLM calls for memory selection.

Figure 3: Cost Analysis

**RUMS-Models reduces cost compared to other under-performing methods.** Because personalization may be applied to every model input, efficiency is critical. We track both the number of memory items selected and the estimated monthly inference cost, computed as the extra tokens from selected items multiplied by a pricing model of $0.01 per 1K tokens for one million daily queries (current GPT-4 rates). Figure 3a shows that RUMS-Models select substantially fewer memory items than most baselines, translating to significant cost savings (Figure 3b): RUMS-Binary costs $6.9K/month and RUMS-Multi costs $8.4K/month, compared to $165.3K for All and $18.6K for the competitive ReContriever. Both RUMS variants are cheaper than GPT4-FewShot, even when the additional cost of memory subset selection required by GPT4 baselines is excluded. Thus, RUMS balances informativeness and efficiency, enabling scalable personalization.

## 5 CONCLUSION

We propose RUMS, a new method for memory subset selection grounded in information theory that provides a lightweight, adaptable solution for large-scale personalization. Through extensive experimentation, we demonstrate that RUMS reliably determines when personalization is beneficial, selects memory items that align with human preferences, and improves response generation quality. Importantly, RUMS achieves these results at up to 95% lower computational cost than existing methods, enabling scalable large-scale personalization.

# 6 ETHICS

We believe this research presents minimal ethical concerns. However, like other methods that steer or guide model generations, our approach could potentially introduce unintentional biases, reinforce stereotypes, or produce harmful outputs. We therefore recommend that practitioners implement comprehensive evaluation protocols before deployment, including stress testing, automated safety evaluations, human assessments, and ongoing monitoring systems. Additionally, since our method incorporates user information during inference, it introduces potential privacy and security vulnerabilities that require careful consideration in production environments.

# 7 REPRODUCIBILITY

To ensure reproducibility, we provide comprehensive supporting materials alongside this work. A complete proof of our main proposition is available in Appendix E, and we will release all code for implementing RUMS and reproducing our experiments via a public GitHub repository. Detailed experimental specifications are provided in Appendix G, including dataset descriptions (Appendix G.2), profile generation procedures (Appendix G.3), baseline configurations and hyperparameters (Appendix G.4), our method's hyperparameters (Appendix G.5), evaluation protocols (Appendix G.7), and software/hardware specifications (Appendix H).

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

# A    APPENDIX

# B    ADDITIONAL EXPERIMENTATION

In this section, we present additional experiments to supplement our main results.

## B.1    ROBUSTNESS ANALYSIS OF UTILITY SCORES

In H1, we hypothesized that RUMS-Utility, the maximum utility scores, can reliably distinguish between user inputs that would benefit from personalization and those that would not. Here, we further investigate how hyperparameters in the RUMS-Utility algorithm affect this reliability and provide justification for these effects. Specifically, we examine the influence of model type, model size, decoding method type, maximum number of tokens generated, number of Monte Carlo samples, and number of user profiles. For each analysis, all other variables are held constant using the LLaMa base model, 8B size, sampling decoding, $T = 20$ tokens, $n = 1$ sample, and averaged over $n = 3$ user profiles.

**Hyperparameter: Model Type**    We first compare the LLaMa-8B base model to an LLaMa-8B instruction-tuned model. As shown in Figure 4, the base model produces more separated distributions of utility scores than the instruct model, with a Jensen-Shannon divergence score of 0.56 compared to 0.32 for the instruct model. This is consistent with prior research showing that instruction-tuning tends to pre-sharpen output distributions (Sorensen et al., 2024). Based on this observation, we chose to use the base model for our experiments.

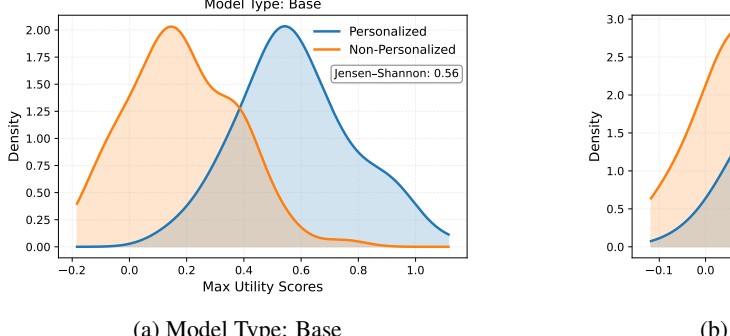 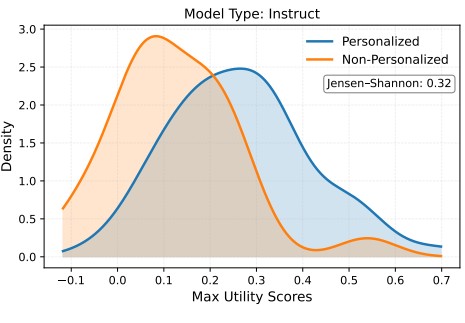

(a) Model Type: Base                          (b) Model Type: Instruct

Figure 4: Distribution of maximum utility scores for a personalized (blue) and non-personalized (orange) dataset differing by type of model used for entropy calculation.

**Hyperparameter: Model Size**    Next, we examined the effect of model size on the entropy calculation. As shown in Figure 5, the distribution of utility scores changes when using LLaMA-1B compared to LLaMA-8B. While the impact is less pronounced than the effect of model type, we observe an increase of 0.11 in the Jensen-Shannon divergence score when using the 8B model. We hypothesize that larger models are better able to capture subtle and complex patterns in their response distributions. Given that the improvement is relatively modest, we opted to use the 8B model to balance performance with computational efficiency and avoid the added cost of larger models.

**Hyperparameter: Decoding Type**    We next examine the effect of decoding type on the sequential average token entropy, comparing greedy and sampling decoding. As shown in Figure 6, sampling decoding produces more diverse distributions than greedy decoding. This is expected, as sampling introduces stochasticity into token selection, allowing the model to explore multiple plausible continuations, whereas greedy decoding always selects the highest-probability token at each step, leading to more concentrated and less variable distributions. Based on this observation, we use sampling decoding in our experiments.

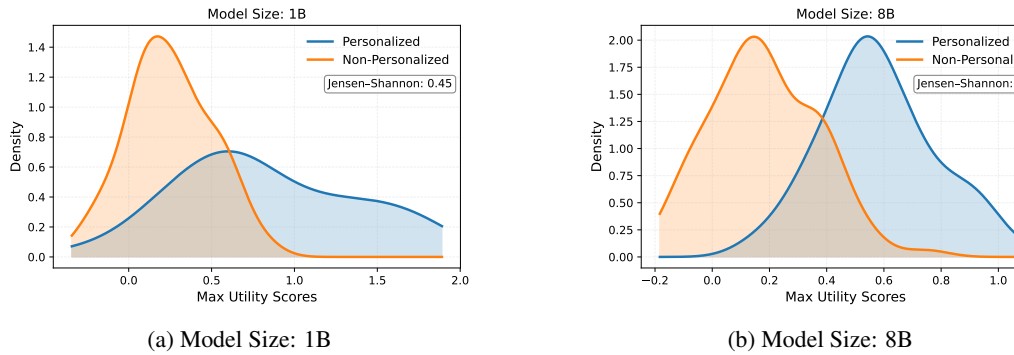

(a) Model Size: 1B                      (b) Model Size: 8B

Figure 5: Distribution of maximum utility scores for a personalized (blue) and non-personalized (orange) dataset differing by size of model used for entropy calculation.

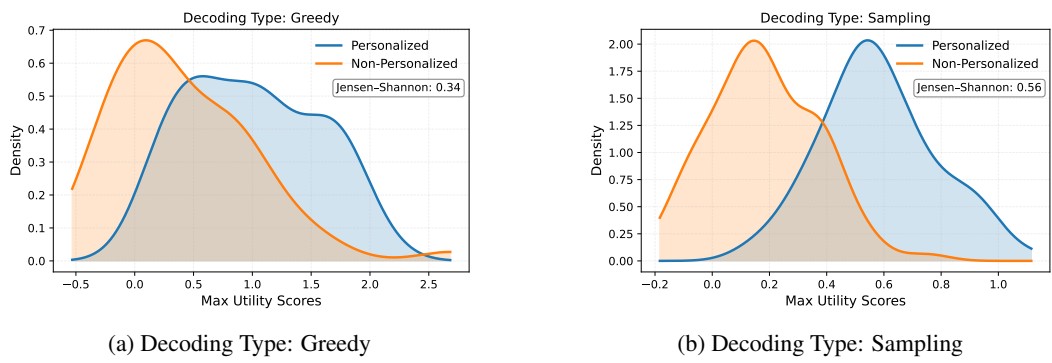

(a) Decoding Type: Greedy                (b) Decoding Type: Sampling

Figure 6: Distribution of maximum utility scores for a personalized (blue) and non-personalized (orange) dataset differing by decoding type.

**Hyperparameter: Number of Tokens** As noted in the main text, we limit the number of tokens used in the calculation of Equation (5) to avoid bias introduced by longer generations, since longer outputs tend to have lower entropy. To study this, we randomly sampled $n = 10,000$ responses from the OpenAssistant dataset (Köpf et al., 2023), using only the first human input in each conversation that was labeled as English and non-toxic. We used this dataset since it is a real world dataset that could provide a wide diversity in types of user-LLM interactions. We then computed the average token-level entropy by token index, capped at 100 tokens. As shown in Figure 7, entropy begins to decrease significantly around $T = 20$ tokens, indicating that longer generations introduce a bias toward lower entropy.

Further analysis in Figure 8 compares the distribution of maximum utility scores when using 20 versus 100 maximum tokens for entropy calculations. Using 100 tokens reduces the Jensen-Shannon divergence by 0.18 compared to using only 20 tokens. Based on these results, we limit the calculation to the first 20 tokens in our experiments.

**Hyperparameter: Number of Samples** Next, we examine the effect of the number of Monte Carlo samples used in Equation (5) on the robustness of the utility score. As shown in Figure 9, increasing the number of samples improves the separation between personalized and non-personalized inputs, which aligns with expectations since the estimate should converge toward the true parameter as the sample size increases if the samples are i.i.d. However, this comes with a tradeoff: more samples require additional computation, longer runtime, and higher resource costs. Balancing these considerations, we chose $n = 5$ samples for our experiments.

**Hyperparameter: Number of User Profiles** Lastly, we examine how the robustness of the average maximum utility score changes as we increase the number of (query,user profile) pairs included in the average. Consistent with estimation theory, we find that the difference between the person-

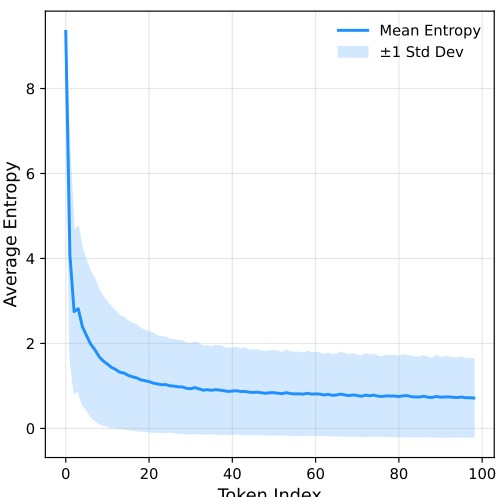

Figure 7: Average entropy by the index of the token being generated.

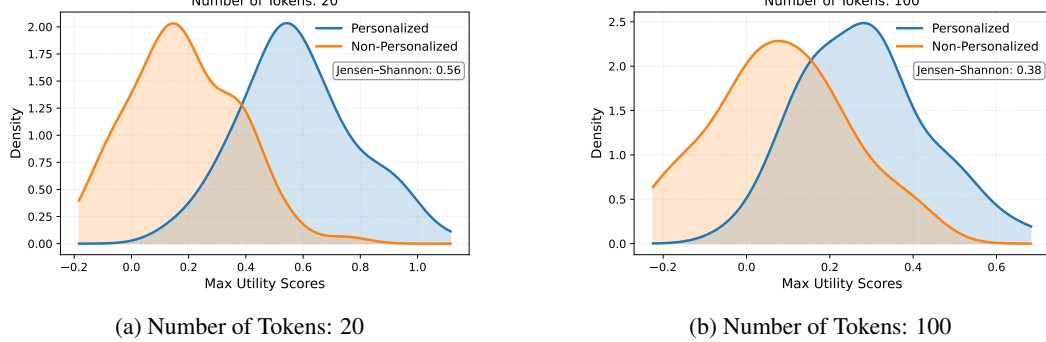

(a) Number of Tokens: 20           (b) Number of Tokens: 100

Figure 8: Distribution of maximum utility scores for a personalized (blue) and non-personalized (orange) dataset using different number of tokens.

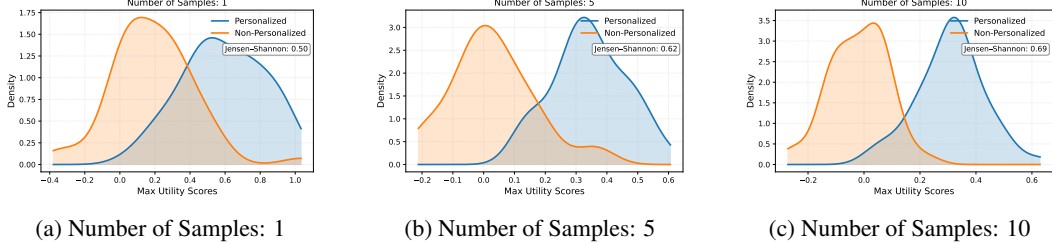

(a) Number of Samples: 1      (b) Number of Samples: 5      (c) Number of Samples: 10

Figure 9: Distribution of maximum utility scores for a personalized (blue) and non-personalized (orange) dataset differing number of Monte Carlo samples average over.

alized and non-personalized distributions grows as the sample size increases. Notably, there is a substantial jump of 0.22 in Jensen–Shannon divergence when increasing from 1 to 5 users, but only a marginal increase of 0.01 when going from 5 to 10 users. Based on this, we choose $u = 5$ users for constructing our training data to balance performance with cost of inference.

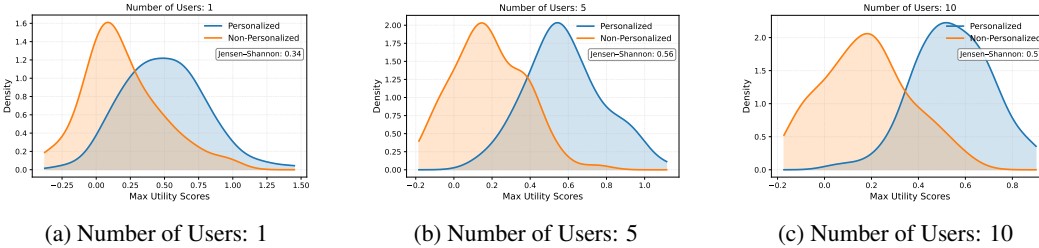

(a) Number of Users: 1    (b) Number of Users: 5    (c) Number of Users: 10

Figure 10: Distribution of maximum utility scores for a personalized (blue) and non-personalized (orange) dataset differing by number of users averaged over.

### B.2  RUMS-UTILITY THRESHOLD ANALYSIS

As mentioned in the main text, RUMS-Utility has a threshold hyperparameter that controls the sensitivity of memory item selection. Adjusting this threshold results in either more or fewer memory items being chosen. Table 4 illustrates how the average number of selected memory items changes as the threshold is varied. As the threshold increases, a stronger reduction in entropy is required for a memory item to be selected, leading to smaller average subset sizes. This analysis highlights that the threshold provides a simple yet effective way to balance the tradeoff between including more context and avoiding unnecessary or low-utility memory items.

| Threshold | PersonaFeedback | FreebaseQA | WildChat-Personal | WildChat-NonPersonal |
|---|---|---|---|---|
| -100 | 2.76 | 1.83 | 2.77 | 2.70 |
| 0.00 | 2.76 | 1.56 | 2.73 | 2.57 |
| 0.29 | 2.43 | 0.21 | 1.31 | 1.38 |
| 0.50 | 1.11 | 0.00 | 0.29 | 0.27 |
| 0.80 | 0.14 | 0.00 | 0.00 | 0..00 |

Table 4: Average number of memory items selected at different thresholds used in RUMS-Utility.

### B.3  EXAMPLES OF SELECTED MEMORY ITEMS

Table 5 illustrates one personalized and one non-personalized case. Consistent with Table 2, GPT4-ZeroShot and Semantic Similarity over-select, adding noise in non-personalized cases; GPT4-FewShot selects less in general but still selects items for non-personalized inputs. RUMS-Binary balances personalization by matching human annotations in the personalized case and abstaining in the non-personalized case.

### B.4  COST ANALYSIS

To better understand the practical implications of our approach, we analyze the tradeoff between computational cost and model performance across all methods. Since our methods and baselines differ in the number of memory items they select, and hence the number of additional tokens they introduce into the prompt, it is important to quantify how these differences affect overall efficiency at scale.

We measure computation overhead with two complementary metrics. First, we calculate the *average number of memory items selected* per dataset and method. This captures how the amount of items each method chooses in order to fulfill personalization or how little for no personalization. Second, we compute the *average number of additional tokens* that must be appended to the prompt during inference in order to include the selected memory items. This provides a direct measure of the inference overhead. Table 6 and Table 7 summarize these results, showing clear differences across methods: baselines such as All or Random predictably add large numbers of items, while information-retrieval heuristics like BM25 are highly selective. Our methods lie in between, striking a balance between selecting informative items and avoiding unnecessary additions.

Table 5: Examples of memory items selected by each method compared to human gold labels. Overlapping memory with items human annotation is **bolded**.

| User Input | Humans* | RUMS-Binary | GPT4-ZeroShot | GPT4-FewShot | Semantic Similarity |
|---|---|---|---|---|---|
| **[PERSONALIZED]**
I became a millionaire recently and want to give money to some local businesses | Location
Hobbies and Interests
Volunteer Activities | **Location**
**Hobbies and interests**
**Volunteer activities**
Personal values
Industry of employment | **Location**
Occupation
Industry of employment
Current projects
Long-term aspirations | **Location**
Recent life events | **Location**
**Volunteer activities**
Financial situation
Annual Income
Pet ownership |
| **[NON-PERSONALIZED]**
Who was the Prime Minister of Gt. Britain at the outbreak of The First World War? | No features selected | **No features selected** | Location
Occupation
Name
Political affiliation | Location
Occupation | Travel history
Occupation
Name
Political affiliation
Current projects |

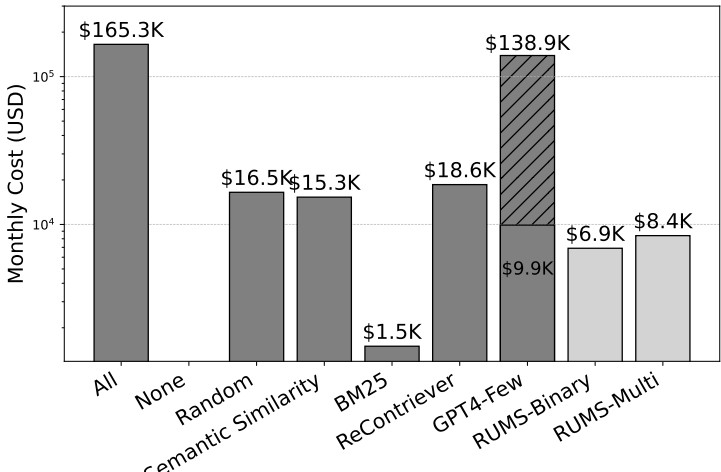

Figure 11: Estimated monthly cost of additional prompt tokens (1M queries per day, 30 days, GPT-4 input pricing at $0.01 per 1K tokens).

To translate these measures into practical cost, we estimate the monthly cost of running one million queries per day for each method using pricing aligned with GPT-4. Specifically, we convert the additional tokens from Table 2 into dollar cost under the assumption that each token incurs a fixed marginal cost of $0.01 per 1K tokens and 30 days per month. This allows us to approximate the total monthly inference cost for each method, and thus assess the economic tradeoff of personalization strategies at scale.

The results are visualized in the Figure 11. As expected, the All baseline is by far the most expensive, exceeding $160K per month due to its maximal inclusion of all memory items. On the other hand, our model RUMS-Binary yields much lower additional cost of only $6.9K which is about 30% lower than just the inference cost of including the memory features selected by GPT4-FewShot, not including the cost to generate these memory items. Our other model, RUMS-Multi, also has low cost of only $8.4K, keeping costs well below stronger baselines like GPT-4, ReContriever, and Semantic Similarity. Overall, this analysis highlights that RUMS-Models can be deployed at scale in a cost-efficient manner, with our methods demonstrating favorable tradeoffs between informativeness and economic feasibility.

### B.5 Full Results of H3

In this section we expand H3 by breaking down results for the Synthetic and Real World datasets by inputs intended to be personalized (left) and not personalized (right). This comparison highlights differences between RUMS-Binary and RUMS-Multi. With the less capable response model, LLaMA-70B, RUMS-Binary (which favors fewer memory items) outperforms most baselines, showing es-

Table 6: Average number of memory items selected by method per dataset.

| Method | FreebaseQA | PersonaFeedback | WildChat-Personal | WildChat-NonPersonal |
|---|---|---|---|---|
| All | 50 | 50 | 50 | 50 |
| None | 0 | 0 | 0 | 0 |
| Random | 5 | 5 | 5 | 5 |
| Semantic Similarity | 4.71 | 4.54 | 4.84 | 4.74 |
| BM25 | 0.39 | 0.81 | 0.28 | 0.42 |
| ReContriever | 5 | 5 | 5 | 5 |
| GPT4-Few | 1.21 | 4.39 | 3.90 | 2.10 |
| RUMS-Binary | 0.02 | 3.14 | 2.80 | 1.64 |
| RUMS-Multi | 1.45 | 3.58 | 2.36 | 1.76 |

Table 7: Average number of extra tokens required for response generation.

| Method | FreebaseQA | PersonaFeedback | WildChat-Personal | WildChat-NonPersonal |
|---|---|---|---|---|
| All | 551.59 | 551.28 | 552.16 | 550.40 |
| None | 0 | 0 | 0 | 0 |
| Random | 54.87 | 54.98 | 54.34 | 54.20 |
| Semantic Similarity | 48.04 | 51.44 | 53.04 | 49.58 |
| BM25 | 1.48 | 8.95 | 3.20 | 4.86 |
| ReContriever | 60.87 | 62.93 | 61.22 | 61.72 |
| GPT4-Few | 12.96 | 49.31 | 46.02 | 23.58 |
| RUMS-Binary | 0.23 | 37.39 | 34.88 | 19.10 |
| RUMS-Multi | 13.88 | 45.34 | 30.90 | 21.08 |

pecially strong gains on non-personalized inputs, with a $38\%$ average win rate over baselines on the Synthetic dataset. Similar patterns hold on the Real World dataset, except for the None and BM25 baselines, which never select items.

By contrast, RUMS-Multi (which selects more memory items) performs better with the stronger GPT-4 model, particularly on personalized inputs, achieving $25\%$ and $26\%$ average win rates on the personalized Synthetic and Real World datasets respectively. These results demonstrate how RUMS can be tuned to different settings, adapting to the response model and personalization needs.

Table 8: Average winrate comparison on Synthetic data under Not Personal (left) and Personal (right) conditions. Positive values (green) indicate better performance, negative values (red) indicate worse performance. Values greater than 0.1 in magnitude are bolded.

| (a) Non-personalized inputs | | | | | (b) Personalized inputs | | | | |
|---|---|---|---|---|---|---|---|---|---|
| | LLaMA 70B | | GPT-4 | | | LLaMA 70B | | GPT-4 | |
| | RUMS-Binary. | RUMS-Multi | RUMS-Binary. | RUMS-Multi | | RUMS-Binary. | RUMS-Multi | RUMS-Binary. | RUMS-Multi |
| None | -0.03 | **-0.79** | 0.02 | 0.00 | none | **0.52** | **0.53** | **0.50** | **0.72** |
| All | **0.45** | **-0.24** | 0.00 | 0.02 | All | **-0.14** | **-0.14** | **-0.31** | 0.01 |
| Random | **0.50** | **-0.12** | -0.03 | -0.01 | Random | **0.18** | **0.37** | **0.15** | **0.39** |
| Semantic Similarity | **0.60** | 0.01 | 0.02 | 0.00 | Semantic Similarity | -0.06 | **0.17** | -0.07 | 0.09 |
| BM25 | **0.12** | **-0.66** | -0.02 | 0.01 | BM25 | **0.25** | **0.40** | **0.42** | **0.40** |
| ReContriever | **0.62** | **-0.14** | 0.01 | 0.03 | ReContriever | -0.09 | 0.09 | -0.09 | 0.09 |
| GPT4-Few | **0.42** | **-0.34** | 0.00 | 0.01 | GPT4-Few | -0.07 | 0.03 | **-0.17** | 0.03 |

## C  ADDITIONAL RELATED WORKS

**Foundations and Variations of BOED**  One of the earliest Bayesian methods for experimental design was proposed by Lindley (1956), which introduced a general framework based on maximizing the expected Shannon entropy. Bernardo (1979) later extended this work to incorporate decision spaces. Since then, BOED has been explored in numerous settings; for a complete review, see Chaloner & Verdinelli (1995). However, in practice, BOED can be computationally demanding because it requires estimating the expected information gain of each candidate experiment. To address

Table 9: Average winrate comparison on Real World data under Not Personal (left) and Personal (right) conditions. Positive values (green) indicate better performance, negative values (red) indicate worse performance. Values greater than 0.1 in magnitude are bolded.

| | (a) Non-personalized inputs | | | | | (b) Personalized inputs | | | |
| | LLaMA 70B | | GPT-4 | | | LLaMA 70B | | GPT-4 | |
| | RUMS-Binary | RUMS-Multi | RUMS-Binary | RUMS-Multi | | RUMS-Binary | RUMS-Multi | RUMS-Binary | RUMS-Multi |
|---|---|---|---|---|---|---|---|---|---|
| None | **-0.36** | **-0.64** | **-0.12** | **-0.26** | None | **0.46** | **0.70** | **0.52** | **0.60** |
| All | 0.08 | **-0.14** | **-0.14** | **-0.16** | All | **-0.16** | -0.04 | **-0.14** | 0.04 |
| Random | -0.02 | -0.06 | -0.06 | **-0.26** | Random | **0.24** | **0.42** | **0.26** | **0.34** |
| Semantic Similarity | 0.04 | -0.02 | -0.02 | **-0.12** | Semantic Similarity | -0.10 | 0.00 | -0.02 | **0.12** |
| BM25 | **-0.40** | **-0.16** | **-0.16** | **-0.26** | BM25 | **0.42** | **0.58** | **0.52** | **0.52** |
| ReContriever | **0.12** | 0.04 | 0.04 | 0.02 | ReContriever | 0.00 | **0.24** | **-0.14** | **0.24** |
| GPT4-Few | 0.08 | **0.12** | **0.12** | -0.02 | GPT4-Few | -0.08 | -0.04 | -0.04 | -0.04 |

this challenge, Foster et al. (2019) introduced a variational BOED approach that leverages amortized inference to efficiently approximate the expected information gain for experiment selection. Similarly, Schütz et al. (2025) formalized adaptive submodularity, showing that greedy information-maximization strategies, such as selecting the feature with the highest expected entropy reduction at each step, are near-optimal in many cases. These works provide the foundational justification for our estimations used in RUMS.

## D  FORMULAS

### D.1  SHANNON ENTROPY

We define the Shannon Entropy as a measure of uncertainty within a system. For a random variable $x \in X$ and a distribution according to $p : X \to [0,1]$ such that $p(x) := \mathbb{P}[X = x]$. Then we formalize the Shannon entropy $H$ as,

$$H(x) = -\sum_{x \in X} p(x) \log p(x)$$

## E  SEQUENCE ENTROPY EQUIVALENCE PROOF

*Proof.* Let $p_\theta(y \mid x)$ denote the model distribution over token sequences $y = (y_1, \ldots, y_T)$ given input $x$. At each step $t$, the conditional next-token distribution is

$$p_\theta(\cdot \mid y_{<t}, x),$$

where $y_{<t}$ is the prefix of tokens before step $t$.

By the chain rule of probability,

$$-\log p_\theta(y \mid x) = \sum_{t=1}^{T} -\log p_\theta(y_t \mid y_{<t}, x).$$

Taking the expectation with respect to $y \sim p_\theta(\cdot \mid x)$ on both sides gives

$$\mathbb{E}_{y \sim p_\theta}[-\log p_\theta(y \mid x)] = \mathbb{E}_{y \sim p_\theta}\left[\sum_{t=1}^{T} -\log p_\theta(y_t \mid y_{<t}, x)\right]$$

$$= \sum_{t=1}^{T} \mathbb{E}_{y \sim p_\theta}[-\log p_\theta(y_t \mid y_{<t}, x)] \qquad \text{linearity of expectation}$$

$$= \sum_{t=1}^{T} \mathbb{E}_{y_{<t} \sim p_\theta}\left[\sum_{y_t} p_\theta(y_t \mid y_{<t}, x) - \log p_\theta(y_t \mid y_{<t}, x)\right] \qquad \text{law of iterated expectation}$$

$$= \mathbb{E}_{y_{<t} \sim p_\theta}\left[\sum_{t=1}^{T} H(p_\theta(\cdot \mid y_{<t}, x))\right].$$

$\square$

# F   RUMS-Utility Algorithm

In Algorithm 1 we provide the formalized algorithm for RUMS-Utility.

---

**Algorithm 1** Memory Selection via RUMS

---

**Require:** User input $x$, user memory $u \in \mathbb{R}^d$
**Ensure:** Selected memory subset $m_{u,S}^*$ that maximizes utility
 1: **Candidate Memory Item Identification:**
 2: Use LLM to generate candidate subsets $S$ for $x$
 3: **Subset Evaluation:**
 4: **for** each $s \in S$ **do**
 5:     Map candidate attritubes to user values $m_{u,S}$
 6:     Prompt Reference LM with:
 7:         (i) $x$ only
 8:         (ii) $x$ and $m_{u,S}$
 9:     Compute estimated utility $\hat{\text{EIG}}(S)$ using Equation (5)
10:     Store utility score for subset $s$
11: **Subset Selection:**
12: $s^* \leftarrow \arg\max_{s \in S} \hat{\text{EIG}}(S)$
13: **if** all MI scores $<$ threshold **then**
14:     $s^* \leftarrow \emptyset$                                    ▷ Select no features

---

# G   Experimental Details

This section provides the full details of the experiments described in Section 4.

## G.1   Experimental Setup

Across all experiments (H1–H3), we consider a fixed set of $n = 50$ memory items (e.g., *location, age, dietary restrictions*). The task is to select a subset of up to $n = 5$ memory items that can improve personalization.

## G.2   Datasets

To evaluate scenarios where personalization is beneficial as well as those where it is not, we use a diverse mixture of user inputs drawn from prior work, including both synthetic and real-world sources. The three datasets we used are described below:

**PersonaFeedback (Tao et al., 2025)**   This is a synthetically created dataset consisting of user-specific queries designed to reflect inputs from realistic and diverse user personas. First, they constructed personas by representing them as a structured dictionary of memory items, including demographics, personality traits, and preferences. Then, open-source content from social media, reviews, and forums were used to populate these items with specific values. Lastly, an LLM generator was used to produce queries based on a sampled persona and specific sub-scenarios. Additional steps such as similarity-based filtering, rephrasing to avoid directly revealing persona information, and manual annotations ensured diversity and realism. The final dataset contains more than 4,000 persona-specific questions. The original data is in Chinese and therefore, we translated to English using a machine translation model (Tiedemann & Thottingal, 2020). Example: *"Do you have any music for the trip?"*

**FreebaseQA (Jiang et al., 2019)**   A dataset of trivia-style questions and answers collected from human trivia competitions. The original dataset contains approximately 28,000 question–answer pairs. Example: *"What type of snake is a sidewinder?"*

**WildChat (Zhao et al., 2024)**  A large-scale collection of more than 1 million real-world user–ChatGPT interactions, encompassing over 2.5 million dialogue turns. The dataset was compiled by offering free access to ChatGPT and GPT-4 in exchange for user consent to share chat histories. It covers diverse conversational scenarios such as ambiguous requests, code-switching, topic shifts, and political discussions. We filter the data to retain only English conversations labeled as non-toxic, and we use only the first human-written input per conversation to ensure that queries contain sufficient context.

To determine whether an input is personalized, two human annotators labeled a random selection of WildChat queries as either beneficial for personalization (WildChat-Personal) or not (WildChat-NonPersonal). Only queries with full annotator agreement were retained. Examples: *"I'm hungry"; "Who is Mohamed Gamal Mohamed Maniieh?"*

**Synthetic vs. Real-World Datasets**  By design, PersonaFeedback is treated as a personalized input set, and FreebaseQA as non-personalized. We collectively refer to these as the "Synthetic" dataset, since they were not originally created for LLM input. WildChat is considered a "Real-World" dataset, as it was collected directly from real user interactions with an LLM. We use both synthetic and real-world datasets to evaluate the robustness of RUMS across many diverse inputs.

**Sample Size by Hypothesis**  The number of samples varies across hypotheses. For H1, we use $n = 500$ samples from PersonaFeedback and FreebaseQA to analyze RUMS-Utility for binary personalization decisions. For H1', we use $n = 25$ inputs each from PersonaFeedback, FreebaseQA, WildChat-Personal, and WildChat-NonPersonal to extend the H1 analysis to correlation with human judgment. For H2, we use $n = 15$ inputs each from PersonaFeedback, FreebaseQA, WildChat-Personal, and WildChat-NonPersonal to examine the correlation between RUMS memory selection and human preference. Finally, for H3, we use $n = 100$ samples from PersonaFeedback and FreebaseQA along with $n = 50$ samples each from WildChat-Personal and WildChat-NonPersonal to analyze the downstream impact of subsets chosen by RUMS-Model.

### G.3 PROFILE GENERATION

User profiles are incorporated during the training data curation phase, where specific user values are used to compute the entropy difference of the model required for the utility function in Equation (5). Also, we use profiles in our test sets for evaluation.

**Memory Item Creation**  To construct user profiles, we prompted GPT-4 (OpenAI et al., 2024) to generate 50 commonly occurring user memory items. These items span demographic, behavioral, and preference-based attributes, providing a diverse set of values for personalization. The complete list is as follows:

- Name
- Gender
- Location (City/State/Country)
- Ethnicity
- Race
- Annual Household Income
- Language spoken at home
- Language preference
- Sexual Preference
- Disabilities
- Occupation
- Political affiliation
- Education level
- Hobbies and interests
- Favorite sports
- Preferred music genre
- Favorite books
- Preferred movie genre
- Favorite TV shows
- Favorite foods
- Dietary restrictions (e.g., vegetarian, vegan)
- Relationship status
- Number of children
- Pet ownership (types of pets)
- Travel history (countries visited)
- Travel preferences (adventurous, relaxing)
- Social media platforms used
- Level of tech-savviness
- Preferred mode of communication (text, voice, video)
- Work schedule (full-time, part-time)
- Industry of employment
- Current projects or goals
- Long-term aspirations
- Financial situation (broad categories)
- Health and fitness goals
- Preferred exercise activities

- Level of environmental consciousness
- Volunteer activities or interests
- Current challenges or pain points
- Personality traits (e.g., introverted, extroverted)

- Favorite pastimes
- Personal values or beliefs
- Cultural background
- Religious beliefs
- Recent life events (e.g., change in job, moved)
- Technology usage (devices, apps)

- Frequency of using the service (daily, weekly)
- Reasons for using the service
- Feedback preferences (detailed, brief)
- Preferred tone of communication (formal, casual)

**Memory Item Values**   Next, we used GPT-4 (OpenAI et al., 2024) to generate consistent user profiles without contradictions (e.g., "vegan" with "favorite food = steak"), following the prompt in Prompt 1. To further ensure logical consistency, we employed GPT-4 to filter out any profiles that appeared illogical, as described in Prompt 2. This process was repeated until we obtained $n = 100$ unique, logically consistent profiles for both training and testing.

This method yields a diverse set of user profiles, as illustrated in the bar charts of memory item distributions shown in Figure 12 through Figure 61.

**Prompts used for Profile Generation**

**Prompt 1.** *For these 50 features, I want you to create 10 very diverse persona profiles. So for each of the 50 features, I want you to select a value (e.g. "age" = 50) that would make up the persona profile. Make sure the values make sense for each other, meaning don't have a person who is 12 years old have a job as a accountant.*

*Output a json file with a dictionary with 10 keys, "Persona_1", "Persona_2",...,"Persona_10", whose corresponding values is a list of the selected values in the same order as the features (e.g. [John, Male, Seattle/WA/USA,...]).*

*Do not include any other markdown or text. Here are the Features:*

**Prompt 2.** *You are an AI assistant that needs to decide if a user profile is logically sound. This means that the information in the user profile should be consistent and non-conflicting.*

*Only output a single word, either 'yes' or 'no', to indicate if the user profile is logically sound.*

*Example 1: If a user profile states that the user is allergic to peanuts, it should not also state their favorite food is peanuts.*

*Example 2: If a user profiles states highest education is a GED, it should not also state their profession is a doctor.*

*Example 3: If a user profile states that the user is 25 years old, it should not also state that they have been working for 30 years.*

*Example 4: If a user profile states that they live in Columbia, it should not state that their main language is French.*

*User Profile:*

### G.4    BASELINES

In this section we describe each baseline that is used in our experimentation.

**GPT-4 Prompting (OpenAI et al., 2024):**   We use GPT-4o as a baseline across H1, H2, and H3. In H1, we apply Prompt 3 for the zero-shot setting and Prompt 4 for the few-shot setting to decide whether personalization is needed for a given query. For the few-shot case, we provide four in-context examples: two requiring personalization and two not requiring personalization.

In H2 and H3, we adapt the prompts to instead elicit the subset of specific user memory items that would be beneficial for personalization. We use Prompt 5 for zero-shot and Prompt 6 for few-shot. In the few-shot setting, we provide two in-context examples: one requiring personalization and one not.

**Prompt 3.** *You are a classifier that determines whether a given user input would benefit from user-specific information for the model to respond.*

*Your response must be a JSON object with a two key "Decision" with a value of either "True" or "False" and "Features" with either a list of user-specific features (at most 10 features) if "Decision" is "True" or "None" if "Decision" is "False" :*

*Output "True" if the input would benefit from user-specific information to better respond to the query.*

*Output "False" if the input would not benefit from user-specific information or would be the same answer regardless of the user.*

*Respond with only the JSON object. Do not include any explanation or additional text.*

*Input: user_input*

**Prompt 4.** *'You are a classifier that determines whether a given user input would benefit from user-specific information for the model to respond.*

*Your response must be a JSON object with a two key "Decision" with a value of either "True" or "False" and "Features" with either a list of user-specific features (at most 10 features) if "Decision" is "True" or "None" if "Decision" is "False" :*

*Output "True" if the input would benefit from user-specific information to better respond to the query.*

*Output "False" if the input would not benefit from user-specific information or would be the same answer regardless of the user.*

*Respond with only the JSON object. Do not include any explanation or additional text.*

*Example 1:*

*Input: "Where should I eat tonight?"*

*Output: "Decision": "True", "Features": ["location", "dietary restrictions", "income"]*

*Example 2:*

*Input: "Can you help me write an essay on feminism?"*

*Output: "Decision": "True", "Features": ["education", "political ideology", "gender"]*

*Example 3:*

*Input: "What is the capital of Austin, Texas?"*

*Output: "Decision": "False", "Features": "None"*

*Example 4:*

*Input: "Can you explain what AGI is?"*

*Output: "Decision": "False", "Features": "None"*

*Now classify the following input:*

*Input: user_input'*

**Prompt 5.** *Given a user-input and list of features, select which features might be important for answering the user-input.*

*Choose up to 5 features. Respond in a JSON with one key "Chosen Features", which has a list of chosen features. Do not include any other text or formatting.*

*User Input: user_input*

*Possible Features: candidate_features_ls*

**Prompt 6.** *Given a user-input and list of features, select which features might be important for answering the user-input. Choose up to 5 features.*

*Respond in a JSON with one key "Chosen Features", which has a list of chosen features. Do not include any other text or formatting.*

*Example Input 1: "whats the weather today"*

*Example Output 1: "Chosen Features": ["Location (City/State/Country)"]*

*Example Input 2: "What is the capital of Texas?"*

*Example Output 2: "Chosen Features": []*

*User Input: user_input*

*Possible Features: candidate_features_ls'*

**Semantic Similarity:** In H1–H3, we also include a simple baseline based on semantic similarity. Here, we compute the cosine similarity between user query embeddings and each memory item embeddings using a lightweight sentence-transformer model (Wang et al., 2020).

In H1, we use the validation set, defined as a 20% hold-out of the data, to tune a threshold for deciding whether personalization is required. Thresholds are determined by grid search over $[-1, 1]$ in increments of 0.5, selecting the value that maximizes agreement with human annotations for each dataset. A query is labeled as requiring personalization if at least one memory item has a similarity score above the selected threshold. The exact thresholds used for each dataset are reported in Table 10.

| Dataset | Threshold |
|---|---|
| All | -1.0 |
| Synthetic | -1.0 |
| Real World | 0.5 |

Table 10: Threshold for H1 analysis of semantic similarity baseline. If at least one memory item had a semantic similarity higher than this threshold, then the query was labeled as needing personalization.

Since H2 and H3 require selecting specific memory items for personalization rather than making a simple binary decision, we set an intuitive threshold of $t = 0$, corresponding to no similarity between the user query and a memory item. If more than five memory items exceed this threshold, we select the top five with the strongest correlations.

**Simple Baselines:** We also compare against several naive baselines. The "None" baseline provides no features ($n = 0$), serving as a lower bound for personalized inputs and an upper bound for non-personalized ones. The "All" baseline includes all available features ($n = 50$), testing the effect of providing maximal but noisy context. Finally, the "Random" baseline samples $n = 5$ items at random, establishing a chance-level reference.

**SOTA Retrieval Methods:** We additionally evaluate two state-of-the-art retrieval methods.

First, BM25 (Lù, 2024), a widely used ranking function in information retrieval, can be adapted for memory selection. It scores memory items (treated as terms) based on their frequency in each instance relative to the entire dataset, balancing how common a memory item is with how discriminative it is. Items with higher BM25 scores are considered more informative and are selected as the most relevant, making BM25 well-suited for sparse or high-dimensional text and categorical data.

Second, ReContriever (Lei et al., 2023) is a retrieval-based method that scores memory items by their ability to help a model retrieve relevant information. It leverages dense embeddings to measure semantic similarity between queries and candidate items, effectively identifying the most informative items for downstream tasks. Memory items that yield higher retrieval relevance are prioritized, making ReContriever particularly effective for selecting contextually important or predictive inputs in text or structured datasets. Similarly to semantic similarity a threshold of $t = 0$ is used as threshold for selecting memory items, with a maximum of up to $n = 5$ selected.

## G.5 RUMS-UTILITY AND RUMS-MODEL DETAILS

In this paper, we describe two main methods: RUMS-Utility, which selects a subset of user memory items with the highest utility for personalization directly, and RUMS-Model, a lightweight trained model built on data generated by RUMS-Utility.

**RUMS-Utility:** We use RUMS-Utility not only for the analyses in H1 and H2 but also to generate training data for fine-tuning RUMS-Models. In general, we compute entropy in Equation (6) using Llama-3.1-8B (AI, 2024b) with a temperature of 0.7 and sampling-based decoding to obtain the response distribution. We restrict the maximum generation length to $T = 20$ tokens, chosen to mitigate bias from longer generations having naturally lower entropy.

Following the algorithm in Section 3.1.2, we first prompt GPT-4 to reduce the candidate memory items from 50 to at least 1 and at most 10 before computing RUMS-Utility (see Prompt 7). Utility scores are then estimated using $n = 5$ Monte Carlo samples. For each input, we generate $n = 5$ distinct user profiles as separate training points, which can yield entirely unique best memory subsets of data depending on the values in the user profile.

The chosen hyperparameters (e.g., model size, number of Monte Carlo samples, etc.) reflect a trade-off: obtaining reliable signals of memory item importance while keeping computations lightweight and feasible for large-scale inference. Further discussion of these choices is provided in Appendix B.1.

**Prompt 7.** *Given a user-input and list of features, select which features might be important for answering the user-input.*

*Choose at least 1 and up to 10 features.*

*Respond in a JSON with one key "Chosen Features", which has a list of chosen features. ' Do not include any other text or formatting.*

**Choosing the Personalization Threshold:** To determine the threshold for deciding whether to apply personalization or select no personalization, we used a held-out validation set of size $n = 100$ from each of PersonaFeedback and FreebaseQA. For each query, we computed the average maximum utility score over the $n = 5$ generated user profiles. We labeled inputs from PersonaFeedback as personalized (1) and inputs from FreebaseQA as non-personalized (0). These scores were combined and labeled by dataset to construct an ROC curve.

We then selected the threshold that maximized Youden's J statistic ($J = \text{TPR} - \text{FPR}$), which identifies the point that best balances sensitivity and specificity. Using this procedure, we determined an optimal threshold of $t = 0.29$, which was applied in all downstream analyses.

**RUMS-Models** For training the downstream models, we sampled approximately 2.5K unique PersonaFeedback queries and 2K unique FreebaseQA queries. Each query was paired with five user profiles, and RUMS-Utility was applied to select the best memory subsets, yielding 13K and 9K training examples, respectively. We excluded all FreebaseQA queries that exceeded the personalization threshold, thereby removing any noisy non-personalized samples. This design choice emphasized that non-personalized inputs should not include any memory items.

We then used these (query, memory subset) pairs to train two models. The first model, used in H1–H3, was a binary logistic regression implemented with DeBERTa-v3-large (He et al., 2021) and a final linear classification layer. To train this model, we converted each (query, memory subset) pair into multiple (query, memory item, label) triplets. Training was performed with a learning rate of $5 \times 10^{-6}$ for 12 epochs, and the best model was selected based on the highest F1 score on a held-out validation set ($n = 1K$).

The second model, used for H3, was a multiclass, multilabel logistic regression implemented again with DeBERTa-v2-large (He et al., 2021) and a final multiclass classification layer. Here, labels were constructed by merging the selected subsets from the $n = 5$ diverse user profiles per query. This produced a single label vector in which all memory items chosen in any of the five samples were marked as significant. This resulted in more dense labels, training the model to select more memory items. Training was performed with a learning rate of $2 \times 10^{-5}$ for 12 epochs, and the best model was again chosen based on the highest F1 score on a held-out validation set ($n = 1K$).

| Method | Model | # Parameters |
|---|---|---|
| RUMS | microsoft/deberta-v3-large | 418M |
| Semantic Similarity | all-MiniLM-L6-v2 | $\sim$22.7M |
| GPT4o | GPT4o | Not disclosed (est. $>$200B) |
| ReContriever | Yibin-Lei/ReContriever | $\sim$110M |

Table 11: Baseline methods and their corresponding model sizes.

**Selecting Regression Threshold**   Since RUMS-Models is trained as a logistic regression model (either binary or multiclass), its output represents the probability of selecting a particular memory item. A common approach is to use a cutoff of $0.5$ to make a binary decision, however, since our task is more specialized, instead we tuned this cutoff using a held-out evaluation set ($n = 1K$). Specifically, we perform a grid search over boundaries in $[0, 1]$ and select the cutoff that achieved the highest F1 score on the evaluation set. This yielded $b = 0.81$ for RUMS-Binary and $b = 0.93$ for RUMS-Multi.

G.6   COMPARING SIZE OF METHODS

In Table 11, we summarize the size of each model used in both our method and the baselines. Our approach uses a model comparable in scale to SOTA baselines, yet it is over $400\times$ smaller than GPT-4.

G.7   EVALUATION METRIC AND ANALYSIS

**H1 Evaluation:**   In H1, we compare the distribution of maximum utility scores between assumed personalized and non-personalized inputs. Specifically, we evaluate utility scores on a non-personalized dataset (FreebaseQA) and a personalized dataset (PersonaFeedback). We sampled $n = 500$ inputs from each dataset and computed the maximum utility score using Equation (5) for every input. The resulting distributions were then analyzed using divergence metrics and hypothesis testing.

For divergence metrics, we use Jensen–Shannon (JS) divergence, a symmetric measure of similarity between two probability distributions $P$ and $Q$. It is defined as:

$$\text{JS}(P\|Q) = \frac{1}{2}\,\text{KL}(P\|M) + \frac{1}{2}\,\text{KL}(Q\|M), \quad \text{where } M = \frac{P+Q}{2}$$

Here, $\text{KL}(P\|M)$ is the Kullback–Leibler divergence from $P$ to the average distribution $M$. JS divergence is always finite, symmetric, and bounded between 0 and $\log 2$.

For hypothesis testing, we use the two-sample Kolmogorov–Smirnov test to assess whether the two samples come from the same distribution. The null hypothesis states that they do, and we test significance at the $\alpha = 0.05$ level.

**H1' Evaluation:**   In H1', performance was evaluated using human labels as the gold standard. For each sample, four human annotators rated whether $n = 100$ samples would benefit from personalization using a 0–2 scale. If the annotators' confidence in personalization was 1 or higher, we considered the input to be personalized. Instructions provided to annotators are given in Prompt 8. We only included inputs with unanimous agreement ($n = 64$) in the evaluation. The average inter-rater agreement between pairs of annotators was $\kappa = 0.59$ using the Cohen's kappa. The primary metrics used were recall, measuring accuracy in identifying personalized inputs, and specificity, measuring accuracy in identifying non-personalized inputs.

**Prompt 8.** *Instructions: Read the user input and then decide if the input would benefit from specific user information.*

*0 = No Personalization Needed, 1 = Personalize is might be helpful, 2= Personalization is definitely helpful*

**H2 Evaluation:** For H2, we evaluated how closely each method's selected subset of memory items matched the subsets chosen by humans. To create the human-selected subsets, four annotators were asked to select all memory items (from the full set of 50) that they felt would be helpful in generating a response. Instructions provided to annotators are given in Prompt 9. For each input, we included only the memory items selected by at least three annotators to form the majority subset. This process resulted in an average of $4.19$ memory items per query for inputs where at least one item was chosen. This average guided our experimental design for H2 and H3, as we restricted each method to select no more than $n = 5$ items.

We evaluated performance using precision, recall, and F1 score. Here, precision measures the fraction of items selected by a method that were also in the human majority subset, recall measures the fraction of items in the human majority subset that were correctly selected by the method, and F1 is the harmonic mean of precision and recall, providing a single metric that balances both.

|  | Personalized | Not-Personalized | Difference |
|---|---|---|---|
| Human Annotation | 6.45 | 1.43 | 5.02 |
| RUMS-Utility | 1.87 | 0.80 | 1.07 |
| RUMS-Binary | 4.87 | **1.56** | **3.31** |
| RUMS-Multi | 3.27 | 1.67 | 1.60 |
| Semantic Similarity | **5.00** | 5.00 | 0.00 |
| GPT4-ZeroShot | 3.30 | 2.61 | 0.69 |
| GPT4-FewShot | 1.80 | 1.57 | 0.23 |

Table 12: Average number of features selected for personalized vs. not-personalized inputs. Methods closes to human performance are **bolded**.

**Number of Memory Items Selected in H2** Lastly, we include table 12 which compares the average number of memory items selected by each method in H2. Of note, we see that human annotations show a clear separation: personalized inputs have on average $6.5$ features selected, compared to only $1.4$ for non-personalized inputs, a difference of $5$. Among automatic methods, RUMS-Model and RUMS-Utility comes closest to this behavior, achieving a range of $3.3$ and $1.08$, whereas the next closest method is GPT4-ZeroShot which achieves only a $0.67$ difference. This further highlights how RUMS excels at both distinguishing when memory items are beneficial as well as which features are beneficial.

**Prompt 9.** *For each input, decide which (if any) user-features would be helpful to better respond to the input.*

**H3 Evaluation:** In H3, we evaluate the downstream impact of the memory subsets chosen by each method on generated responses. For each user input, responses are generated using the query combined with the subset of memory items selected by each method. This evaluation is performed with two models: GPT-4 (OpenAI et al., 2024), a high-performing model with robust capabilities, and LLaMA 70B Instruct (AI, 2024a), a smaller model more susceptible to noise. This setup demonstrates RUMS-Models's flexibility in adapting to models with varying capacities and usage scenarios.

Performance is measured using *generation win rate*, defined as the proportion of pairwise comparisons in which RUMS-Models's output is judged preferable to that of a baseline. GPT-4 is used as the judge, comparing responses generated with RUMS-Models's selected subset against those generated with each baseline's subset. LLM-as-Judge is validated to align closely with third-party human annotations for personalization when combined with in-context examples, model confidence scores (Dong et al., 2024).

To ensure robust evaluation, our methodology includes:

- *In-context examples*: Three examples are provided in the prompt for each evaluation to demonstrate the expected style of reasoning and scoring.
- *Confidence thresholds*: Only responses for which the model reports a self-generated confidence score above the 70% percentile are included in the final analysis.

- *Chain-of-thought prompting*: Evaluation prompts encourage the model to reason step-by-step and justify its preference for one response over another, improving consistency and interpretability of the win-rate comparisons.

For personalized datasets (PersonaFeedback and WildChat-Personal), the evaluation metrics in the chain-of-thought include:

- *Completeness*: Does the response fulfill the user's request?
- *Personalization*: Does the response appropriately incorporate relevant details from the user profile?
- *Overpersonalization*: Does the response include irrelevant, distracting, or unnecessary user details?

For non-personalized datasets (FreebaseQA and WildChat-NonPersonal), the metrics are:

- *Completeness/Accuracy*: Does the response correctly and fully answer the user's request?
- *Neutrality*: Is the response unbiased and general, rather than tailored to the user?

Full evaluation prompts can be found in Prompt 11 for personalized datasets, Prompt 12 for non-personalized datasets, and Prompt 10 for the three in-context examples used in all evaluation.

**Prompt 10.** *Example 1:*

*"user profile": ['Preferred exercise activities: Weightlifting', "Hobbies and interests: ['Surfing', 'Photography']", 'Location (City/State/Country): Melbourne/Victoria/Australia', "Social media platforms used: ['Facebook', 'LinkedIn']", "Technology usage (devices, apps): ['Smartphone', 'Laptop']],*

*"user input": " Why are there five rings in Olympic logo? And what does they symbolize?",*

*"response_1": " The five rings in the Olympic logo represent the five continents: Africa, the Americas, Asia, Europe, and Oceania. They are interlinked to symbolize the unity and friendship among nations, highlighting the idea of bringing together athletes from around the world to compete in the spirit of sportsmanship. The colors of the rings (blue, yellow, black, green, and red) were chosen because at least one of these colors appears in the national flags of all participating countries.",*

*"response_2": "The five rings in the Olympic logo represent the five continents of the world, Africa, the Americas, Asia, Europe, and Oceania. This design, created by Pierre de Coubertin in 1913, symbolizes the unity and connection among the nations and cultures participating in the Olympic Games. The rings are interlocked, indicating that these continents are united through the spirit of sport, competition, and the Olympic ideals of excellence, friendship, and respect. The colors of the rings (blue, yellow, black, green, and red) along with the white background were chosen because at least one of these colors appears on the flag of every nation competing in the Olympics.",*

*"overall": "0",*

*"overall_explanation": "Although they are worded a bit differently, they generally have the same information and do not include unhelpful user-information in the answer."*

*Example 2:*

*"user profile": ["Social media platforms used: ['Instagram', 'Facebook']", 'Dietary restrictions (e.g., vegetarian, vegan): None', 'Gender: Female', "Pet ownership (types of pets): ['Cat']", 'Current projects or goals: Continue education', 'Religious beliefs: Christian', 'Favorite TV shows: ["Greyś Anatomy"]', 'Reasons for using the service: Medical updates', 'Current challenges or pain points: Shift work', 'Travel preferences (adventurous, relaxing): Relaxing'],*

*"user input": "The 2011 album 'The King of Limbs' was released by which English rock band?",*

*"response_1": "The 2011 album 'The King of Limbs' was released by the English rock band Radiohead.",*

*"response_2": "The album 'The King of Limbs' was released by the English rock band Radiohead",*

*"overall": "0",*

*"overall_explanation": "These answer are the same, minus only minor differences"*

*Example 3:*

*"user profile": ['Current challenges or pain points: Commuting stress', 'Location (City/State/Country): Bangalore/Karnataka/India', 'Favorite sports: Cricket', "Personal values or beliefs: ['Kindness', 'Resilience']", 'Political affiliation: Centrist', 'Name: Arjun Mehta', 'Reasons for using the service: Access tools', 'Financial situation (broad categories): Financially modest', 'Level of tech-savviness: Moderate', 'Cultural background: South Asian', 'Frequency of using the service (daily, weekly): Weekly', "Favorite TV shows: ['Sacred Games']", 'Travel preferences (adventurous, relaxing): Relaxing', 'Preferred mode of communication (text, voice, video): Voice', "Favorite foods: ['Dosa', 'Chaat']", 'Health and fitness goals: Improve health'],*

*"user input": " Are there any recommended effective ways to enhance the fun of English grammar learning?",*

*"response_1": "Hi Arjun! Here are some recommended ways to enhance the fun of learning English grammar: 1. **Games and Quizzes**: Engage in grammar games or online quizzes that challenge your understanding in a fun way. Websites like Kahoot! or Quizlet offer interactive tools. 2. **Group Learning**: Form a study group with friends or peers. This can make the learning process more social and less intimidating. You can discuss grammar topics over chai or dosa! 3. **Music and Lyrics**: Listen to English songs and try to analyze the lyrics. Identifying verb tenses or other grammatical structures within your favorite songs can be both educational and entertaining. 4. **Humor in Learning**: Explore humorous content, such as grammar memes or funny videos about grammar mistakes, to lighten the mood. 5. **Role Play**: Practice dialogues using correct grammar in fun role-playing scenarios that interest you, perhaps even related to cricket! By combining these methods with your existing interests and values, like resilience and kindness, you can make grammar learning a much more enjoyable experience. Happy learning!",*

*"response_2": "Improving the enjoyment of English grammar learning can be approached in various fun and engaging ways. Here are some recommendations that might resonate with you: 1. **Gamify Learning**: Use educational games and apps that focus on grammar. Many platforms have quizzes, challenges, and rewards that make learning feel like a fun competition. 2. **Interactive Activities**: Engage in activities like grammar charades or sentence-building races with friends or family. This adds a social element to learning. 3. **Threshold Learning**: Set small, achievable goals and reward yourself when you reach them. This creates a sense of accomplishment and makes the process more enjoyable. 4. **Join a Language Group**: Participate in local or online language exchange groups where you can practice speaking with others. This can be a relaxing way to enhance your skills without the pressure of formal study. Would you like any specific resources or activities tailored to your preferences?",*

*"overall": "2",*

*"overall_explanation": "Response 1 is over personalized to the profile and inserts user-information in an unnatural and unhelpful way"*

**Prompt 11.** *You are an impartial and knowledgeable evaluator. You will be shown a user profile, a user input, and two model-generated responses. Your task is to decide if **only one of the responses** meets the evaluation criteria. If both responses do (or neither does), mark it as a tie (0).*

*Evaluation Criteria:*

*- **Completeness**: Does the response fulfill the user's request?*

*- **Personalization**: Does the response appropriately use relevant details from the user profile?*

*- **Overpersonalization**: Does the response include irrelevant, distracting, or unnecessary details about the user?*

*- **Overall**: Taking into account completeness, and being appropriately personalized without being overly personal, is one response clearly better than the other?*

*Important Notes:*

*- Assign a score of 1 if **only response 1** satisfies the criterion.*

*- Assign a score of 2 if \*\*only response 2\*\* satisfies the criterion.*

*- Assign a score of 0 if \*\*both responses\*\* satisfy it or \*\*neither does\*\*.*

*- Do not favor length or style unless it clearly affects the criterion.*

*- Be strict: if the difference is marginal, mark it as a tie (0).*

*Output Instructions:*

*Each of the 4 score fields must be:*

*- '"1"' = only response 1 satisfies the criterion*

*- '"2"' = only response 2 satisfies the criterion*

*- '"0"' = both or neither satisfy the criterion*

*Additionally, assess your confidence in this decision by assigning a certainty level from 1 to 100. Use the following guidelines to assign the certainty level:*

*1–20 (Uncertain): There is insufficient or minimal evidence. The decision is largely based on weak or indirect hints.*

*21–40 (Moderately Confident): There is noticeable evidence supporting a preference, though it is not comprehensive, and other interpretations are possible.*

*41–60 (Quite Confident): There is clear and convincing evidence that supports this decision, though it is not entirely decisive.*

*61–80 (Confident): There is strong evidence that clearly supports this decision, with very little ambiguity.*

*81–100 (Highly Confident): There is direct and explicit evidence that decisively supports this prediction.*

*Lastly, include a \*\*brief (1–2 sentences)\*\* and justify the score using the criteria above.*

*You must return a JSON object with these 12 keys:*

*- "completeness", "completeness_explanation", "completeness_confidence"*

*- "personalization", "personalization_explanation", "personalization_confidence"*

*- "overpersonalization", "overpersonalization_explanation", "overpersonalization_confidence"*

*- "overall", "overall_explanation", "overall_confidence"*

*\*\*Do not include any text outside the JSON block.\*\* Here is the evaluation task:*

*Given: "user input": "user_input", "response_1": "response_a", "response_2": "response_b" Generation:*

**Prompt 12.** *f"'You are an impartial and knowledgeable evaluator. You will be shown a user input and two model-generated responses. Your task is to decide if \*\*only one of the responses\*\* meets the evaluation criteria. If both responses do (or neither does), mark it as a tie (0).*

*Evaluation Criteria:*

*- \*\*Completeness & Accuracy\*\*: Does the response correctly and fully answer the user's request?*

*- \*\*Neutrality\*\*: Is the response unbiased and general, rather than tailored to the user?*

*- \*\*Overall\*\*: Taking into account completeness, accuracy, and neutrality is one response clearly better than the other?*

*Important Notes:*

*- Assign a score of 1 if \*\*only response 1\*\* satisfies the criterion.*

*- Assign a score of 2 if \*\*only response 2\*\* satisfies the criterion.*

*- Assign a score of 0 if \*\*both responses\*\* satisfy it or \*\*neither does\*\*.*

*- Do not favor longer responses unless they add essential information.*

*- Be strict: if the difference is marginal, mark it as a tie (0).*

*Output Instructions: Each of the 4 score fields must be:*

*- '"1"' = only response 1 satisfies the criterion*

*- '"2"' = only response 2 satisfies the criterion*

*- '"0"' = both or neither satisfy the criterion*

*Additionally, assess your confidence in this decision by assigning a certainty level from 1 to 100. Use the following guidelines to assign the certainty level:*

*1–20 (Uncertain): There is insufficient or minimal evidence. The decision is largely based on weak or indirect hints.*

*21–40 (Moderately Confident): There is noticeable evidence supporting a preference, though it is not comprehensive, and other interpretations are possible.*

*41–60 (Quite Confident): There is clear and convincing evidence that supports this decision, though it is not entirely decisive.*

*61–80 (Confident): There is strong evidence that clearly supports this decision, with very little ambiguity.*

*81–100 (Highly Confident): There is direct and explicit evidence that decisively supports this prediction.*

*Lastly, include a \*\*brief (1–2 sentences)\*\* and justify the score using the criteria above.*

*You must return a JSON object with these 12 keys:*

*- "completeness_accuracy", "completeness_accuracy"_explanation", "completeness_accuracy"_confidence"*

*- "neutrality", "neutrality_explanation", "neutrality_confidence"*

*- "overall", "overall_explanation", "overall_confidence"*

*\*\*Do not include any text outside the JSON block.\*\**

*Here is the evaluation task:*

*Given: "user input": "user_input", "response_1": "response_a", "response_2": "response_b" Generation:*

## H   OTHER DETAILS

### H.1   SOFTWARE

We used Python 3.13.2, Pytorch 2.7.0, and HuggingFace Transformers 4.52.4.All code is licensed under the Apache License 2.0.

### H.2   HARDWARE

All experiments were run on a cluster with 4 NIVIDIA H100 GPUs and 1 NIVIDIA A100 GPU with 80B memory. For most inference jobs we used one GPU. For supervised fine-tuning, we used one GPU. The training for twelve epochs took about 2 hours.

### H.3   ARTIFACT TERMS OF USE

PersonaFeedback(Tao et al., 2025): Apache-2.0 FreebaseQA (Jiang et al., 2019): Creative Commons Attribution-NonCommercial-ShareAlike 3.0 International License WildChat (Zhao et al., 2024): AI2 ImpACT License

# I    INFORMATION ABOUT USE OF AI ASSISTANTS

In this project, AI assistants were used for results visualization code (e.g., figures and tables) and for minor writing edits.

## DISTRIBUTION OF PROCESSED PROFILE FEATURES

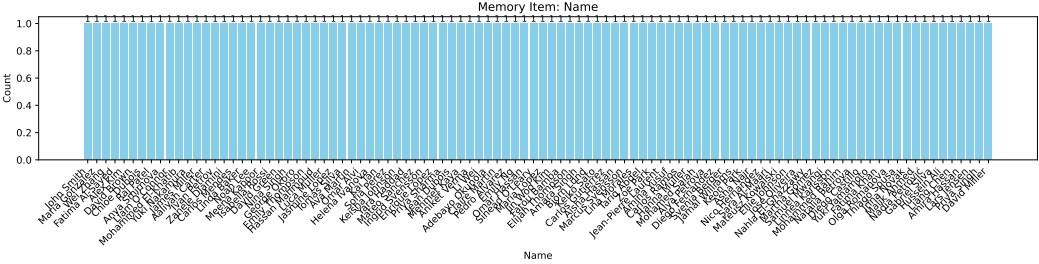

Figure 12: Distribution of Name

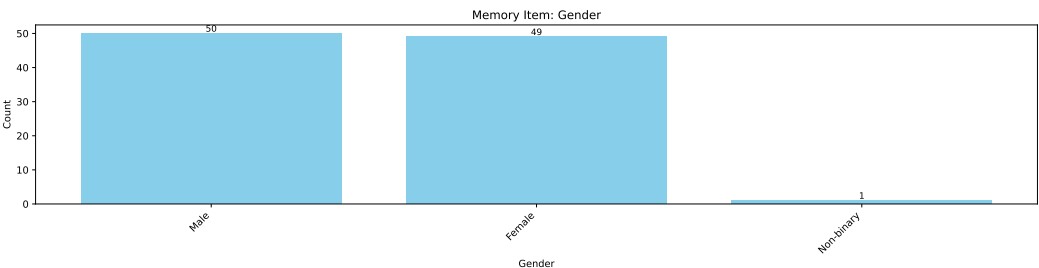

Figure 13: Distribution of Gender

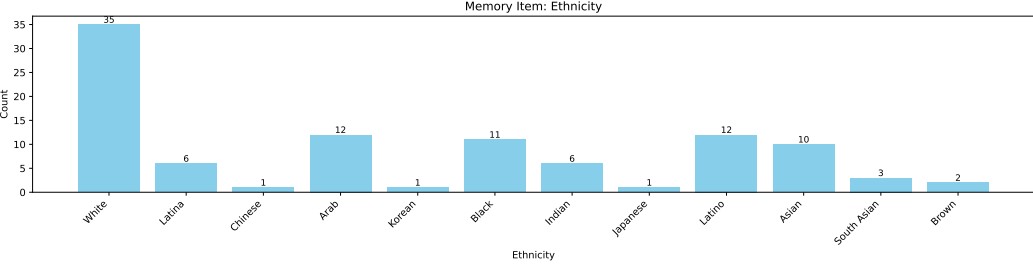

Figure 14: Distribution of Ethnicity

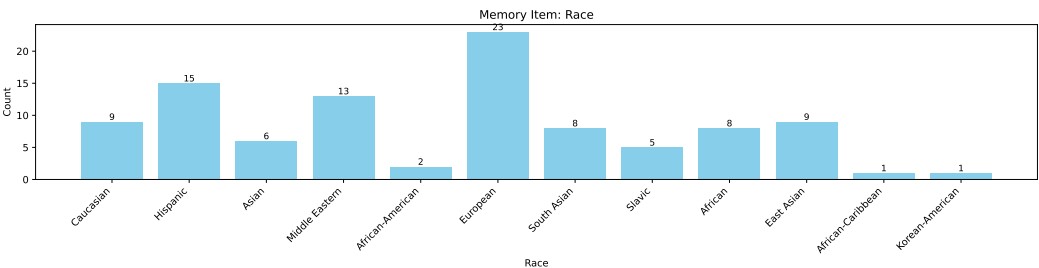

Figure 15: Distribution of Race

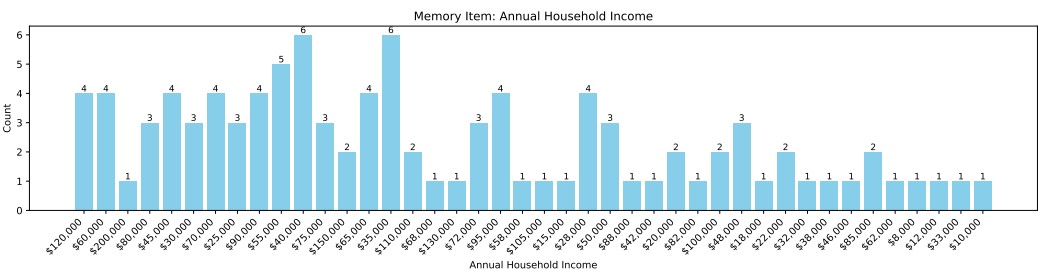

Figure 16: Distribution of Annual Household Income

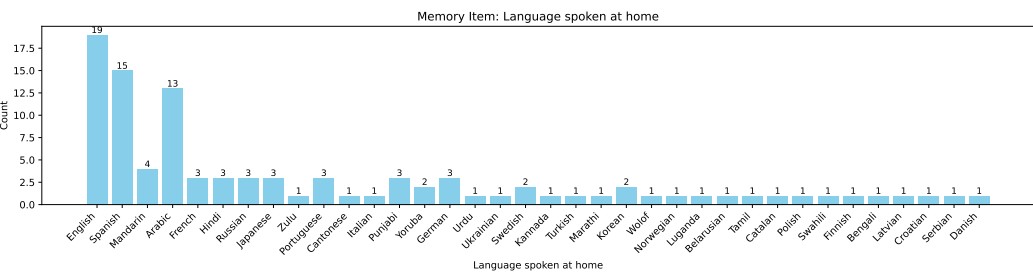

Figure 17: Distribution of Language spoken at home

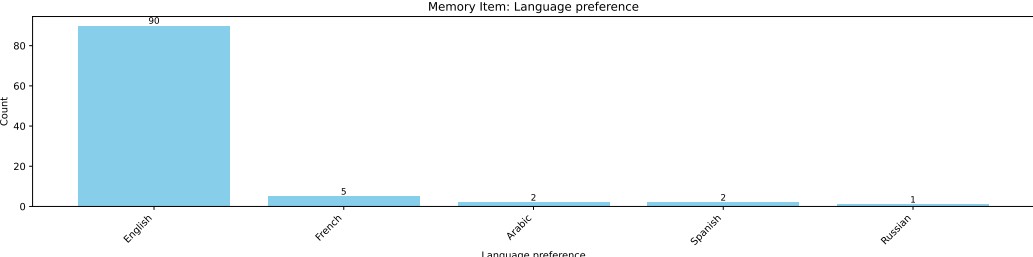

Figure 18: Distribution of Language preference

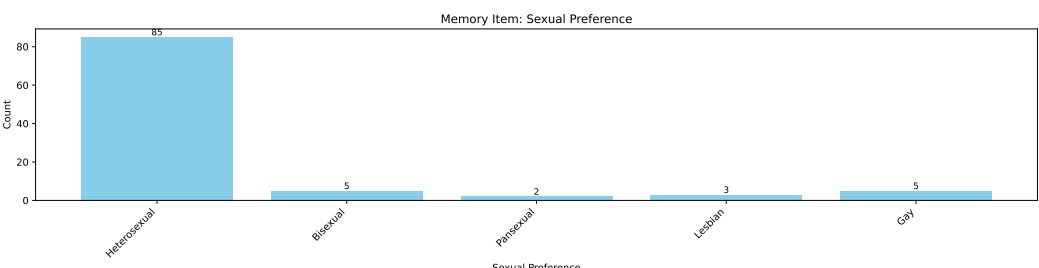

Figure 19: Distribution of Sexual Preference

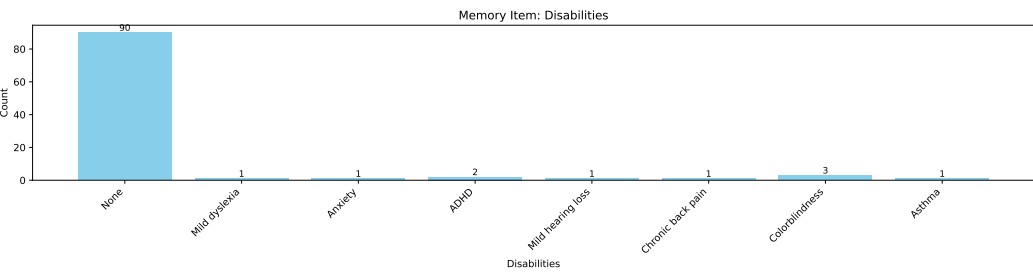

Figure 20: Distribution of Disabilities

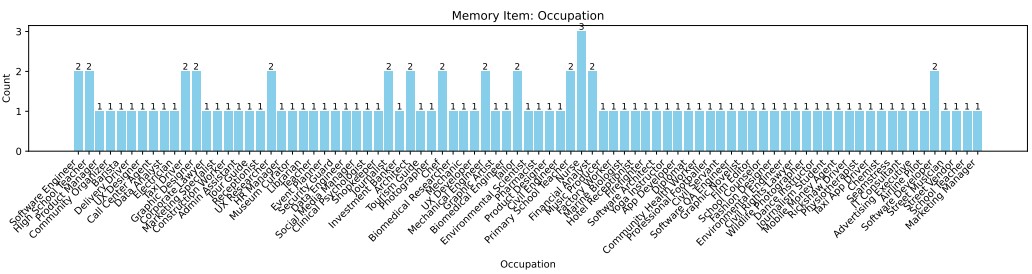

Figure 21: Distribution of Occupation

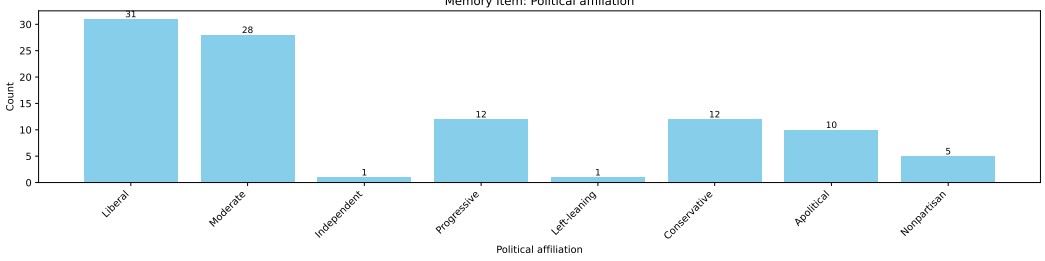

Figure 22: Distribution of Political affiliation

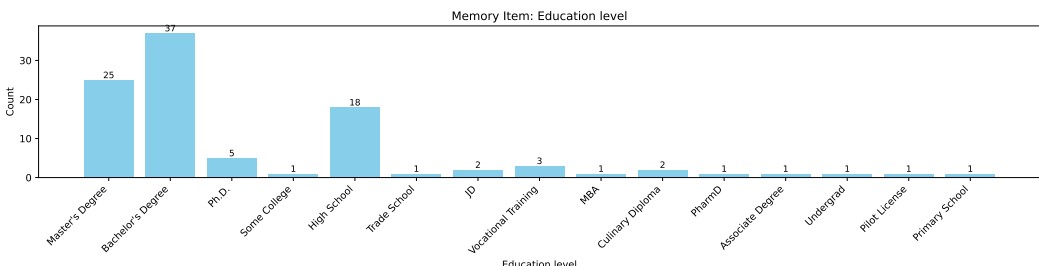

Figure 23: Distribution of Education level

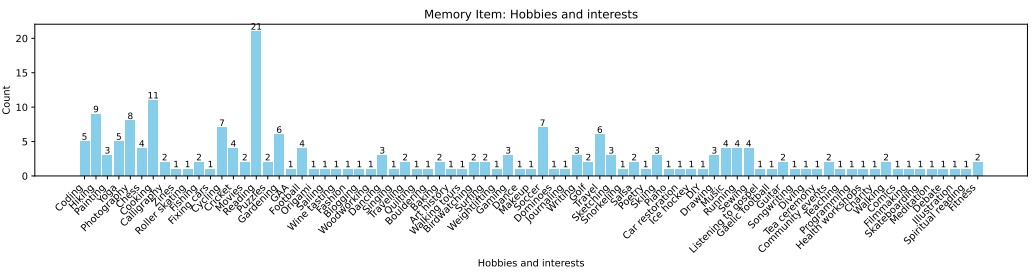

Figure 24: Distribution of Hobbies and interests

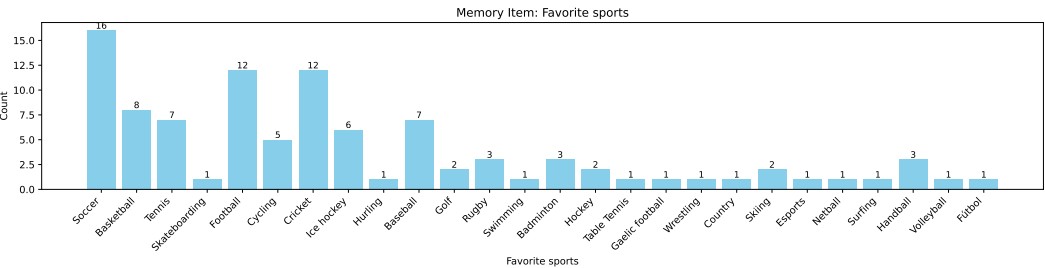

Figure 25: Distribution of Favorite sports

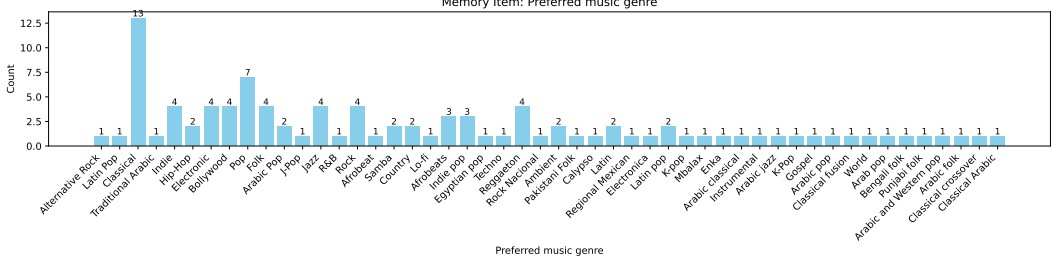

Figure 26: Distribution of Preferred music genre

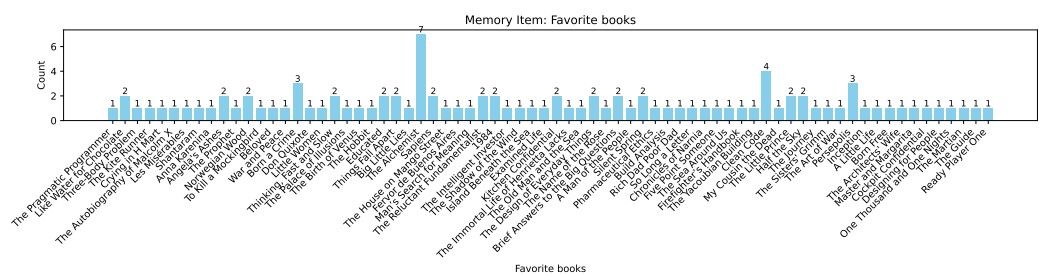

Figure 27: Distribution of Favorite books

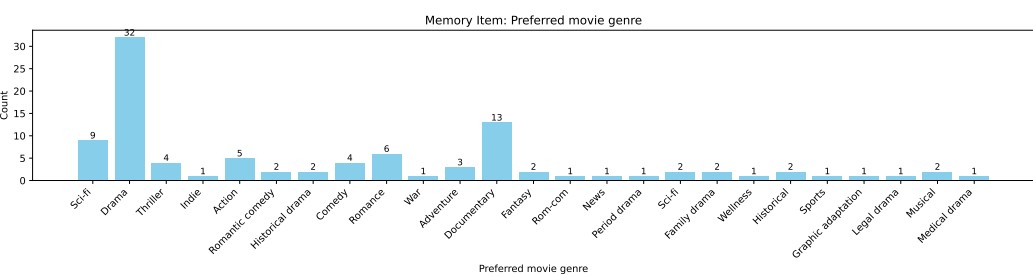

Figure 28: Distribution of Preferred movie genre

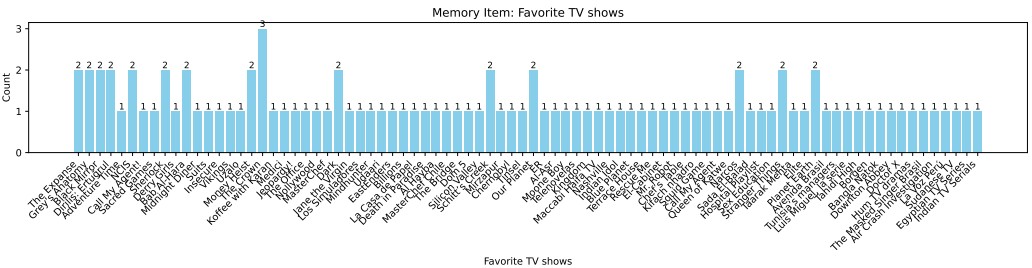

Figure 29: Distribution of Favorite TV shows

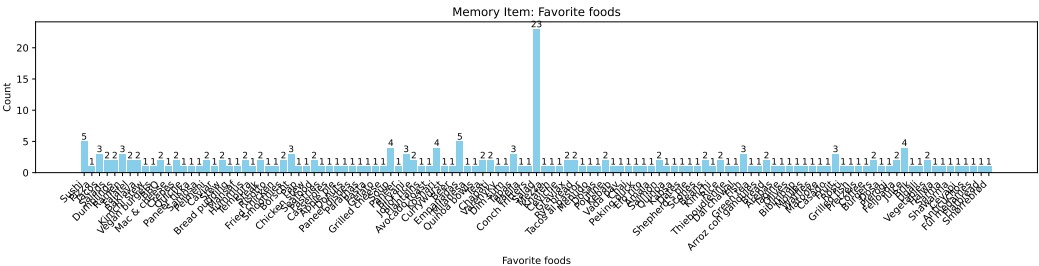

Figure 30: Distribution of Favorite foods

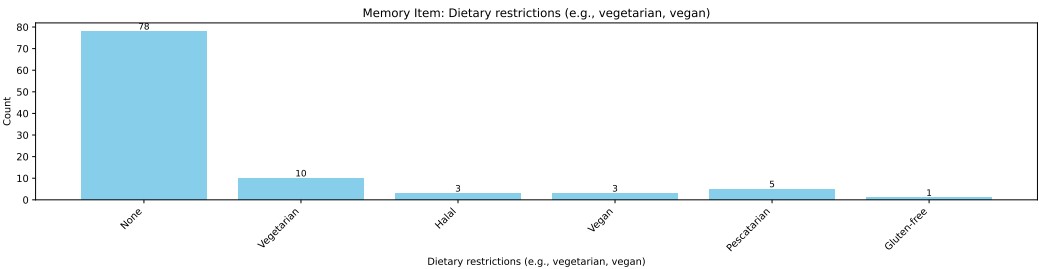

Figure 31: Distribution of Dietary restrictions (e.g., vegetarian, vegan)

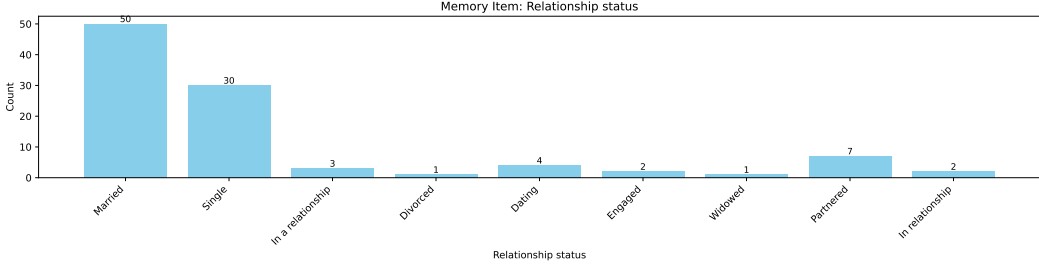

Figure 32: Distribution of Relationship status

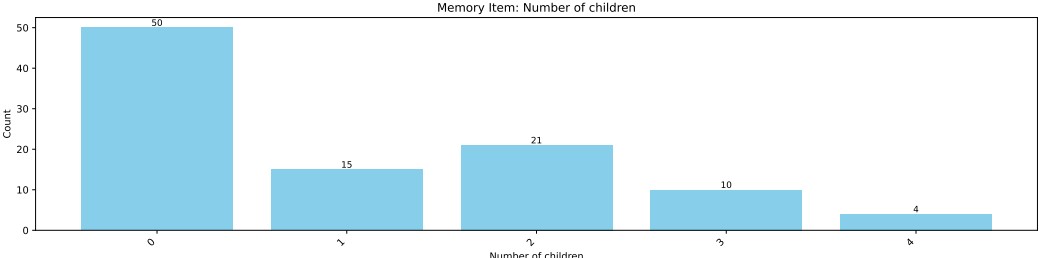

Figure 33: Distribution of Number of children

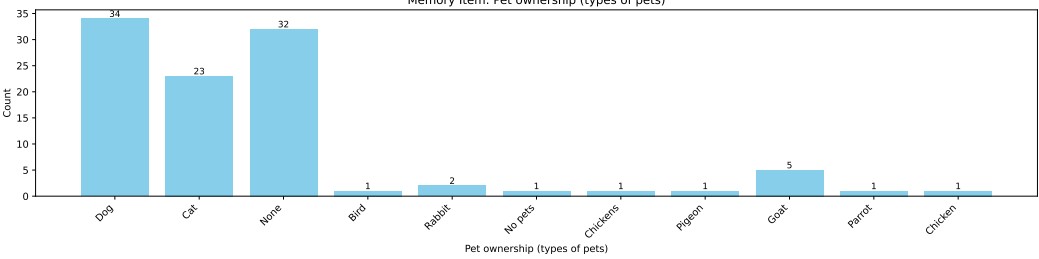

Figure 34: Distribution of Pet ownership (types of pets)

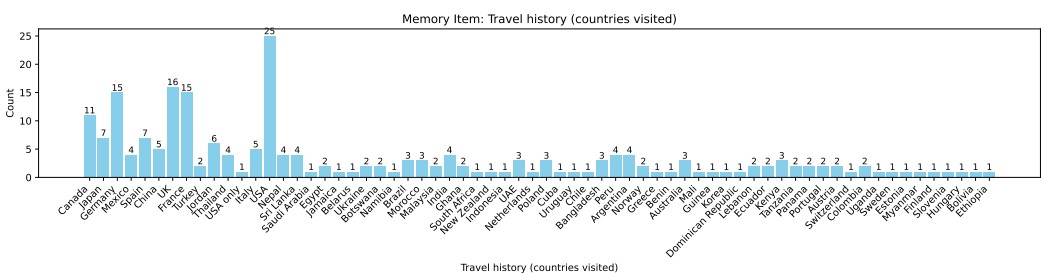

Figure 35: Distribution of Travel history (countries visited)

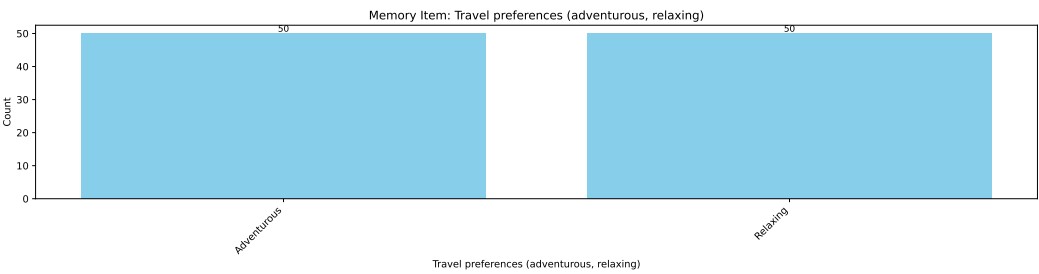

Figure 36: Distribution of Travel preferences (adventurous, relaxing)

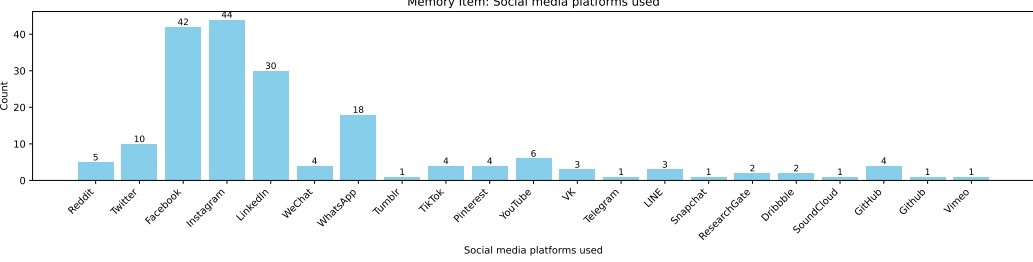

Figure 37: Distribution of Social media platforms used

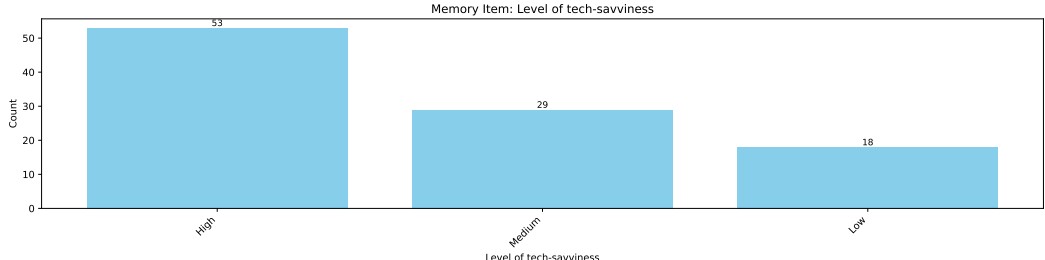

Figure 38: Distribution of Level of tech-savviness

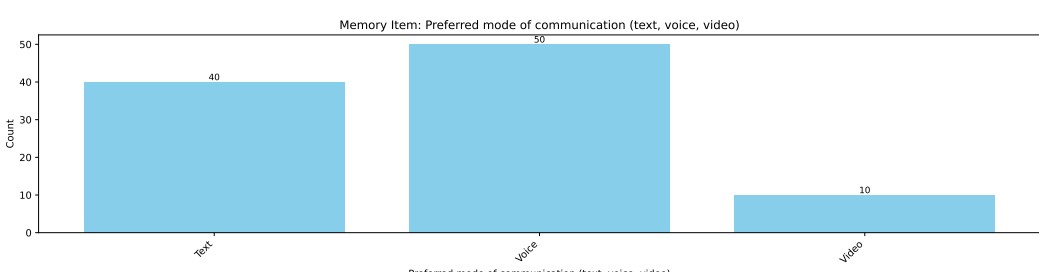

Figure 39: Distribution of Preferred mode of communication (text, voice, video)

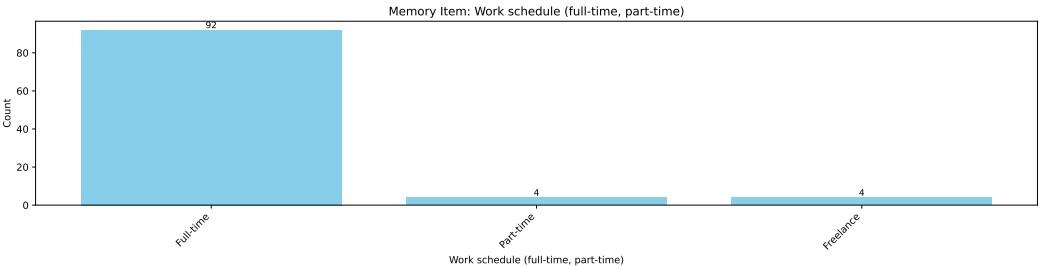

Figure 40: Distribution of Work schedule (full-time, part-time)

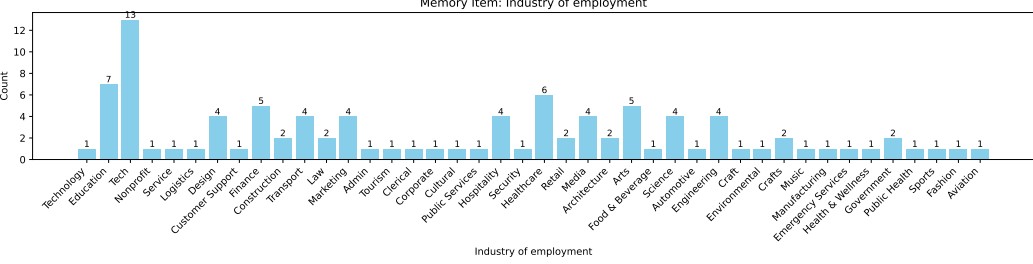

Figure 41: Distribution of Industry of employment

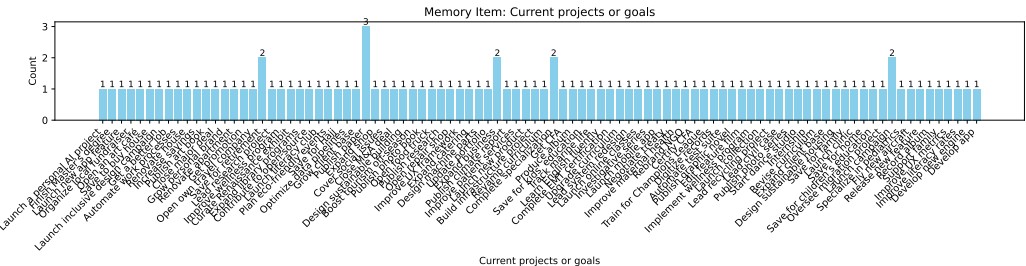

Figure 42: Distribution of Current projects or goals

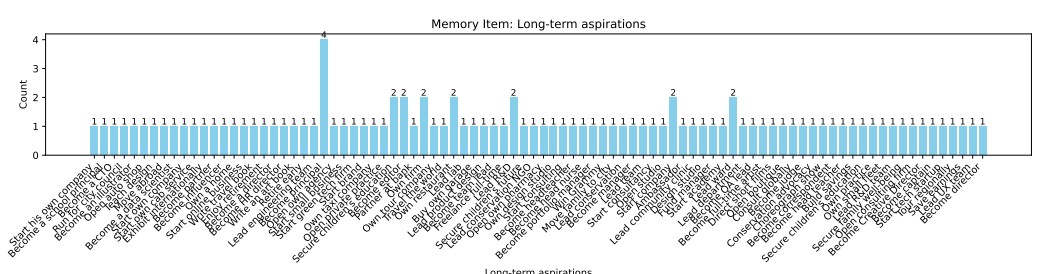

Figure 43: Distribution of Long-term aspirations

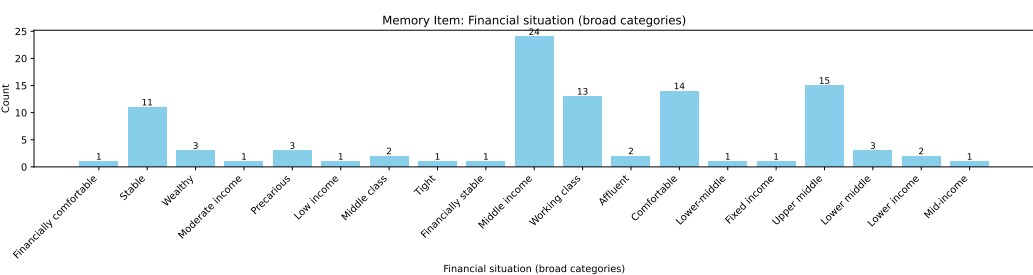

Figure 44: Distribution of Financial situation (broad categories)

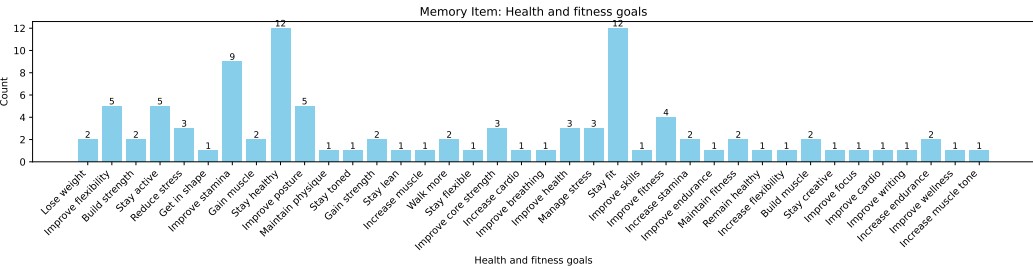

Figure 45: Distribution of Health and fitness goals

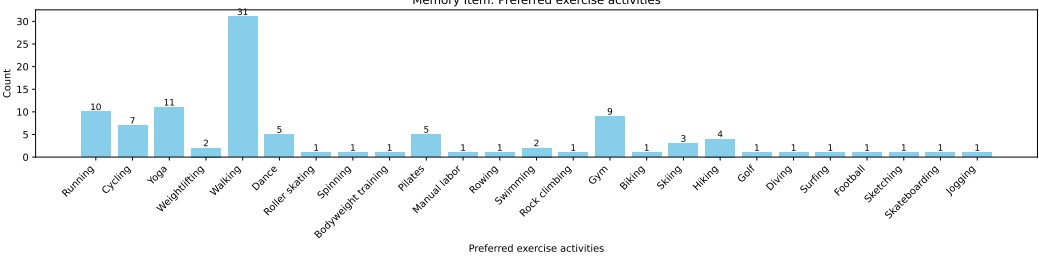

Figure 46: Distribution of Preferred exercise activities

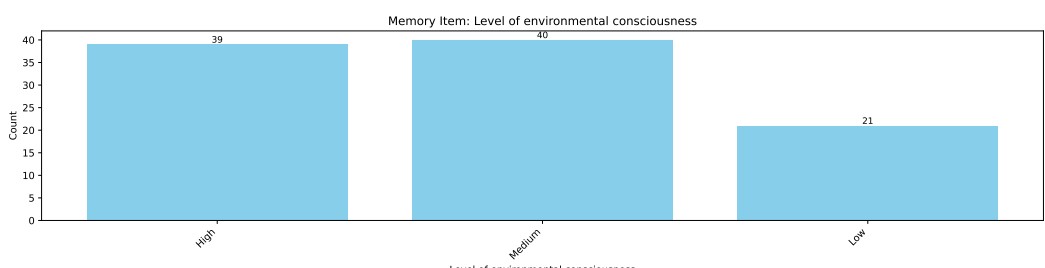

Figure 47: Distribution of Level of environmental consciousness

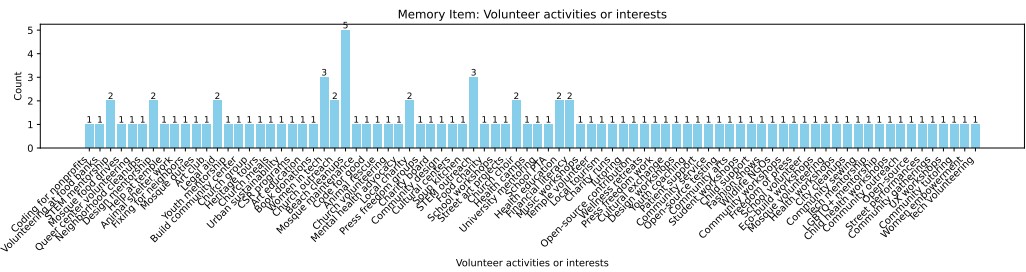

Figure 48: Distribution of Volunteer activities or interests

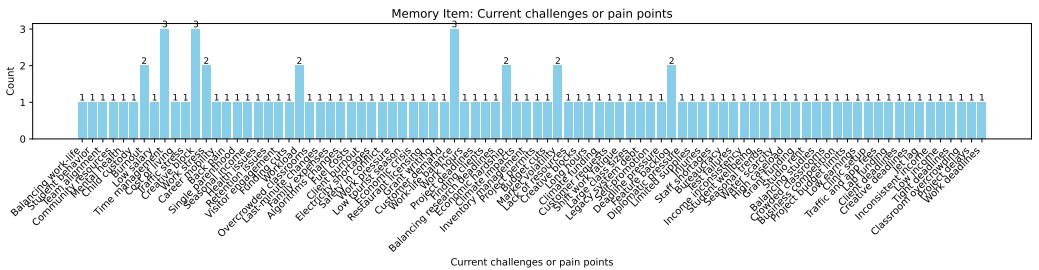

Figure 49: Distribution of Current challenges or pain points

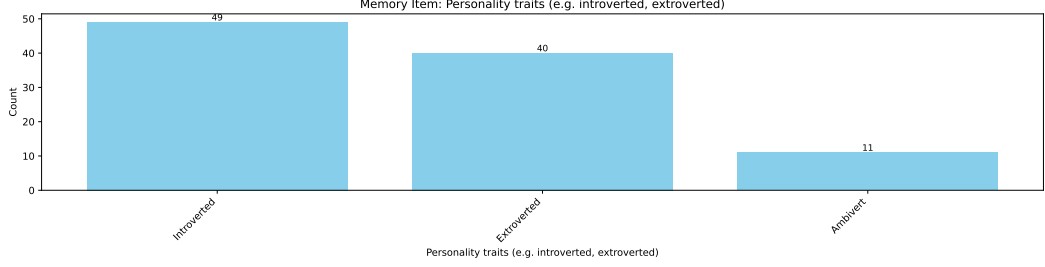

Figure 50: Distribution of Personality traits (e.g. introverted, extroverted)

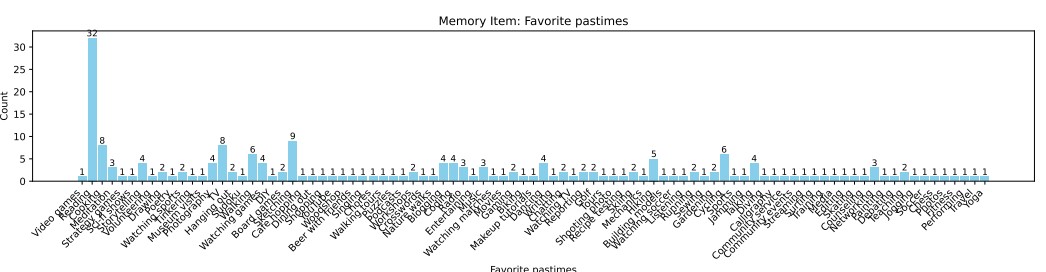

Figure 51: Distribution of Favorite pastimes

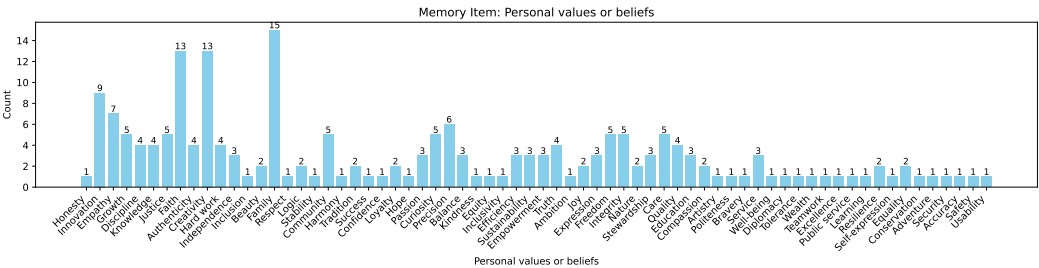

Figure 52: Distribution of Personal values or beliefs

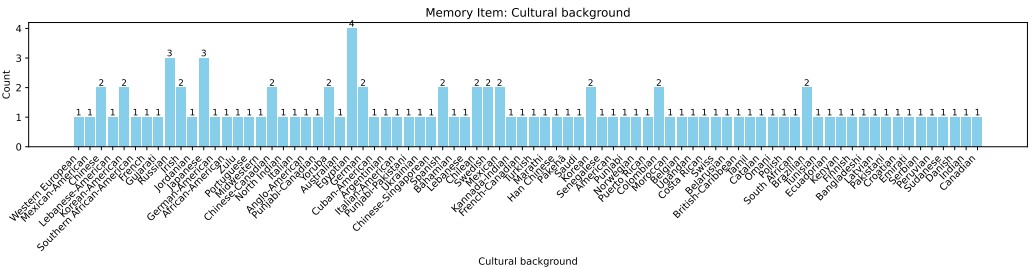

Figure 53: Distribution of Cultural background

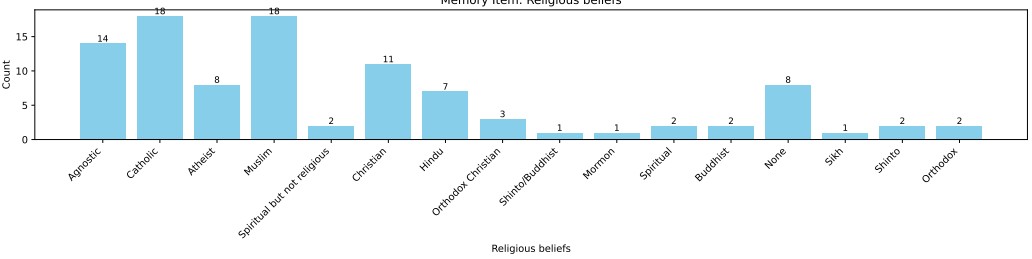

Figure 54: Distribution of Religious beliefs

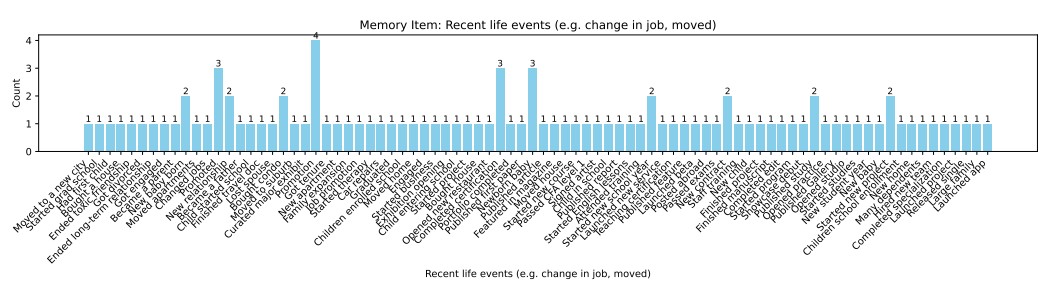

Figure 55: Distribution of Recent life events (e.g. change in job, moved)

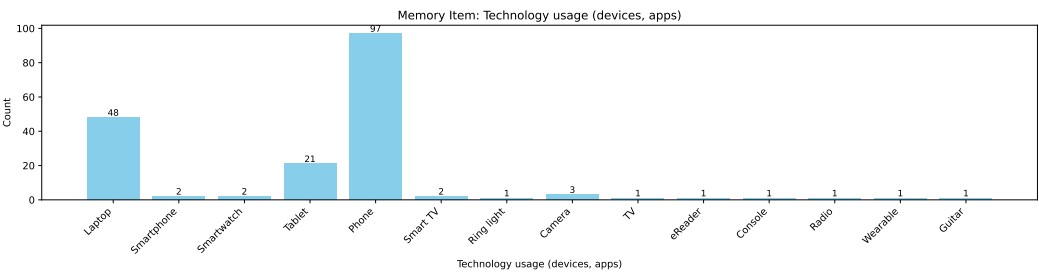

Figure 56: Distribution of Technology usage (devices, apps)

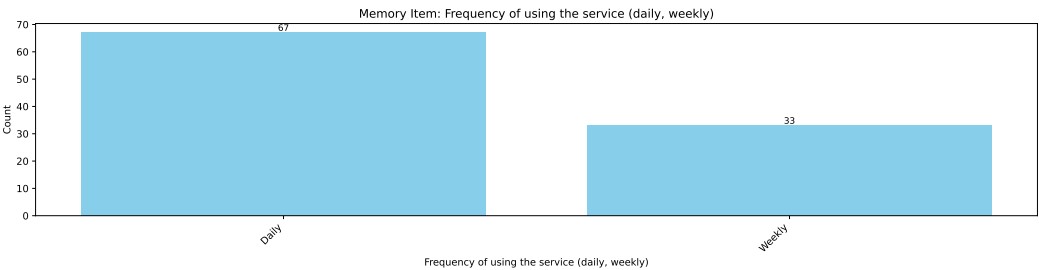

Figure 57: Distribution of Frequency of using the service (daily, weekly)

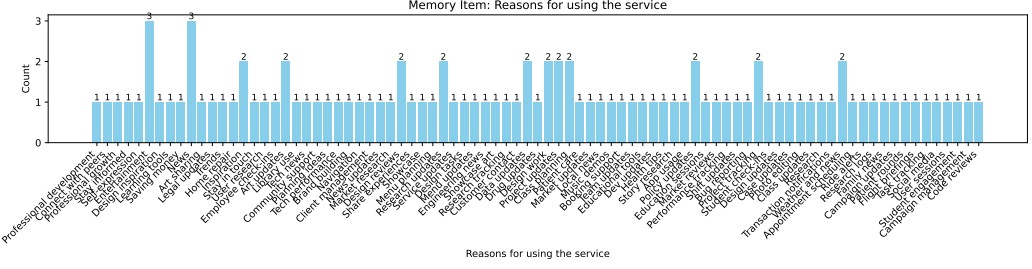

Figure 58: Distribution of Reasons for using the service

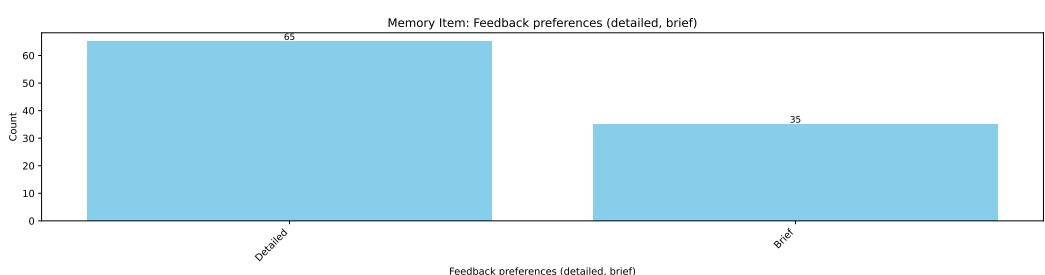

Figure 59: Distribution of Feedback preferences (detailed, brief)

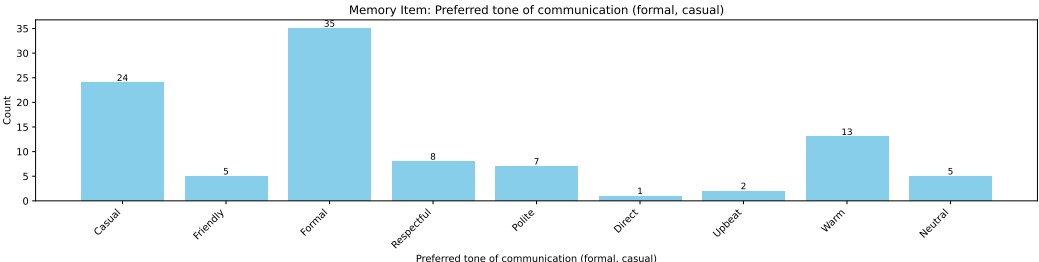

Figure 60: Distribution of Preferred tone of communication (formal, casual)

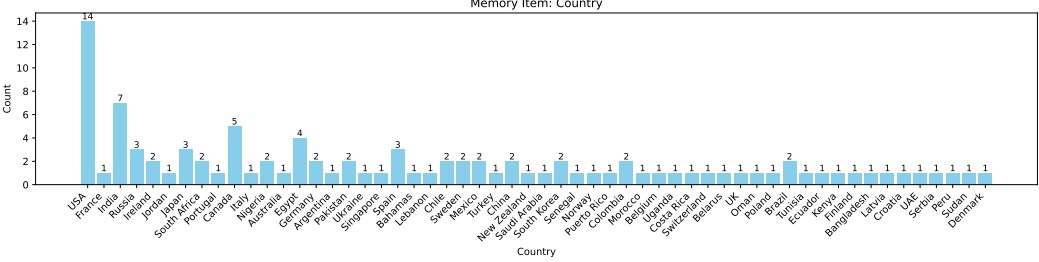

Figure 61: Distribution of Country

