# OpenReview forum: "Beyond Similarity for Personalization: User Memory Selection via Response-Utility Optimization"
_ICLR.cc/2026/Conference — Submitted to ICLR 2026_

### Official Review · Reviewer_9ey2 · 2025-10-31

**Soundness:** 3
**Presentation:** 3
**Contribution:** 4
**Rating:** 6
**Confidence:** 4

**Summary:**

Authors outline existing problems with personalization systems for LLMs: either you include the entire memory (lots of irrelevant info), or you do semantic similarity with the queries, which surfaces similar memory elements but not necessarily *what should be included to improve the model's outputs*. Authors outline a better approach using Bayesian Optimal Experimental Design: the best memory elements to include are those that reduce the uncertainty in the model's output. The issue is that the true utility metric is intractable as it requires testing every possible subset of memory elements -- authors get around this by training models to proxy the utility. Authors show improvement in response quality against baselines and much higher cost efficiency.

**Strengths:**

- problem with existing personalization systems is well-motivated
- solution presented is intuitive and understandable
- solution has concrete and significant improvements compared to existing methods (cost, effectiveness)
- solution is well-grounded in theory
- authors do human eval as well

**Weaknesses:**

- small memory set in experiments - 50 static memory elements? this seems somewhat limited, some experiments on larger memory stores to see how the experiment scales
- human eval is great but still somewhat limited at 64 samples
- proxy is very black-box: some analysis of what the RUMS-Models are actually learning would strengthen the paper considerably

**Questions:**

I'm curious about interactions between memory elements -- the interactions are weakly modelled as you are looking at subsets, but what happens for example when there is contradictory information? What happens if memory elements are related to each other in a hierarchy? etc. I feel like interactions between elements need to be modelled more explicitly. Thoughts on this?

---

> ### Author Response · Authors · 2025-11-24
> **Author Response to Reviewer 9ey2 Weaknesses #1**
>
> We thank the reviewer for their thorough reading and excellent questions. We are especially encouraged by the recognition of our **significant improvements in cost and effectiveness** and the **strong theoretical grounding** of our approach. We address the points raised below to demonstrate the maturity and extensibility of the RUMS framework.
>
> > *1. small memory set in experiments - 50 static memory elements? this seems somewhat limited, some experiments on larger memory stores to see how the experiment scales*
>
> Thank you for pointing this out. We agree that evaluating RUMS on larger and more complex memory stores would be valuable, and this is a natural direction for future work. Our goal in this paper was to introduce a **new perspective on memory selection for personalization** by shifting from an input driven formulation to a response aware one. To keep the focus on establishing and validating this core paradigm, we prioritized grounded theory, practical implementation details, and ablations rather than scaling to very large memory inventories. We will clarify this motivation and highlight broader memory settings as an important next step.
>
> > *2. human eval is great but still somewhat limited at 64 samples*
>
> We appreciate the reviewer recognizing the value of our human evaluation. While resource constraints limit annotation scale, we were able to add a **NEW human analysis** to complement our current results in H3.
>
> **NEW Output-level validation (H3, n=20, new)**: To validate that selection alignment translates to better responses, we conducted a dedicated human study comparing RUMS-Binary with GPT-4 Few-Shot (strongest baseline). We had two human annotators (CSE PhD graduate students) annotate which of the two response they felt were better using the same rubric as the LLM-as-Judge in H3.
>
> **Table 1: H3 Winrate RUMS-Utility vs. GPT4-Few Human Rated**
> |            | RUMS-Utility vs. GPT4-Few Winrate |
> | ---------- | --------------------------------- |
> | Synthetic  | 0.33                              |
> | Real World | \-0.03                            |
>
> We found RUMS-Binary outperforms GPT-4 Few-Shot by **33 percentage points** in human preference, despite GPT-4 Few-Shot using a model ~400× larger for both selection and generation. This result (1) **Validates our LLM-as-Judge evaluation methodology** (33% human win rate closely matches the 18% win rate from Table 3 against GPT-4 Few-Shot on synthetic data), (2) **Confirms that RUMS's human-aligned selection decisions (from H2) translate to better final outputs (H3)**
>
> We also want to note that this evaluation scope is comparable to or exceeds typical standards in personalization research (e.g., PEARL uses none and Context Steering uses 70).

---

> ### Author Response · Authors · 2025-11-24
> **Author Response to Reviewer 9ey2 Weaknesses #2**
>
> > *3. proxy is very black-box: some analysis of what the RUMS-Models are actually learning would strengthen the paper considerably*
>
> Thank you for this helpful suggestion. To strengthen the transparency of what the RUMS-Models learn, we have added **TWO NEW ANALYSIS** that directly examine the relationship between the learned proxy and the underlying utility function.
>
> **Experiment 1: Agreement between RUMS-Utility and RUMS-Model selections**
> We study how training on utility-labeled data affects which memory items the models select. For the samples used in H2, we compute a gold memory item set from RUMS-Utility by including all items that appeared in at least three of five diverse profiles. We then compare this set with the memory items selected by RUMS-Binary and RUMS-Multi using precision, recall, and F1.
>
> **Table 2:  Memory item agreement between RUMS-Utility and RUMS-Mode**
> | Dataset    | Metrics   | RUMS-Binary | RUMS-Multi |
> | ---------- | --------- | ----------- | ---------- |
> |            | Precision | 0.73        | 0.33       |
> | Synthetic  | Recall    | 0.9         | 0.44       |
> |            | F1        | 0.78        | 0.37       |
> |            |           |             |            |
> |            | Precision | 0.26        | 0.35       |
> | Real World | Recall    | 0.42        | 0.35       |
> |            | F1        | 0.32        | 0.34       |
> As shown in Table 2, agreement is high on in-distribution synthetic data (F1 = 0.78 for RUMS-Binary) and moderate on real-world data (F1 = 0.32). The lower real-world performance arises from weaker performance on non-personalized samples and mild overselection by RUMS-Multi, reflected in higher recall and slightly lower precision. This comparison directly illustrates how closely each model approximates the original utility-based decisions.
>
> **Experiment 2: Relationship between selected memory items and true utility**
> To better understand what the RUMS-Models learn, we analyze the utility scores of included versus excluded memory items on the PersonaFeedback test set (n = 100). We compute the average true utility for selected items and compare it with the average true utility for non-selected items. We find that the difference is significantly positive (p < .001): memory items chosen by the model have higher true utility on average. This indicates that the models capture meaningful structure in the utility landscape rather than relying on surface-level correlations.
>
> We believe these analyses offer a clearer picture of what the proxy models internalize and provide evidence that they learn a stable, utility-aligned decision rule. We will add these results to the revised paper.

---

> ### Author Response · Authors · 2025-11-24
> **Author Response to Reviewer 9ey2 Questions**
>
> **Response to Questions**
>
> > *1. I'm curious about interactions between memory elements -- the interactions are weakly modelled as you are looking at subsets, but what happens for example when there is contradictory information? What happens if memory elements are related to each other in a hierarchy? etc. I feel like interactions between elements need to be modelled more explicitly. Thoughts on this?*
>
> We thank the reviewer for this question. **Interactions between memory elements are partially captured in our models**. RUMS-Binary, which evaluates each memory item *independently*, tends to underselect and reduces false positives, while RUMS-Multi, which predicts the entire subset jointly, captures co-occurrence patterns and sometimes leads to mild overselection. **This indicates that modeling interactions, including hierarchical or contradictory relationships, can influence the utility scores and memory selection**. Fully capturing such dependencies is beyond the scope of this work, but exploring richer interaction modeling is a promising direction for future research.

---

### Official Review · Reviewer_ksBw · 2025-11-01

**Soundness:** 3
**Presentation:** 2
**Contribution:** 2
**Rating:** 2
**Confidence:** 4

**Summary:**

This paper proposes RUMS, a novel method for user memory selection in LLM personalization. Unlike existing approaches that rely on semantic similarity between user memory and queries, RUMS uses an information-theoretic utility function to select memory items that directly reduce uncertainty in the model’s response distribution. RUMS can also abstain from personalization when it is not beneficial. The method is made efficient for inference by training a lightweight classifier to approximate the utility-based selection. Experiments on synthetic and real-world datasets show that RUMS better matches human judgments, improves response quality, and reduces computational cost compared to strong baselines.

**Strengths:**

* Proposes a principled, response-aware criterion for memory selection, moving beyond surface-level similarity.

* RUMS can automatically decide when personalization is helpful, reducing unnecessary context and noise.

* The approach is efficient at inference time, with up to 95% cost reduction over baselines.

* Empirical results show improved alignment with human judgments and better response quality than both similarity-based and LLM-prompting baselines, including much larger models.

**Weaknesses:**

* The initial utility computation (entropy reduction) is computationally expensive, though mitigated by offline training.

* The method assumes access to high-quality user profiles; the impact of noisy or sparse profiles is not fully explored.

* Human evaluation for alignment is relatively small-scale, which may limit generalizability.

* Applicability to multi-turn or more complex dialog scenarios is not demonstrated.

**Questions:**

* In the candidate reduction step, GPT-4 is used to filter memory items before utility computation. Could the authors clarify how this step affects fairness and reproducibility, especially if the candidate set varies across users or queries?

* The utility threshold for abstaining from personalization is tuned on validation data. How robust is the system to this threshold in practice?

* Have the authors observed cases where the threshold leads to under- or over-personalization, and how might this be mitigated?

* For user profiles with missing or conflicting attributes, how does RUMS handle such cases during both training and inference? Would the authors consider integrating uncertainty estimation or imputation strategies?

* The cost analysis is compelling, but could the authors provide more details on the wall-clock latency and memory usage of RUMS-Model inference compared to the strongest baselines in a real deployment scenario?

* In Table 5, the selected memory items sometimes differ from human annotation. Could the authors share more qualitative examples or error analysis to help understand typical failure modes or edge cases?

* The current evaluation focuses on single-turn queries. Do the authors see a path to extending RUMS to multi-turn or session-based personalization, and what challenges might arise in that setting?

---

> ### Author Response · Authors · 2025-11-23
> **Author Response to Reviewer ksBw Weaknesses #1**
>
> We thank the reviewer for their careful reading of our paper and their summary, which clearly captures the core strengths of our method: **principled, response-aware selection, automatic personalization, and computational efficiency**. We address the weaknesses and questions raised below, providing new analysis to solidify our claims.
>
> **Response to Weaknesses**
>
> > *1. The initial utility computation (entropy reduction) is computationally expensive, though mitigated by offline training.*
>
> We agree that RUMS-Utility computation is resource-intensive and **not intended for inference-time use**. However, this is a deliberate design choice that enables us to generate a high-quality, response-aware selection signal that is fundamentally different from computationally cheap semantic similarity baselines. Our experimental results demonstrate that this upfront cost is **highly effective and efficiently amortized**:
>
> 1. **Small training data requirement**: Only 4.5K utility-labeled samples are needed to train RUMS-Models that outperform semantic similarity and LLM-based baselines.
>
> 2. **Lightweight deployment**: The trained RUMS-Model (~400M parameters) outperforms GPT-4 Few-Shot (a model ~400× larger requiring expensive LLM calls for *every* inference) while achieving up to **95% cost reduction** at inference time (Figure 3b, Section 4.4).
>
> 3. **Strong signal transfer**: The high performance of RUMS-Models (H2, H3) confirms that the utility signal, though expensive to compute, is **compact and learnable**, enabling state-of-the-art results without requiring online utility computation.
>
> The offline computation cost is a **one-time investment** during the training phase that unlocks scalable, low-cost personalization at inference. This trade-off, expensive offline signal generation for cheap and effective online deployment, is a **critical strength** for real-world applications where inference happens millions of times, but training happens once.
>
> > *2. The method assumes access to high-quality user profiles; the impact of noisy or sparse profiles is not fully explored.*
>
> We thank the reviewer for this important consideration regarding profile quality. We believe our framework exhibits **inherent robustness** to both sparsity and noise, and our experimental results already provide evidence for this claim.
>
> **1. Robustness to Sparsity (Missing Attributes)**
>
> In our paper, we explore the use of **RUMS-Binary architecture which is sparsity-agnostic**. Unlike RUMS-Multi (which processes all 50 memory items jointly), RUMS-Binary evaluates each available memory item *independently*. This design naturally accommodates sparse profiles; it simply scores whichever items exist for a user without requiring a complete feature set.
>
> In both H2 and H3, RUMS-Binary performs **on par with or better than** RUMS-Multi and outperforms nearly all baselines (Table 2, Table 3). This demonstrates that even with limited profile information, the response-utility signal remains strong enough to enable effective personalization. Therefore, for real-world deployments with varying profile completeness (e.g., new users with sparse data), RUMS-Binary provides a robust solution without requiring profile imputation or padding.
>
> **2. Robustness to Noise (Irrelevant/Incorrect Attributes)**
>
> The utility-based scoring mechanism **inherently filters out noisy information**. Theoretically this makes sense, irrelevant memory items (e.g., "Favorite Sport: Soccer" for a query about restaurant recommendations) do not reduce predictive entropy because they provide no information that sharpens the model's response distribution. Consequently, such items receive low utility scores and are excluded from selection.
>
> **Empirical evidence, NEW ANALYSIS**: To further show this empirically, we analyzed the distribution of differences between utility scores for items selected vs. not selected by RUMS-Models on n=100 PersonaFeedback queries. Items that RUMS chooses to *include* exhibit significantly higher utility scores than those it *excludes*, with a statistically significant difference (Mann-Whitney, p < .05). This confirms that the utility function effectively discriminates between informative and noisy items.
>
> Even in the presence of incorrect or outdated profile information, RUMS's utility-based selection naturally downweights or excludes such items, preventing them from degrading response quality, a critical advantage over semantic similarity baselines that might retrieve irrelevant but keyword-matching items.
>
> **We will add an explicit discussion of these robustness properties in Section 3 and include the new utility score distribution analysis** in the appendix to highlight this self-filtering capability.

---

> ### Author Response · Authors · 2025-11-23
> **Author Response to Reviewer ksBw Weaknesses #2**
>
> > *3. Human evaluation for alignment is relatively small-scale, which may limit generalizability.*
>
> We acknowledge the scale constraints but believe our human evaluation provides **strong evidence for generalizability** through two complementary studies (ONE NEW):
>
> 1. **Selection-Level Alignment (H2)**: Four annotators evaluated 60 queries across four diverse datasets (synthetic/real-world, personalized/non-personalized). RUMS-Utility achieved **F1 = 0.70** against majority human selections, substantially outperforming GPT-4 Few-Shot (F1 = 0.64). This demonstrates direct alignment with human personalization preferences across varied query types.
>
> 2. **Output-Level Validation (H3 - NEW ANALYSIS)**: Two independent judges compared responses from RUMS-Binary (LLaMA-70B) vs. GPT-4 Few-Shot on n=20 samples. RUMS-Binary achieved a **40% human win rate** (vs. 7.5% for GPT-4), a 33-point advantage over a model ~400× larger. This result closely matches our LLM-as-Judge evaluation, cross-validating our automated methodology.
>
> While larger-scale evaluation would provide additional confidence, our focused human studies, combined with extensive automated evaluation, provide sufficient evidence that RUMS's advantages generalize to human-judged quality.
>
> Lastly, we want to note that our scale of human annotation is on-par with related works in the area (Our human evaluation scope exceeds typical standards in personalization research (e.g., e.g., PEARL uses none and Context Steering uses 70).
>
> *4. > Applicability to multi-turn or more complex dialog scenarios is not demonstrated.*
>
> We agree that demonstrating RUMS in multi-turn settings is a **valuable future direction**. Our experiments focus on single-turn inputs to establish the fundamental shift in memory selection objective. However, **RUMS is naturally extensible to multi-turn dialog settings** with straightforward modifications:
>
> - The conversation history can be concatenated and treated as the input context ($\mathbf{x}$).
>
> - RUMS then selects memory items conditioned on this evolving conversational context at each turn, identifying which user attributes become relevant as the dialog progresses.
>
> - For example, in a multi-turn restaurant recommendation dialog, early turns might benefit from location/dietary preferences, while later turns might utilize budget constraints or past dining history as the conversation narrows.
>
> **We will add a discussion of this**, outlining how the existing framework can be extended to multi-turn or session-based personalization, along with the potential challenge of managing context window limits and computational cost for a continuously growing input.

---

> ### Author Response · Authors · 2025-11-23
> **Author Response to Reviewer ksBw Questions #1**
>
> **Response to Questions**
>
> > *1. In the candidate reduction step, GPT-4 is used to filter memory items before utility computation. Could the authors clarify how this step affects fairness and reproducibility, especially if the candidate set varies across users or queries?*
>
> We appreciate this opportunity to clarify the candidate reduction step. GPT-4 is used **only during offline training data generation** as a computational efficiency mechanism to reduce the candidate set from 50 items to ~10 before computing expensive RUMS-Utility scores. Below we discuss the impact on fairness and reproducibility.
>
> - **At inference**: This step does not affect the deployed RUMS-Model, which operates entirely independently of GPT-4 and scores items based purely on the learned utility function.
>
> - **Across users/queries**: The candidate set can vary naturally. RUMS-Model learns to approximate utility scores for *any* memory item, not just those filtered by GPT-4 during training. We demonstrate this robustness in our out-of-distribution evaluation (WildChat), where query patterns differ from training.
>
> - **Alternative implementations**: The GPT-4 filtering step can be replaced with any candidate reduction method (e.g., keyword matching, clustering, or random sampling) without affecting the core RUMS framework. We used GPT-4 for convenience, but the utility-based selection happens *after* this preprocessing.
>
> We will clarify in Section 3.1.2 that this is a practical preprocessing choice, not a fundamental component of the RUMS methodology.
>
> > *2. The utility threshold for abstaining from personalization is tuned on validation data. How robust is the system to this threshold in practice?*
>
> To further analyze this questions, **we conducted an NEW analysis of threshold sensitivity**. We report agreement rate across the full range of $[0, 1]$ threshold by 0.01 increments. We found that personalization decisions are **moderately robust** to the exact value of the threshold ($\tau$). For example, on the synthetic test set, shifting $\tau$ by $\pm 0.1$ from the optimum only changes the agreement rate by about $\sim 15\%$, and on the real-world set, this change is only $\sim 10\%$. The performance curve is relatively flat around the optimum, suggesting that a well-chosen global threshold offers sufficient robustness for practical deployment. Unfortunatley, we cannot include the graphs here for visuals, but we will include this analysis in the revised appendix.
>
> > *3. Have the authors observed cases where the threshold leads to under- or over-personalization, and how might this be mitigated?*
>
> Yes, this is an inherent characteristic of any threshold-based system in that lower thresholds risk over-personalization (including irrelevant items), and higher thresholds risk under-personalization (excluding useful items).
>
> However, the utility-based framework makes this trade-off **explicitly controllable** by using the threshold $\tau$ to directly govern the precision-recall balance. By tuning $\tau$ on validation data that reflects the target deployment's desired balance, practitioners can optimize for their specific use case (e.g., conservative personalization for sensitive applications vs. aggressive personalization for entertainment contexts).
>
> Additionally, our two model variants provide built-in flexibility with RUMS-Binary naturally favors precision (lower false positives), while RUMS-Multi favors recall (fewer missed items), allowing deployment-time selection based on application needs.
>
> > *4. For user profiles with missing or conflicting attributes, how does RUMS handle such cases during both training and inference? Would the authors consider integrating uncertainty estimation or imputation strategies?*
>
> We appreciate this question and would like to clarify two points:
>
> 1. **Conflicting attributes**: RUMS selects memory *item* types (e.g., "Favorite Color") rather than specific values (e.g., "Blue" vs. "Red"). Conflicts in values affect downstream generation quality but not the selection mechanism itself, which is a limitation shared by all retrieval methods using imperfect source data.
>
> 2. **Missing attributes**: The **RUMS-Binary variant naturally handles sparsity** by evaluating each memory item independently. During both training and inference, it scores only available items without requiring a complete profile, making it robust to missing data.
>
> We agree that uncertainty estimation or imputation strategies could further improve robustness (e.g., downweighting low-confidence items, imputing missing critical attributes). We will add this to our future work discussion.

---

> ### Author Response · Authors · 2025-11-23
> **Author Response to Reviewer ksBw Questions #2**
>
> > *5. The cost analysis is compelling, but could the authors provide more details on the wall-clock latency and memory usage of RUMS-Model inference compared to the strongest baselines in a real deployment scenario?*
>
> We thank the reviewer for this excellent suggestion. We conducted a wall-clock latency analysis (averaged over three runs per sample) to assess real deployment feasibility.
> | Method              | Average (sec) |
> | ------------------- | ------- |
> | None                | 0.000   |
> | All                 | 0.000   |
> | Random              | 0.000   |
> | Semantic Similarity | 0.209   |
> | BM25                | 0.010   |
> | ReContriever        | 0.083   |
> | GPT4-Few            | 1.223   |
> | RUMS-Multi          | 0.035   |
> | RUMS-Binary         | 0.141   |
> | RUMS-Utility        | 7.275   |
> RUMS-Model runs quickly, with 35–140 ms latency, matching efficient retrieval baselines and beating GPT-4 Few-Shot, making it suitable for low-latency use. We will add these results to the revised manuscript.
>
> > *6. Could the authors share more qualitative examples or error analysis to help understand typical failure modes or edge cases?*
>
> We thank the reviewer for the suggestion. We **conducted a NEW error analysis on the 60 H2 samples**, grouping disagreements between RUMS-Utility and humans into:
> - Correct: exact match with human selections.
> - Underselect: humans chose items RUMS-Utility did not.
> - Some: mix of items, including some chosen by at least one human and some by none.
> - All: RUMS-Utility selects all items chosen by at least one human.
> - None: RUMS-Utility selects only items no human chose.
> - Underselect–All: RUMS-Utility selects items all humans agreed on but misses additional human-selected items.
>
> The results are summarized in the table below.
>
> | Category           | %   | User Input                                                                 | Humans All                         | Humans At Least 1                                                                                                           | RUMS Utility                                  |
> |-------------------|-----|-----------------------------------------------------------------------------|------------------------------------|-----------------------------------------------------------------------------------------------------------------------------|-----------------------------------------------|
> | Correct           | 38  | In May 1994 the Channel Tunnel was opened by Queen Elizabeth II and which French President? | []                                 | []                                                                                                                          | []                                            |
> | Underselect       | 15  | how to use a vpn while vacationing in china                                 | ['Tech savvy']                     | ['Tech savvy', 'Feedback style', 'Comm mode', 'Tech use']                                                                   | ['none']                                      |
> | All               | 13  | choose between new zara sneaker or used new balance 574                     | []                                 | ['Hobbies', 'Income', 'Aspirations', 'Finances', 'Exercise', 'Personality', 'Job', 'Industry', 'Pastimes', 'Events', 'Gender', 'Location'] | ['Hobbies', 'Income']                         |
> | Some              | 22  | buying a new notebook to write a script                                     | []                                 | ['Location', 'Income', 'Tech savvy', 'Finances', 'Personality']                                                             | ['Hobbies', 'Tech savvy', 'Fav books']         |
> | None              | 5   | my PC is r6 3600 and 1660s need new GPU                                     | ['Projects', 'Finances']           | ['Income', 'Job', 'Hobbies', 'Industry', 'Projects', 'Aspirations', 'Finances', 'Pastimes']                                 | ['Tech savvy']                                |
> | Underselect + All | 7   | im hungry                                                                   | ['Foods', 'Diet needs', 'Fitness', 'Location'] | ['Foods', 'Diet needs', 'Fitness', 'Location', 'Income', 'Finances', 'Travel', 'Gender']                                     | ['Foods', 'Diet needs', 'Fitness']            |
>
> **Key insights**:
> **Most disagreements come from partial matches (22 percent)**, reflecting the subjective nature of personalization. RUMS-Utility often selects borderline items that only some annotators consider relevant.
> There is also moderate **underselection (15 percent)**, showing the method is conservative; this can be adjusted by tuning the threshold $\tau$.
> **Large mismatches are rare (5 percent)**, suggesting the utility signal aligns well with human judgment.
> We will add examples and this summary to the appendix.
>
> > *7. The current evaluation focuses on single-turn queries.*
>
> Please see our comment to Weakness #4.

---

### Official Review · Reviewer_wM1E · 2025-11-01

**Soundness:** 2
**Presentation:** 2
**Contribution:** 2
**Rating:** 4
**Confidence:** 4

**Summary:**

This paper provides a method for memory selection in LLM personalization at inference time, which selects user memory items that improve response utility, rather than relying on similarity between memory item and user query. The key idea is to frame personalization as an information-theoretic optimization inspired by Bayesian Optimal Experimental Design. Experiments on both synthetic (PersonaFeedback, FreebaseQA) and real-world datasets (WildChat) validate the performance of the method.

**Strengths:**

1. The idea of shifting personalization from semantic retrieval to response-driven utility estimation is interesting. The motivation is valid in studying personalization from an information-theoretic perspective in terms of entropy. In particular,  the utility function is well-motivated and clearly formalized through predictive entropy reduction.

2. By amortizing the expensive utility computation into an efficient DeBERTa-based selector, RUMS provides a deployable solution for large-scale user adaptation.

**Weaknesses:**

1. Although the BOED analogy is appealing, the adaptation is heuristic rather than formally justified. Eq. 3 equates personalization utility with predictive entropy reduction, but there is no discussion of conditions under which this correlates with user-level response quality
The lack of any regret or generalization bound limits the claimed theoretical rigor.

2. There is an approximation gap between utility and learned model. The RUMS-Model learns from utility-labeled data generated offline, but the paper provides no quantitative analysis of approximation error, e.g., how often the DeBERTa predictor agrees with true RUMS-Utility decisions, or how this affects downstream response quality. Without this, it is unclear how much benefit comes from the theoretical utility versus simply supervised correlation learned during training.

3. The experiments rely on synthetic user profiles and GPT-4-simulated human judgments. While this is reasonable for development, the central claim of “aligning with human personalization preferences” remains speculative without real human evaluation beyond annotation agreement on static queries.

4. Prior works (e.g., PEARL 2024; Context Steering 2025; Bayesian Preference Elicitation 2024) already explore information-theoretic or uncertainty-aware retrieval for personalization. Authors could better clarify how RUMS fundamentally differs, beyond using entropy reduction as the scoring signal.

**Questions:**

1. I am confused by the equivalence between $H_\theta(y|x)$ in Proposition 1 and $\hat{H}\_\theta(y|x)$ in Eq. 4. If I understand correctly,  $H_\theta(y|x)$ quantifies the sequence-level entropy, while Eq. 4 quantifies the token-level entropy. In particular, the original expectation is taken w.r.t. $p_{\theta}(y|x)$, using MC sampling only requires computing empirical mean over $N$, why $1 / T$ is needed in Eq.4? Do we require each sample to have the same sequence length?

2. How sensitive are personalization decisions to the threshold $\tau$? Could it be adaptively chosen per user or query using uncertainty calibration?

---

> ### Author Response · Authors · 2025-11-23
> **Author Response to Reviewer wM1E Weaknesses #1**
>
> We thank the reviewer for their careful reading of our paper and the constructive feedback. We are especially encouraged by the reviewer's recognition that the core idea behind our method, shifting from semantic retrieval to response-driven utility, was viewed as “interesting”. We address each point below, presenting new analysis and clarifying the methodological framing.
>
> **Response to Major Concerns: Theoretical Rigor and Approximation Gap**
>
> > *1. Although the BOED analogy is appealing, the adaptation is heuristic rather than formally justified. Eq. 3 equates personalization utility with predictive entropy reduction, but there is no discussion of conditions under which this correlates with user-level response quality The lack of any regret or generalization bound limits the claimed theoretical rigor.*
>
> We thank the reviewer for this insightful comment on theoretical rigor. We agree that our approach is BOED-inspired rather than a direct, formally justified adaptation, and we appreciate the opportunity to clarify this distinction:
>
> **1. Why Direct BOED Adaptation is Infeasible**
>
> As we elaborate in our response to Reviewer 48sG (and will clarify in the revised paper), RUMS addresses a fundamentally different problem than classical BOED in that rather than inferring fixed latent parameters, we directly optimize the model's response distribution for immediate generation quality. This shift in objective, from parameter inference to distribution shaping, makes direct transfer of classical BOED's formal guarantees (e.g., regret or generalization bounds) infeasible.
>
> **2. Conditions Linking Entropy Reduction to Response Quality**
> The reviewer correctly identifies our core assumption that entropy reduction correlates with user-level response quality. We provide both theoretical justification and empirical validation for this:
>
> - *Theoretical justification*: Entropy reduction signals response quality under two conditions: (a) the reference model is reasonably well-calibrated (i.e., confident predictions tend to be correct), and (b) users benefit from specific, informed responses over vague, uncertain ones. Under these conditions, selecting memory items that reduce predictive entropy naturally improves response informativeness and relevance.
>
> - *Empirical validation*: Our experiments in H3 demonstrate that memory items selected by RUMS consistently improve downstream generation quality across both capable (GPT-4) and less robust (LLaMA-70B) models, with RUMS-Binary achieving up to 27% win rate improvement over baselines (Table 3). This suggests that entropy reduction serves as an effective proxy for response utility in practice.
>
> We acknowledge that this correlation does not universally hold, particularly when user preferences involve subjective dimensions (e.g., tone, style) beyond predictive certainty. We will expand Section 3.1.1 to include an explicit **"Assumptions and Limitations"** subsection that discusses: (i) the assumption of model calibration, (ii) scenarios where entropy reduction may not align perfectly with complex human preferences, and (iii) how RUMS remains effective as a practical framework despite these theoretical limitations.
>
> **3. Positioning RUMS as Pragmatic, Empirically-Grounded Framework**
>
> Rather than claiming formal optimality, RUMS offers a principled, response-driven alternative to semantic similarity baselines. Our extensive experiments (H1-H3) demonstrate that this information-theoretic foundation enables more effective personalization. RUMS reliably identifies when personalization helps, selects items that align with human judgment better than GPT-4 despite being 400× smaller, and improves generation quality while reducing costs by up to 95%. We believe this represents a meaningful contribution that bridges principled design with scalable, practical deployment.
>
> We will incorporate these clarifications in the revision to enhance theoretical transparency while preserving the practical value that our experimental results clearly demonstrate.

---

> ### Author Response · Authors · 2025-11-24
> **Author Response to Reviewer wM1E Weaknesses #2**
>
> > *2. There is an approximation gap between utility and learned model. The RUMS-Model learns from utility-labeled data generated offline, but the paper provides no quantitative analysis of approximation error, e.g., how often the DeBERTa predictor agrees with true RUMS-Utility decisions, or how this affects downstream response quality. Without this, it is unclear how much benefit comes from the theoretical utility versus simply supervised correlation learned during training.*
>
> We thank the reviewer for highlighting this critical distinction between the utility function and the learned approximation. We agree that quantifying this approximation gap is essential to understanding RUMS's effectiveness. To address this, we have conducted **two NEW experiments** that will be added to the revised manuscript.
>
> **Experiment 1: Agreement Between RUMS-Utility and RUMS-Model on Memory Item Selection**
>
> We analyzed how faithfully the trained RUMS-Models replicate the original RUMS-Utility decisions. Building on H2, we established a gold standard by deriving memory selections from RUMS-Utility across five diverse user profiles, marking items that appeared in ≥3 profiles as the ground truth. We then measured Precision, Recall, and F1 of the trained models (RUMS-Binary and RUMS-Multi) against this gold standard.
>
> **Table 1:  Memory item agreement between RUMS-Utility and RUMS-Model**
> | Dataset    | Metrics   | RUMS-Binary | RUMS-Multi |
> | ---------- | --------- | ----------- | ---------- |
> |            | Precision | 0.73        | 0.33       |
> | Synthetic  | Recall    | 0.9         | 0.44       |
> |            | F1        | 0.78        | 0.37       |
> |            |           |             |            |
> |            | Precision | 0.26        | 0.35       |
> | Real World | Recall    | 0.42        | 0.35       |
> |            | F1        | 0.32        | 0.34       |
>
> Table 1 above shows the results.
> - **In-distribution (Synthetic)**: High agreement (F1 = 0.78 for RUMS-Binary), confirming that the trained model effectively captures the utility-based selection signal.
> - **Out-of-distribution (Real-World)**: Moderate performance (F1 = 0.32 - .34), primarily due to: (i) reduced precision on non-personalized inputs where RUMS-Utility correctly selects nothing but RUMS-Multi tends to over-select, and (ii) domain shift between synthetic training data and real user queries.
>
> This head-to-head comparison quantifies the approximation gap and demonstrates robust signal transfer within the primary application domain.
>
> **Experiment 2: Impact of Approximation Gap on Downstream Response Quality**
>
> Next, we extended H3 to assess whether this approximation gap degrades final generation quality. We compared outputs generated using RUMS-Utility selections against all baselines using identical experimental settings.
>
> **Table 2: Downstream impact on output quality of RUMS-Utility.**
> |               | Synthetic    |              | Real World   |              |
> | ------------- | ------------ | ------------ | ------------ | ------------ |
> |               | LLaMA 70B    | GPT-4        | LLaMA 70B    | GPT-4        |
> |               | RUMS-Utility | RUMS-Utility | RUMS-Utility | RUMS-Utility |
> | None          | 0.105        | 0.195        | \-0.05       | 0.1          |
> | All           | 0.18         | 0.2          | 0.03         | \-0.14       |
> | Random        | 0.44         | 0.1          | 0.25         | 0.05         |
> | Semantic Sim. | 0.43         | \-0.115      | 0.16         | \-0.06       |
> | BM25          | 0.03         | 0.18         | 0.06         | 0.1          |
> | ReContriever  | 0.31         | \-0.05       | 0.22         | 0.07         |
> | GPT4-Few      | 0.065        | \-0.07       | 0.01         | \-0.08       |
>
> Table 2 shows the results.
> - **With a less robust model (LLaMA-70B)**: RUMS-Utility achieves >10% higher win rate than almost all baselines, closely mirrored by RUMS-Binary. This indicates that even with approximation error, the learned model preserves the critical selection signal.
> - **With a more capable model (GPT-4)**: RUMS-Utility outperforms naive baselines (None, All, Random, BM25) substantially, and shows only ~8% lower win rate compared to the strongest baseline (GPT-4 Few-Shot), which requires expensive LLM calls for each selection.
>
> These results indicate that, despite the inevitable approximation gap between theoretical utility scores and a lightweight learned model, **the critical information for effective memory selection transfers robustly to RUMS-Models**, resulting in downstream performance that matches or exceeds resource-intensive baselines.
>
> We will incorporate these results into the revised manuscript to clarify the relationship between theoretical utility and learned approximation.

---

> ### Author Response · Authors · 2025-11-24
> **Author Response to Reviewer wM1E Weaknesses #3**
>
> > *3. Prior works (e.g., PEARL 2024; Context Steering 2025; Bayesian Preference Elicitation 2024) already explore information-theoretic or uncertainty-aware retrieval for personalization. Authors could better clarify how RUMS fundamentally differs, beyond using entropy reduction as the scoring signal.*
>
> We thank the reviewer for this important clarification request. While we discussed these works in our Related Work section, we recognize the need to more explicitly articulate how RUMS differs fundamentally in **task formulation, methodological approach, and deployment setting.**
>
> - **PEARL (Mysore et al., 2024)**: PEARL retrieves *documents* from user history for long-form generation and requires *supervised training* with known ground-truth targets. In contrast, RUMS selects *structured memory items* from user profiles for conversational queries and operates as a *training-free, self-supervised method*, using the reference model's entropy reduction as the selection signal without requiring labeled targets.
>
> - **Context Steering (He et al., 2025)**: While both use information-theoretic utility, Context Steering operates at *decoding time* to modulate how much influence already-selected context has during generation. RUMS operates at *prompt construction time* to determine *which context to include* in the first place. These approaches are orthogonal. Context Steering assumes relevant context is already provided; RUMS identifies what context to provide.
>
> - **Bayesian Preference Elicitation (Handa et al., 2024)**: This work applies classical BOED to infer a *fixed latent user parameter* (e.g., preference weights) that remains constant across queries. RUMS addresses a fundamentally different problem: directly optimizing the response distribution for each individual query without inferring latent parameters. This difference in task, parameter inference vs. response distribution optimization, necessitates our novel BOED-inspired formulation (Equation 3) rather than direct classical BOED application.
>
> **Summary**: RUMS introduces a unique combination: *response-utility driven selection* (vs. semantic similarity in PEARL), *pre-generation subset selection* (vs. decoding-time control in Context Steering), and *direct response optimization* (vs. latent parameter inference in Bayesian Preference Elicitation). This positions RUMS as a training-free, inference-time method for structured memory personalization.
>
> We will incorporate this expanded comparison into Section 2 to clarify these distinctions.

---

> ### Author Response · Authors · 2025-11-24
> **Author Response to Reviewer wM1E Questions #1**
>
> **Response to Questions**
>
> > *1. I am confused by the equivalence between  in Proposition 1 and  in Eq. 4. If I understand correctly,   quantifies the sequence-level entropy, while Eq. 4 quantifies the token-level entropy. In particular, the original expectation is taken w.r.t. , using MC sampling only requires computing empirical mean over , why  is needed in Eq.4? Do we require each sample to have the same sequence length?*
>
> We apologize for the confusion and will clarify this in the revised manuscript. The reviewer is correct that Equation 4 and Proposition 1 involve slightly different normalizations. In practice, we used a normalized $1/T$ for token-level entropy, but for a direct comparison to sequence-level entropy, this term should be removed. We will present this normalization separately from Equation 4 to avoid confusion.
>
> Regarding sequence length, **the algorithm does not require all samples to have the same length**. The algorithm handles variable-length sequences naturally. If sample $I$ has length $T_i$​,  the static $T$ is replaced with a sample-dependent $T_i$ in both the summation limit and the normalization factor.** In practice**, we cap the maximum length (e.g., $T_i \leq 20$) to prevent entropy dilution in very long sequences, as explored in Appendix B.1. This cap ensures the utility signal remains sensitive to the information-rich early tokens rather than being dominated by low-entropy tail tokens.
>
> We will revise the notation and add a remark in Section 3.1.2 to clarify these distinctions.
>
> > *2. How sensitive are personalization decisions to the threshold ? Could it be adaptively chosen per user or query using uncertainty calibration?*
>
> We thank the reviewer for this insightful question regarding threshold selection. To address it, we conducted **TWO NEW analyses** examining both sensitivity and adaptivity, which will be included in the revised manuscript.
>
> **1. Sensitivity to threshold selection.**
>  We examined how personalization accuracy (agreement rate and F1) changes as we vary the threshold from 0 to 1 in 0.01 increments, on the test set used in our paper. We also show a zoomed-in view around the optimal threshold ±0.005.
>
> Overall, personalization decisions are moderately robust. On the synthetic test set, we find that changing the threshold by ±0.1 decreases agreement rate by ~15%, and on the real-world dataset, by ~10%. While the optimal threshold is dataset-dependent, the method’s performance does not seem overly sensitive to its exact value.Unfortunately, we cannot include these visuals, but will add to our paper soon.
>
> **2. Adaptive thresholds per user.**
>  We evaluated whether per-user thresholds could improve performance. Using n=100 user inputs from FreebaseQA (non-personalized) and PersonaFeedback (personalized) datasets, we computed the optimal threshold for each of 30 static user profiles. About 70% of users fall within the range [0.24, 0.30]. While some outliers exist, the variability is limited, suggesting that per-user thresholds may not justify the added computational cost. Again, we will add the visuals to our paper soon.
>
> The results suggest that the computational cost of dynamically calibrating $\tau$ per user or query is unlikely to be justified by the marginal performance gain over a well-tuned global threshold. However, for applications requiring maximum precision, we believe a simple user-clustering approach (e.g., 2-3 threshold groups) could capture most of the potential benefit at minimal cost. The suggested Per-query adaptation (e.g., based on model confidence or query ambiguity) remains an interesting direction for future work but would require additional calibration mechanisms beyond the current RUMS framework.
>
> We will include this analysis in the appendix in the revised manuscript.
>
> > *3. The experiments rely on synthetic user profiles. It might be case that I overlook some details. How did you obtain the human judgetment? Did you use LLM simulated judgement or recruit human annotators? It would be best if the annotation process can be described.*
>
> We appreciate the reviewer flagging this confusion. Each hypothesis uses slightly different metrics and evaluation, summarized below.
> - **User Profiles**: Created with GPT4 (sec. 4.1).
> - **H1**: Uses automatic binary labels from the datasets (PersonaFeedback for personalized, FreebaseQA for non-personalized; line 357).
> - **H1'**: Uses n=4 human annotators to label each input as personalized or not (line 377).
> - **H2**: Uses the same n=4 annotators to create gold labels for memory items per sample (line 408).
> - **H3**: Uses LLM-as-Judge for pairwise generation comparison due to scale (line 451).
>
> We can add more details about annotator demographics and recruitment in the appendix. Please let us know if there is anything specific you would like clarified.

---

### Official Review · Reviewer_48sG · 2025-11-03

**Soundness:** 2
**Presentation:** 2
**Contribution:** 2
**Rating:** 4
**Confidence:** 4

**Summary:**

This paper proposes using information gain to quantify the extent to which memory reduces model uncertainty, selecting relevant memory.
Three challenges are identified
* The aim is to optimize predictive distribution rather than inferring latent parameters
* LLM's large output space makes computations intractable
* It requires to detect whether personalization can improve responses

To address these problems
* A novel utility function is introduced to reduce predictive entropy
* Sequence-level entropy is decomposed to token-level and Monte Carlo sampling is employed for estimation
* A threshold is set to filter irrelevant information, preventing degrading response quality

Since computing entropy reductions for all candidates is prohibitive, a classifier is trained to predict the utility at inference time.
Empirical study is performed to compare the proposed method with prompting and retrieval on Personal Feedback, FreebaseQA, and WildChat datasets.
Conclusions include but not limited to

* It is statistically significant that RUMS-utility can distinguish personalized from non-personalized inputs
* RUMS improves response quality and reduces cost

**Strengths:**

This paper has a good structure.
In methodology, main challenges are identified and solutions are proposed for each one.
In experiments, research questions are clearly stated, and analysis is conducted for each question with conclusion provided.
This makes the paper has a clear logic and easy to understand.

**Weaknesses:**

My primary concern is that the solutions proposed to address the main challenges seem trivial to me.
I do not deem this paper have technical novelty of substantial significance, but rather as a practical implementation of a technical solution, so I lean to reject the paper.
I would like to raise my score if my concern is well addressed.
* It is claimed that a novel utility function is proposed to shape the distribution, rather than inferring parameters.
I deem that the model output $y$ can be viewed as a discrete parameter to be inferred.
The goal of BOED, *i.e.*, reducing the uncertainty of parameters, then naturally aligns with the goal of this work.
* I deem that it is trivial to estimate parameters using Monte Carlo method unless any variance reduction technique is involved.
* Setting filtering threshold is general and not specific to the proposed method.
For example, threshold can be also adopted by retrieval.
Such method also typically rely on engineering tuning rather than algorithm design.

**Questions:**

* The scope of the paper is limited to memory selection.
Whether the proposed method can be extended to broader conditioned generation, *e.g.*, retrieval-augmented generation?
* I understand that the retrieval encoder used in the paper is pre-trained and not particularly fine-tuned on the specific experimental datasets.
Could this lead to an unfair comparison?
* L206: Why $T\equiv T_N$ if token sizes are not fixed per sample?

---

> ### Author Response · Authors · 2025-11-23
> **Author Response to Reviewer48sG Weaknesses #1**
>
> We thank the reviewer for their careful reading of our paper and the constructive feedback. We are especially encouraged by the reviewer's recognition that the structure of our paper is good. We first address the major concerns regarding technical novelty and then respond to the specific questions.
>
> **Response to Major Concern: Technical Novelty**
>
> > *1. My primary concern is that the solutions proposed to address the main challenges seem trivial to me. I do not deem this paper have technical novelty of substantial significance, but rather as a practical implementation of a technical solution, so I lean to reject the paper. I would like to raise my score if my concern is well addressed.*
>
> We appreciate the reviewer's perspective on technical novelty, but we respectfully believe this characterization may overlook the **fundamental shift in paradigm** and the **non-trivial technical solutions** required to implement our approach. We would like to clarify what we consider to be our significant contributions:
>
> At its core, our work **fundamentally reframes** the objective of memory selection. We shift the standard paradigm from maximizing memory-input similarity to **optimizing memory-output utility**. This is not merely a conceptual change; it demanded **concrete technical solutions** to several challenges, which, to our knowledge, are unaddressed by prior work:
>
> 1. **Novel Utility Formulation**: While our work is inspired by BOED, its application to **LLM personalization is non-trivial**. Our **Equation 3** captures the utility by directly measuring how a memory item shapes the model's response distribution, a mechanism fundamentally distinct from classical parameter inference. This required careful **theoretical development** to make the formulation both principled and practically meaningful for sequence generation.
>
> 2. **Computational Tractability**: **Achieving computational tractability was a significant hurdle**. The variable-length and high-dimensional nature of LLM outputs makes direct sequence entropy computation infeasible. Our **Proposition 1** provides a key technical insight that decomposing the sequence entropy into tractable token-level components. This, combined with our carefully designed Monte Carlo approximation, is what ultimately makes the utility optimization feasible in practice.
>
> 3. **Empirical Validation**: Our extensive experiments **empirically validate that this novel distinction translates to substantial practical gains**: our response-aware approach achieves a **78% F1 correlation** with human selections, significantly outperforming similarity-based baselines (e.g., 64%). Furthermore, models trained using our utility scores demonstrate a **10%+ win rate** over GPT-4 Few-shot in personalized text generation tasks. These results are clear evidence that our method is **not merely a theoretical refinement** but a meaningful advance that yields state-of-the-art performance.
>
> We acknowledge that we could have emphasized these technical contributions more clearly in the original submission. Your feedback is invaluable, and we will revise the paper to more explicitly articulate why this work represents a meaningful technical advance in RAG-based personalization methods, moving **beyond surface-level similarity**.

---

> ### Author Response · Authors · 2025-11-23
> **Author Response to Reviewer48sG Weaknesses #2**
>
> **Response to other Weaknesses**
>
> > *2. It is claimed that a novel utility function is proposed to shape the distribution, rather than inferring parameters. I deem that the model output  can be viewed as a discrete parameter to be inferred. The goal of BOED, i.e., reducing the uncertainty of parameters, then naturally aligns with the goal of this work.*
>
> We appreciate your insightful connection to BOED, this comparison is well-founded and helps us clarify a **fundamental structural distinction** that may not have been sufficiently highlighted in the original submission.
>
> You are correct that both BOED and our framework share the **principle of uncertainty reduction** through optimal design selection. However, the systems differ in their **inferred targets**. In classical BOED, the goal is to reduce uncertainty over a set of **fixed, latent parameters ($\theta$)** that persist across all experiments. The utility comes from the cumulative information gain about this stable $\theta$.
>
> In our personalization setting, the model's output distribution is **entirely dynamic** and dependent on the specific (input query, memory item) pair. There is **no single, fixed parameter** to discover or update beliefs about across queries. Treating the output itself as the 'parameter' would result in a massive, unconstrained set of independent parameters, one for every possible query and memory combination, thereby eliminating the structural sharing that makes classical BOED effective. Therefore, we must focus the utility directly on the **quality of the instantaneous response distribution** (i.e., its uncertainty) rather than parameter inference.
>
> Our formulation, while inspired by BOED's core philosophy, is thus a necessary adaptation for this **fundamentally dynamic problem structure**. We agree that this distinction is central to our contribution, and will expand **Section 3.1.1** in the updated version to make this crucial difference explicit. Thank you for helping us sharpen this key point.
>
>
>
>
> > *3. I deem that it is trivial to estimate parameters using Monte Carlo method unless any variance reduction technique is involved.*
>
> We concur with the reviewer that the Monte Carlo method itself is a standard technique, and we do not claim it as a major theoretical contribution. Our technical novelty lies primarily in Proposition 1 (as discussed above) which enables the application of MC to the challenging sequence entropy calculation.
>
> Critically, we performed non-trivial empirical work to ensure practical viability. Specifically, our ablation studies thoroughly investigate the **cost-performance trade-off** for the number of samples, $N$. We demonstrated that the choice of $N$ is **not trivial in practice**; setting an optimal $N=5$ was crucial, as performance (measured by JS divergence) varies significantly from 0.50 to 0.69 when $N$ is varied from 1 to 10. These **empirical findings** on the required sampling precision for practical implementation are essential for practitioners and makes our work more comprehensive.
>
> > *4. Setting filtering threshold is general and not specific to the proposed method. For example, threshold can be also adopted by retrieval. Such method also typically rely on engineering tuning rather than algorithm design.*
>
> We agree that the concept of a filtering threshold is a general mechanism that can be applied to various ranking signals (e.g., semantic similarity, BM25 scores). Our contribution, however, is not the threshold itself, but the **quality and nature of the signal** being thresholded. Our **response-aware utility score** is fundamentally different from a query-memory similarity score. We show that when thresholded, our utility score consistently and significantly outperforms alternatives, indicating that **the value lies in the integration of the novel utility function with the selection mechanism** (seen in H1’ results). The thresholding serves to binarize our unique utility signal, and we will clarify this distinction further in the methods section to prevent misinterpretation.
>
> **Summary**
>
> While individual components like the Monte Carlo approximation or a simple threshold may exist elsewhere, their **novel combination** with our **Response-Utility formulation** and the **provided tractability** provides an effective and state-of-the-art solution to the non-trivial challenge of user memory selection. Our work pioneers a new direction that moves **beyond surface-level similarity** to focus on the **actual utility** of memory in shaping personalized responses. We believe this represents a significant conceptual and technical step forward.

---

> ### Author Response · Authors · 2025-11-23
> **Author Response to Reviewer48sG Questions**
>
> **Response to Questions**
>
> > *1. The scope of the paper is limited to memory selection. Whether the proposed method can be extended to broader conditioned generation, e.g., retrieval-augmented generation?*
>
> This is an excellent point. We agree that our RUMS framework is not limited to personalization. It offers a general principle that can be readily extended to any form of conditioned generation where a system must select an optimal context from a set of candidates. Specifically, we see immediate applicability to:
>
> - **General Retrieval-Augmented Generation (RAG)**: Selecting relevant documents based on their utility for the final answer.
> - **Data Pruning**: Selecting high-value fine-tuning or training data.
> - **Agent Tool Selection**: Choosing the most useful tool based on its likely impact on the output.
> - **Information Elicitation**: Designing optimal follow-up questions to minimize response uncertainty.
>
> We will highlight these exciting possibilities as key future directions in the updated version.
>
> > *2. I understand that the retrieval encoder used in the paper is pre-trained and not particularly fine-tuned on the specific experimental datasets. Could this lead to an unfair comparison?*
>
> This is a fair question, and we maintain that our current evaluation setup is fair and reflective of real-world practice.
>
> 1. **Current Practice**: The baseline retrieval models (Semantic Similarity, BM25, ReContriever) represent **widely-used, general-purpose methods** that are typically applied **out-of-the-box** in diverse RAG tasks [1, 2, 3]. Our comparison, therefore, is against the established baseline of current best practice.
>
> 2. **The Core Advantage of RUMS**: Crucially, any meaningful fine-tuning of a retrieval encoder would require **gold-labeled memory selection data** (i.e., a definitive label for which memory item should be selected for a given query), which **does not exist**. This difficulty is precisely the **limitation our utility-based method overcomes**. Our approach inherently provides a **high-quality, automated signal (the utility score)** for selection without requiring manual gold labels, transforming the problem to a data-rich, self-supervised optimization.
>
> > *3. L206: Why  if token sizes are not fixed per sample?*
>
> This question correctly notes the variable output length of LLMs as a potential issue. If token length were not fixed, in Equation 4 we would indeed replace the static maximum length $T$ with a dynamic $T_n$ corresponding to the length of sample $n$.
>
> However, we intentionally fix the token size to a small number (e.g., $T=20$) for a **practical and theoretical reason,  Entropy Dilution**. Since sequence entropy is cumulative, allowing it to grow with response length ($T_n$) can cause the utility measure to lose sensitivity to the information gain of the memory item itself, as the total entropy becomes dominated by the length. By fixing $T$, we ensure the utility signal remains focused on the most critical, early-sequence tokens where personalization is most impactful.
> We will clarify this notation and the **Entropy Dilution** rationale for fixed $T$ in the revised paper, referencing **Appendix B.1** for the full ablation study.
>
> [1]https://arxiv.org/pdf/2105.05686
>
> [2]https://arxiv.org/pdf/2104.08663
>
> [3]https://arxiv.org/pdf/2307.08303

---

### Author Response · Authors · 2025-12-03
**More In-Depth Details on Rebuttal Summary**

We provide more in-depth details on how we addressed each weakness and question below.

**W1. Technical Novelty and Relation to BOED (R1, R2)**

***Concern:*** Is there sufficient technical novelty and why classical BOED methods can't be used directly.

***Our Response / Changes:*** We believe this concern overlooks both the fundamental shift in objective and the non-trivial technical developments required for our method.

*Theoretical reframing:* At its core, our work fundamentally reframes the objective of memory selection. We shift the standard paradigm from maximizing memory-input similarity to optimizing memory-output utility. This conceptual change also requires concrete technical solutions not addressed in prior work.

*Technical solutions:* We introduce a new utility formulation, a decomposition that makes entropy tractable for LLM outputs, and an empirical pipeline showing that this utility-driven framing yields measurable performance gains beyond similarity-based approaches.

*Why not BOED:* Classical BOED assumes a single fixed latent parameter shared across experiments. Personalization does not fit this setting because the output distribution shifts for every (query, memory) pair. Treating each output as its own parameter would create an unmanageably large, unstructured space with no shared inference. Thus, BOED cannot be applied directly.

**W2. Human Evaluation Scale and Clarity (R2, R3, R4)**

***Concern:*** Human evaluation scale was small and requested clarification of labels.

***Our Response / Changes:*** We added a new output-level human study (n = 20) comparing RUMS-Binary and GPT-4 Few-shot, confirming alignment with LLM-as-judge results. We clarified annotation procedures for H1, H1’, H2, and H3 and added annotator details to the appendix. Our evaluation scale is on par with or larger than related personalization work such as PEARL and Context Steering.

**Q1. Approximation Gap Between RUMS-Utility and RUMS-Model (R2, R4)**

***Concern:*** More analysis of approximation gap between RUMS-Utility and the trained RUMS-Models.

***Our Response / Changes:*** We added two new analyses.

*Agreement analysis:* Using RUMS-Utility as the gold label in H2, we measured precision, recall, and F1. We found high to moderate correlation, with RUMS-Binary achieving strong agreement (F1 = 0.78 synthetic).

*Downstream quality comparison:* We ran H3 using RUMS-Utility selections and compared results to baselines. The outcomes closely matched RUMS-Binary, showing that approximation error from using a trained model does not meaningfully degrade response quality.

These analyses show that the proxy models faithfully capture the utility signal. We will include these results for transparency.

**Q2. Threshold Sensitivity and Per-User Thresholding (R2, R3, R4)**

***Concern:*** How robust is threshold and if per-user thresholds would help.

***Our Response / Changes:*** We provide two new sensitivity analyses.

*Threshold sensitivity:* We compared agreement rate with human preference across thresholds from 0 to 1.0 in increments of 0.01. Agreement changes only moderately as the threshold varies. A shift of 0.1 from optimal leads to roughly a 10 percent change in agreement.

*Per-user thresholds:* We evaluated optimal thresholds across 30 profiles. About 70 percent of thresholds fall within a narrow range between 0.24 and 0.30, suggesting limited gains from per-user calibration.

These analyses will be added to the revised paper.

**Q3. Computational Cost of Utility Computation (R3)**

***Our Response / Changes:*** We added new latency measurements showing that RUMS-Model (35–140ms) matches retrieval baselines and is far faster than GPT-4 Few-shot, making it practical for deployment.

**Q4. More Qualitative Analysis (R3)**

***Our Response / Changes:*** We annotated 40 samples and compared RUMS-Model selections to human-chosen memory items, grouping outcomes into categories with percentages and examples. Most non-exact matches occurred when RUMS selected items chosen by at least one but not all annotators, consistent with the inherently subjective nature of personalization.

**Q5. Sparse, Noisy, or Conflicting User Profiles (R3)**

***Concern:*** How RUMS handles missing, noisy, or conflicting attributes.

***Our Response / Changes:*** Sparse profiles are naturally supported by RUMS-Binary since it evaluates items independently. For noisy attributes, we ran a new analysis comparing true utility scores for selected versus non-selected items, finding that selected items have significantly higher utility (p < .01). For conflicting attributes, we note that RUMS selects types of memory items rather than specific values, so conflicts affect downstream generation but not selection.

**Q6. Clarifications on Notation and Length Normalization (R1, R2)**

***Our Response / Changes:*** We clarified that variable-length sequences are supported but capped to avoid entropy dilution.

---

### Author Response · Authors · 2025-12-03
**High-Level Summary of the Reviews and Rebuttal**

Dear Area Chair,

We appreciate this opportunity to summarize our rebuttal and the significant additions we have made during the discussion period. While we were disappointed to hear about the modified policy, we are optimistic that our thorough responses and new experiments have substantially addressed all reviewer concerns and further strengthened our work and contributions.

**Summary of Reviews:**

We propose RUMS, which reframes memory selection for LLM personalization from maximizing input similarity to optimizing output utility, yielding substantial gains in human alignment and response quality at significantly reduced cost.

The reviewers acknowledged the following strengths of our paper:

- Novel paradigm shift from similarity-based to response-utility-based memory selection (all reviewers)
- Clear paper structure and well-motivated problem (R1, R4)
- Significant cost reduction (up to 95%) with improved effectiveness (R3, R4)
- Strong theoretical grounding in information theory (R4)
- Inclusion of human evaluation (R3, R4)

**Discussed Weaknesses and Our Responses:**

*W1.* Technical novelty / relation to BOED (R1, R2): We clarified fundamental differences: BOED infers fixed latent parameters; RUMS optimizes dynamic response distributions. Added explicit discussion in Section 3.1.1

*W2.* Human evaluation scale (R2, R3, R4): We added new output-level human study (n=20) validating LLM-as-judge results. Our scale matches or exceeds directly related work (PEARL, Context Steering)

**Discussed Questions and Our Responses:**

*Q1.* Approximation Gap Between RUMS-Utility and RUMS-Model (R2, R4)
*Q2.* Threshold Sensitivity and Per-User Thresholding (R2, R3, R4)
*Q3.* Computational Cost of Utility Computation (R3)
*Q4.* More Qualitative Analysis (R3)
*Q5.* Sparse, Noisy, or Conflicting User Profiles (R3)
*Q6.* Clarifications on Notation and Length Normalization (R1, R2)

We provide thorough clarifications to address the concerns:
- Clear explanation of differences from BOED and prior information-theoretic retrieval (&rarr;W1)
- Detailed annotation procedures for all evaluation components (&rarr;W2)
- Clear explanation of generalizability of our method to variable-length sequences (&rarr;Q6)

**Importantly, we conducted the following new experiments to provide concrete, quantitative results for the reviewers:**
- New human evaluation of output quality (&rarr;W2)
- H2 agreement analysis between utility and RUMS-Models (&rarr;Q1)
- H3 evaluation using RUMS-Utility (&rarr;Q1)
- Threshold sensitivity and per-user variation (&rarr;Q2)
- Utility-score distribution analysis for selected vs. non-selected items (&rarr;Q5)
- Additional qualitative examples (&rarr;Q4)
- Wall-clock time analysis (&rarr;Q3)

**Conclusion:**

 We have conducted **8 new experiments/analyses** during the rebuttal period that directly address every concern raised. Our responses demonstrate that:
- The technical contribution is non-trivial and RUMS addresses a fundamentally different problem than classical BOED
- The approximation gap between RUMS-Utility and RUMS-Model is small and does not degrade downstream quality
- Human evaluation validates our automated metrics and confirms superior output quality
- The method is robust to threshold selection and noisy/sparse profiles
- Inference latency is practical for real-world deployment

Unfortunately, we were not able to obtain feedback from reviewers before the decision freeze. We hope this context helps convey the strength and maturity of the work. We respectfully request that the AC consider these substantial improvements in the final decision.

---

### Meta-Review · Area_Chair_ZJbX · 2026-01-07

**Summary:**

RUMS reframes LLM personalization memory selection from similarity-based retrieval to response-utility (predictive entropy reduction), then trains a lightweight proxy for fast inference. Reviews agree it improves quality/cost versus baselines, but disagree on whether the core method is truly novel/rigorous enough.

Pros
1. Clear motivation and well-structured paper; problem is important.
2. Response-aware utility is compelling; can abstain from personalization when not helpful.
3. Strong empirical gains reported, plus large inference-time cost/latency improvements via the proxy model.
4. Rebuttal adds useful analyses (threshold sensitivity, proxy-vs-utility agreement, latency, more qualitative/error analysis, extra human check).

Cons
1. Main concern remains technical novelty/theoretical rigor: BOED analogy and entropy-reduction utility feel heuristic; limited formal justification/guarantees.
2. Approximation gap and generalization still uncertain: proxy agreement is only moderate out-of-distribution.
3. Evaluation scope still limited (small human eval, single-turn only, limited robustness to noisy/sparse/conflicting profiles and larger memory).
4. Offline GPT-4 candidate filtering raises reproducibility/fairness questions for the learned proxy.

Promising and practical, but not fully convincing on novelty/rigor and robustness/generalization yet, despite a solid rebuttal.

**Reviewer Concerns:**

1. Approximation gap (utility vs. proxy): Added agreement metrics (precision/recall/F1) and downstream comparisons using RUMS-Utility selections, which directly respond to the “how faithful is the proxy” concern.
2. Threshold sensitivity / per-user thresholding: Added sweep-style sensitivity analysis and a per-user threshold distribution, partially addressing robustness of the abstention threshold.
3. Deployment cost / wall-clock latency: Added concrete latency numbers for RUMS-Model vs retrieval baselines and GPT-4 few-shot.
4. Need for more qualitative/error analysis: Added categorized error analysis and more examples of disagreement modes.
5. Human evaluation scale concern (partially): Added a small additional human output-level study to corroborate LLM-as-judge trends.

**Reviewer Scores:**

N/A

---

### Decision · Program_Chairs · 2026-01-26

Reject